# Particle emissions from a modern heavy-duty diesel engine as ice nuclei in immersion freezing mode: a laboratory study on fossil and renewable fuels

Kimmo Korhonen[1], Thomas Bjerring Kristensen[2], John Falk[2], Vilhelm B. Malmborg[3], Axel Eriksson[3], Louise Gren[3], Maja Novakovic[4], Sam Shamun[4], Panu Karjalainen[3,5], Lassi Markkula[5], Joakim Pagels[3], Birgitta Svenningsson[2], Martin Tunér[4], Mika Komppula[6], Ari Laaksonen[7,1] and Annele Virtanen[1]

[1] University of Eastern Finland, Dept. of Applied Physics. P.O. box 1627, FI-70211 Kuopio, Finland
[2] Lund University, Department of Physics, P.O. box 118, SE-22100, Lund, Sweden
[3] Lund University, Ergonomics and Aerosol Technology, P.O. box 118, SE-22100, Lund, Sweden
[4] Lund University, Division of Combustion Engines, P.O. box 118, SE-22100, Lund, Sweden
[5] Tampere University, Aerosol Physics Laboratory, P.O. box 1001, FI-33014, Tampere, Finland
[6] Finnish Meteorological Institute, Atmospheric Research Centre of Eastern Finland, P.O. box 1627, FI-70211 Kuopio, Finland
[7] Finnish Meteorological Institute, P.O. box 503, FI-00101, Helsinki, Finland

*Correspondence to:* Kimmo Korhonen (Kimmo.korhonen@uef.fi)

**Abstract.** We studied ice nucleating abilities of particulate emissions from a modern heavy-duty diesel engine using three different types of fuel. The polydisperse particle emissions were sampled during engine operation and introduced to a continuous flow diffusion chamber (CFDC) instrument at a constant relative humidity $RH_{water} = 110\ \%$, while the temperature was ramped between -43 and -32 ºC (T-scan). The tested fuels were EN 590 compliant low-sulfur fossil diesel, hydrotreated vegetable oil (HVO) and rapeseed methyl ester (RME), all were tested without blending. Sampling was carried out at different stages in the engine exhaust after-treatment system, with and without simulated atmospheric processing using an oxidation flow reactor. In addition to ice nucleation experiments, we used supportive instrumentation to characterize the emitted particles for their physicochemical properties and presented six parameters. We found that the studied emissions contained no significant concentrations of ice nucleating particles likely to be of atmospheric relevance. The substitution of fossil diesel with renewable fuels, using different emission after-treatment systems such as a diesel oxidation catalyst, and photochemical aging of total exhaust had only minor effect on their ice-nucleating abilities.

## 1 Introduction

Atmospheric aerosols affect the energy budget of the Earth and thus climate in different ways: directly through absorption and scattering of heat and light, respectively, and indirectly via affecting cloud formation and lifetime. Direct effects, including e.g. aerosol optical thickness and light scattering and absorption are being routinely monitored through a global network of ground-based instruments, such as sun photometers (Toledano et al., 2012) and lidars (Althausen et al., 2009), respectively. The indirect effects, however, remain less understood (Boucher et al, 2013; Kreidenweis et al., 2018) due to complexity of the

processes within the clouds that contribute to the total effect. One indirect effect of the aerosol particles is related to mixed-phase clouds (MPCs), where certain types of particles may promote formation of ice crystals within them via immersion freezing (Murray et al., 2012). Understanding mixed-phase cloud processes is of utmost importance as cloud lifetimes and their radiative properties depend strongly on the cloud phase (Korolev et al., 2017).

Homogeneous freezing of cloud droplets requires a temperature of approximately -38 ℃ (Pruppacher and Klett, 1997; Lohmann et al., 2016). However, previous studies (viz. Despres et al., 2012; Hoose and Möhler, 2012) have shown that atmospheric aerosol particles, both of natural and anthropogenic origins, can induce ice formation in much higher temperatures than those required for homogeneous ice nucleation. Particles that are ice-active temperatures above -30 ℃ are relatively rare in the lower troposphere with number concentrations on the order of $0.01$ $cm^{-3}$ in many regions (DeMott et al., 2010), but there is evidence that combustion emissions from hydrocarbon fuels can act as ice nuclei at temperatures higher than that. For instance, soot particles from combustion of acetylene (DeMott, 1990), kerosene (Diehl and Mitra, 1998) and a soot generator (Gorbunov et al., 2001) have been reported to be active INPs at -24, -20, and -10 ℃, respectively, in immersion and condensation freezing. In addition to them, Thomson et al. (2018) observed elevated INP concentrations near a ship channel, where usual background INP number concentrations are low; similar ice-nucleating behavior was observed from aircraft emissions by Ikhenazene et al. (2020). On the other hand, there are multiple studies reporting a wide range of soot particles being inefficient ice nucleating particles (INPs) at MPC-relevant temperature and humidity conditions (Adams et al., 2020; Kanji et al., 2020; Schill et al. 2016 and 2020; Vergara-Temprado et al., 2018; Falk et al., 2021). In most of those past immersion freezing ice nucleation studies of soot particles dating back to pre-2010s, little if any information about the aerosol particle properties is provided. Korhonen et al. (2020) reported that the immersion freezing ability of freshly emitted soot particles from biomass combustion depended on minor differences in the combustion conditions, but their analysis on six different physicochemical properties of the studied particles revealed no correlation to their ice-nucleating ability. In addition to that, atmospheric aging processes may change the ice nucleating (IN) abilities of different soot particles (Mahrt et al., 2020a; Mahrt et al., 2020b; Häusler et al., 2018; Zhang et al., 2020). The studies mentioned above demonstrate the challenge in associating soot INP properties to the ambient soot particle population.

The inconsistency in reported observations on soot IN activity implies that it is unclear which types of soot are efficient INPs, and it has been addressed by multiple studies such as Hoose and Möhler (2012), and Bond et al. (2013). Besides, the Intergovernmental Panel of Climate Change (IPCC) have identified gaps in our knowledge of ice nucleation activity of soot in their Fifth Assessment Report (Boucher et al, 2013). The wide range in the reported immersion freezing ability of soot particles is reflected by the available parameterisations spanning several orders of magnitude (Vergara-Temprado et al., 2018). Consequently, the variation leads to high uncertainty in estimation of the radiative forcing via modeling; for instance, the climate forcing due to anthropogenic soot particles immersion freezing has been reported to span from 0.1 W $m^{-2}$ to 1 W $m^{-2}$ , depending on their ice nucleating abilities (Yun et al., 2013).

The diesel engine is the most widely used internal combustion engine type in transportation of goods globally (Davis et al., 2017). The popularity of this engine type increased significantly after the 1930s when outdated coal-fired steam engines became largely replaced by diesels particularly in heavy-duty engines, such as ones used in rail and marine traffic. In road traffic, diesel is the most common engine type in heavy trucks, utility vehicles and nonroad mobile machinery due to its high energy efficiency. Despite the high efficiency, the particulate emissions have been a major drawback of the diesel engine before the increased use of diesel particulate filter (DPF) in road vehicles in developed markets. Klimont et al. (2017) reported a decline of 20-65% in black carbon (BC) emissions from heavy-duty road diesel engines in developed markets between 1990-2010, during the period when advanced emission after-treatment systems, such as the DPF and diesel oxidation catalyst (DOC), became mandatory in new diesel-powered road vehicles. However, Klimont et al. (2017) also reported a much slower reduction in less developed regions, which is due to less stringent emission regulations but likely also due to old vehicles being exported there near the end of their life span in the developed areas. Diesel emissions, especially from old vehicles without emission after-treatment systems such as DPF or DOC, can severely impair air quality in urban areas and thus adversely affect human health. Both the properties of the solid core particles and carcinogenic components adsorbed to the particles may affect the adverse effects (Bové et al., 2019; Bendtsen et al., 2020). These effects of diesel combustion emissions are known; however, their total indirect climate effect, including the potential of diesel emission particles to act as INPs remains less studied.

Particulate emissions from diesel engines can form a notable fraction of total aerosol burden in urban areas, especially in regions where diesel-powered vehicles outnumber ones using different types of fuel (DeWitt et al., 2014). For instance, such regions can comprise arterial roads near seaports, transport hubs, or other facilities whose functions rely on heavy-duty transport. This diesel vehicle exhaust is emitted into the planetary boundary layer (PBL) close to the ground level. From mid to high latitudes, low-level mixed-phase clouds may be present within the PBL (e.g. Gierens et al., 2020) and can potentially get influenced by diesel vehicle emissions. Alternatively, soot particles can be transported to higher altitudes and over long distances (Storelvmo, 2012), so they can potentially influence clouds far from the source region. Soot particles have been identified in ice crystal residues in mixed-phase clouds (Cozic et al., 2008; Ebert et al., 2011), and thus it is of relevance to study their potential to facilitate immersion freezing. Besides, studies focusing on urban impacts on climate and weather show evidence of precipitation anomalies downwind from large cities (Han et al., 2012; Han et al., 2013; Zhong et al., 2015). Hence, understanding the role of urban aerosol particles in cloud processes is essential for further understanding of the origins of the abovementioned anomalies in precipitation due to urban aerosol particles. The number of previous studies focusing on the ice nucleating potential of fossil and renewable diesel fuels (Schill et al., 2016; Chou et al., 2013), however, is limited, to our knowledge. Understanding the potential changes in atmospheric INP budget due to altering anthropogenic emissions, such as ones reported by Klimont et al. (2017) in this context, is essential for a further understanding of processes that may affect the climate change.

Chou et al. (2013) used a continuous flow diffusion chamber (CFDC) to study the immersion freezing ice nucleating potential of particles emitted from two small diesel engines belonging to a passenger car with a DOC and to a transporter without any aftertreatment, respectively. Freshly emitted particles and particles exposed to photochemical aging were studied at temperatures of -35 and -30 ºC. No immersion freezing was observed with one exception of a single measurement at -35 ºC, where α-pinene was added in the aging phase. Schill et al. (2016) used a similar CFDC set-up with a sample temperature of -30 ºC, to study the immersion freezing of freshly emitted and aged particles from a small diesel engine with combustion of diesel and biodiesel, respectively. They observed no concentrations of INPs that differed significantly from the background in any of the experiments. Kulkarni et al. (2016) studied the deposition freezing ice nucleating potential at temperatures from -50 to -40 ºC of fresh and aged diesel engine emissions and they reported heterogeneous freezing in many cases and found that humidification of organic-coated (aged) soot particles increased their ice nucleating abilities. The physicochemical properties of particles emitted from diesel engines depend on (1) the engine, (2) engine operation, (3) the fuel and lubrication oil applied, (4) emission aftertreatment, and (5) atmospheric aging. Hence, it is necessary to study a wide variety of such different parameters, to fully assess the ice nucleating potential of diesel engine emissions.

This study was a part of an extensive measurement campaign where the particulate emissions from a modern commercial heavy-duty diesel engine that was modified for single-cylinder operation were studied with multiple different foci and more information about the characteristics of the studied particles are presented in separate publications. For instance, Gren et al. (2021) studied the properties of particulate and gas-phase emissions of the test fuels, and how different emission after-treatment systems such as DOC and DPF affected them. This study focuses on immersion freezing ice nucleating abilities of the particulate emissions studied in this experiment campaign. While smaller diesel engines may get replaced by more sustainable alternatives in the near future, it is likely that heavy-duty diesel engines will be in use further than that. For the experiments, we used one fossil diesel and two different modern renewable diesel fuels, and two emission aftertreatment systems used both individually and combined; both fresh and photochemically aged particles were studied. Our ice nucleation experiments aimed towards detection of potentially low ice nucleating abilities in immersion freezing mode, down to temperatures where homogeneous freezing starts to dominate. Furthermore, we provide a detailed physicochemical characterization of the studied aerosol particles.

## 2 Experiment methodology

### 2.1 Experiment set-up

The experiments were conducted at the Combustion Engine Laboratory of Lund University. The test engine used was a six-cylinder inline Scania D13 heavy-duty diesel engine that was modified for single-cylinder operation. During the experiments, the engine was in a separate compartment of the laboratory and operated remotely from the control room. The standard

experiment procedure involved studies of fresh and aged emissions for the same engine operation. Our configuration allowed for one full temperature scan with the CFDC for each particle type studied within a single experiment.

Figure 1 shows the sampling strategy for ice nucleation experiments and collection of relevant supportive data. The set-up allowed multiple types of combinations for exhaust emission aftertreatment, secondary organic aerosol (SOA) production and simulated aging. In the experiments, the sampling workflow procedure was as follows: first, the fresh particle emissions entered emission aftertreatment section with three different options: 1) Engine-out (no aftertreatment), 2) DOC (Diesel Oxidation Catalyst), 3) DOC+DPF (Diesel Particulate Filter). From these options, the engine-out represents a situation where the

emission aftertreatment systems either are non-existent or have been disabled in regions where equipping them is not mandatory. Using the DOC reduces engine emissions via completing the oxidation of organic particulate matter (PM), hydrocarbon (HC), and carbon monoxide (CO) components from exhaust gas (Russell and Epling, 2011), while the soot mass remains unaffected (Gren et al., 2021) Modern diesel vehicles are mandatorily equipped with a DPF in developed markets. This DPF is typically located downstream from the DOC, to effectively remove also BC / soot. After passing the aftertreatment

system, the emissions were cooled down and diluted using a porous tube diluter and ejector dilution at the ratio of ~100 times; the purpose of this was to simulate atmospheric dilution and particle formation/transformation when the emission gases exit the exhaust pipe in real-life operation.

Downstream the aftertreatment section, the particles entered the sample treatment section that also enabled three different

modes: 1) no treatment, 2) photochemical aging in potential aerosol mass oxidation flow reactor (PAM) and 3) thermodenuder after the PAM to remove volatile particle components. It should be pointed out that just as upon atmospheric dilution after the tailpipe, a fraction of the primary organic aerosol will evaporate as the aerosol is progressively diluted towards atmospherically relevant concentrations. Furthermore, the OA formed will partition depending on OA loadings: high concentration favors particle phase partitioning. The OA concentration in the PAM chamber ranged from 0.2-25 $\mu g \; m^{-3}$, which are atmospherically

relevant mass concentrations. The PAM reactor used in this study consisted of a 13 L steel chamber containing two Hg lamps with peak intensities at 185 and 254 nm. The UV light generates ozone and hydroxyl radicals (OH) that oxidize the aerosol as it moves through the chamber. The flow rate through the PAM was controlled to 5–7 L $min^{-1}$. The same UV light intensity was used in all experiments. During the experiments involving the PAM, the OH concentrations varied between 3.2 x $10^{11}$ – 1.3 x $10^{12}$ molecules $cm^{-3}$ s, resulting in corresponding atmospheric aging between 2.5-9.9 days, when an average OH

concentration of $1.5 \times 10^6$ molecules $cm^{-3}$ (Mao et al., 2009) was assumed. Extensive SOA formation occurred in the PAM in all engine-out experiments, while less SOA was formed in measurements after the DOC. The thermodenuder (Aerodyne Inc.,) was held at 250 ºC in all experiments where it was used.

Before introduction to the CFDC and supportive instruments, the sample emissions were further diluted by 1:5-10 using dry

compressed air at room temperature and ejector dilution, to ensure desiccation down to < 10 % $RH_{water}$. In addition to ice

nucleation experiments with the CFDC, the following instruments were used for collection of supportive data: an SMPS system (classifier TSI 3082 equipped with aerosol neutralizer TSI 3088 and condensation particle counter – CPC, TSI 3775). The SMPS scanned continuously during experiments, repeating 180 s scans with a sample-to-sheath flows 1 L min$^{-1}$ and 5 L min$^{-1}$, respectively. The mobility diameter sampling range of the SMPS was set between 11 and 500 nm using multiple charge correction in the software. A cloud condensation nuclei counter (CCNC, Droplet Measurement Technologies, CCN-100) was used in parallel with an aerosol particle mass analyzer (APM, Kanomax) to measure the cloud condensation nuclei (CCN) activity ($\kappa_\alpha$) and effective density ($\rho_{eff}$) of the size-selected quasi-monodisperse particles, respectively. For the size selection, a TSI model 3071 electrostatic classifier (DMA$_{CCN}$ in Fig. 1) was used, and different sizes were selected via variation of DMA voltage during sampling. The emission sampling time of 30 minutes for each aftertreatment/sample treatment combination enabled three CCNC and APM scans for four different dry particle mobility diameters of about 60 nm, 100 nm, 200 nm, and 300 nm, respectively. A soot particle aerosol mass spectrometer (SP-AMS, Aerodyne Inc.) was used for chemical analysis of the exhaust particles. An aethalometer (Magee Scientific AE33, sampling rate 1 Hz) measured soot optical properties, equivalent black carbon (eBC) and absorption Angstrom exponent (AAE). A detailed description of emission particle characterization methodology is presented in Sect. 2.3. For the CFDC experiments, the sample aerosol was desiccated further to < 5 % $RH_{water}$, prior to introduction to the CFDC.

The engine test procedure comprised two phases: pre-heating (duration 1-1.5 hours) and then approximately 1.5 hours of emission sampling, during which freshly emitted and aged particles were sampled. The engine was considered warm when the cylinder head temperature reached 85 ⁰C. During each experiment, the engine was operated at constant speed of 1200 rpm and load of 6 bar gross indicated mean effective pressure, with common rail injection pressure being 2000 bar. A low level of exhaust gas recirculation (EGR) setting was used, corresponding to 18% oxygen intake air to the combustion cylinder. The engine was lubricated with synthetic low-ash motor oil (Shell Mysella S3 N40) in all experiments. The fuels included EN 590 compliant ultra-low sulfur fossil diesel (Swedish classification MK1:B0), hydrotreated vegetable oil (HVO), and rapeseed methyl ester (RME). RME is one type in the group of fatty-acid methyl ester (FAME) biodiesel fuels. The motivation for the choice of the test fuels was that substituting fossil diesel by renewable diesel fuels, such as HVO and FAME-type biodiesel, to mitigate $CO_2$ emissions, changes the physicochemical characteristics of the diesel engine exhaust emissions (Dimitriadis et al., 2018; Giakoumis et al., 2012; Gren et al., 2020; Karavalakis et al., 2017; Murtonen et al., 2010). Using the HVO and FAME-type fuels can significantly reduce PM, HC, and CO emissions in diesel exhaust (Giakoumis et al., 2012; McCaffery et al., 2020), and change the particle size distribution and soot nanostructure (Lapuerta et al., 2008; Savic et al., 2016) in comparison to fossil diesel fuel. It is worth emphasizing that all test fuels consisted of single component only and no fuel blends were tested. The engine set-up, after-treatment systems and aging set-up are described in more detail by Gren et al. (2021).

## 2.2 Ice nucleation experiments

The instrument used for the ice nucleation experiments was the SPIN (Spectrometer for Ice Nuclei) which is a commercial ice nuclei (IN) counter manufactured by Droplet Measurement Technologies Inc., Colorado, USA (Garimella et al., 2016). The SPIN is a CFDC with parallel-plate chamber geometry similar to the Portable Ice Nuclei Chamber (PINC) introduced by Chou et al. (2011) and the Zurich Ice Nuclei Chamber (ZINC) described by Stetzer et al. (2008). During standard operation, the aerosol is sampled at a flow of 1 standard liter per minute (SLPM). When the sample flow passes the IN chamber, it is sandwiched between two sheath flows of 4.5 SLPM each, and the residence time inside the chamber is approximately 10 seconds. The freezing conditions are created via generating a diffusional flux of water vapor across the chamber by setting the ice-covered chamber plates to different sub-zero temperatures. The properties such as temperature, relative humidity, and the ideal path of the sample flow, from here on referred to as the aerosol-lamina, are modelled using a 1-D flow model by Rogers (1988). Prior to detection by an optical particle counter (OPC), the aerosol passes through an isothermal evaporation section (residence time approximately 2 seconds) whose temperature was set to follow the average aerosol-lamina temperature. Before starting the experiments, the instrument was prepared following the procedures introduced by Korhonen et al. (2020). The chamber was cooled down to -32 ºC and filled with de-ionized water for 5 seconds to create a thin ice layer on both chamber plates, and after purging the chamber it was vacuumed down to 70 mbar for 5 min to remove loose ice and make the ice layer smoother inside the IN chamber. This procedure reduced the background signal, i.e., unwanted ice crystal counts, significantly and it was typically less than 1 particle per liter in the beginning of experiments. The chamber was re-iced when the background signal exceeded 10-15 particles per liter. This practice, together with averaging the SPIN measurement data to 10-second periods, allowed lower detection limit (smallest detectable ice-activated fraction) in the order of $10^{-6}$ when the sample aerosol number concentrations were $10^3$-$10^4$ cm$^{-3}$, in a similar fashion to one used in transient experiments by Korhonen et al. (2020).

The ice nucleation experiments were done using T-scans (i.e., ramping the sample temperature upwards) at constant $RH_{water} =$ 110%, at temperatures from -43 to -32 ºC. The high relative humidity on water enabled investigation of the IN activity in immersion freezing mode as it is defined by Vali et al. (2015) because liquid formation on the sample particles is expected before possible ice formation takes place; immersion and condensation freezing modes are indistinguishable in the SPIN. Operating the SPIN at $RH_{water} =$ 110% led to co-existence of liquid droplets and potential ice crystals under certain conditions. Although this approach focusing on immersion and condensation freezing only is uncommon for most CFDC studies, the approach and its advantages are discussed in more detail by Korhonen et al. (2020). In the present study, the main motivation for this operation procedure was 1) to ensure that particles with low CCN activity form droplets inside SPIN due to a relatively high supersaturation, and 2) that immersion freezing over a wider temperature range can be investigated within a single run. The co-existence of droplets and ice is due to the evaporation section of the SPIN, which is less efficient than in most other CFDCs due to insufficient residence time of approximately 2 seconds (Garimella et al., 2017). Although it can be expected that the droplets shrink when passing the evaporation section, the residence time is insufficient to evaporate them completely.

The SPIN OPC detects depolarization of the light scattered from the detected particles, which is designed to be used for discrimination of ice crystals from liquid particles in particle-by-particle analysis. However, we found that the depolarization data recorded with the OPC of the SPIN5 was not of a sufficient quality for this analysis. Therefore, we used basic size-separation for ice crystals and droplets and consider particles larger than 4 µm ice crystals. Given the high relative humidity,

110% over water and 150-163% over ice (depending on lamina temperature during the T-scan), the formed ice crystals grow fast inside the SPIN to optical sizes much higher than the 4 µm threshold used, typically to a range between 7-15 µm. We define the ice-activated fraction $\alpha$ as:

$$\alpha = \frac{N_{ice}}{N_{CPC}},$$

(1)

where $N_{ice}$ is the background-corrected number concentration of ice crystals detected by the OPC, from the combined sample and sheath flows exiting the SPIN. The internal background signal of the SPIN was measured by sampling dry, filtered air for at least 5 minutes before initiation of the T-scan and again after the scan had finished. Background correction means subtraction of background counts from the OPC signal during these background checks. The background values between them (i.e., during

the T-scans) were linearly interpolated to indicate the background signal also during sampling. $N_{CPC}$ is the number concentration of sample particles detected by the CPC, in parallel with the SPIN. The random uncertainty from CPC and OPC is estimated via assuming that the respective counts are randomly distributed, and their standard deviation equals to the inverse of the square root of counts (Taylor, 1997). From that, the total random uncertainty was calculated via propagation of uncertainty, considering IN counts, IN background counts and total particle count as variables involved in definition of $\alpha$ (see

Eq. 1). It is worth noting that this error analysis comprises only the statistical random errors; the additional CFDC-related biases, such as particle losses at the laminar flow are discussed in detail by Korhonen et al. (2020). In this study, we define ice onset as an ice-activated fraction of $10^{-3}$ which is a commonly used threshold in the literature.

Table 1 summarizes the experiments on different fuels and aftertreatment combinations. We report a total of 14 experiments

which were conducted with the SPIN. Polydisperse aerosol was sampled in each of them during engine operation. The particle number size distributions all had maxima for sizes well below 100 nm, with a tail towards larger sizes. The sampled particle number concentration was diluted to 2000-20000 cm$^{-3}$ before introduction to the SPIN (marked 'dilution' in Fig. 1), depending on sample treatment; higher number concentrations were used in experiments involving the PAM with extensive formation of new relatively small secondary aerosol particles. It is known that too high sample particle number concentrations may lead to

depletion of water vapor inside a CFDC (Levin et al., 2016), while too low number concentrations reduce the measurement sensitivity. We cannot rule out that the effective water saturation inside the lamina could had been biased slightly low when the highest particle number concentrations were studied (Levin et al., 2016). However, only a fraction of the studied particles was focused within the lamina, and only a fraction of the particles focused in the lamina was likely to activate into cloud

droplets due to small sizes and/or hydrophobicity. Hence, it is unlikely that cloud droplet concentrations inside SPIN exceeded 10000 cm$^{-3}$, for which minor water vapor depletion may occur for slightly different CFDC operation conditions (Levin et al., 2016). The motivation of choosing a relatively high sample concentration was that most of the particles were rather small and/or hydrophobic, so water vapor depletion due to cloud droplet activation inside the CFDC is not expected to be significant, which will be discussed in more detail in Sect. 3.1. For comparison to homogeneous freezing, a homogeneous freezing test of liquid droplets was conducted on quasi-monodisperse 350 nm dry electrical mobility ammonium sulfate (AS) particles that were introduced to the SPIN in a number concentration of approximately 150 cm$^{-3}$. We present the results from ice-nucleation experiments calculated with two methods: first, we present the background-corrected AFs and compare them to the homogeneous freezing reference described above. Additionally, we present the ice nucleation active site (INAS) density model (Schill et al., 2015, 2016; Genareau et al., 2018): the formulation of this approach is included in Appendix A.

### 2.3 Emission particle characterization

To characterize the studied emission particles, we used the supportive instrumentation (see Fig. 1) and inferred the following particle properties. The CCNC was operated in flow scan mode with the same approach as used by Korhonen et al. (2020) and described in more detail by Kristensen et al. (2021). Briefly, quasi-monodisperse particle populations with mobility diameters of about 58, 107, 196 and at times 296 nm, respectively, were selected with the DMA$_{CCN}$ (see Fig. 1). The CCN number concentrations never exceeded 1000 cm$^{-3}$ in these experiments. The particles were introduced into the CCNC and the supersaturation was scanned by up-scans in the flow rate linearly from 0.20 to 1.0 L min$^{-1}$. The total flowrate of the CCNC set-up was maintained at 1.0 L min$^{-1}$ by operating a mass-flow controller (MFC) in parallel with the CCNC with flow rates decreasing from 0.8 to 0.0 L min$^{-1}$ during a CCNC up-scan. The temperature gradient along the CCNC column was kept constant during scans at either 10 or 18 ºC allowing for a full range of supersaturations from about 0.15 to 2.4%. The supersaturations of the CCNC were calibrated with ammonium sulfate particles immediately prior to the campaign, and the calibration procedure is presented in Appendix B. The apparent hygroscopicity, parameter $\kappa_\alpha$, was inferred from the CCNC data following Petters and Kreidenweis (2007). In many cases, the supersaturation did not suffice to activate the studied particles, and results are presented only when a full CCN spectrum could be identified.

The aethalometer measured the attenuation of light at seven different wavelengths between 370 nm to 950 nm, from which the absorption Angstrom exponent (AAE) was inferred. The aethalometer eBC concentration using the standard optical parameters provided by the manufacturer yielded values with a factor of ~2.8±0.6 higher compared to elemental carbon (EC) concentrations (variability < 25% between replicates). Therefore, the eBC concentrations of this study have been scaled to EC by dividing the nominal eBC concentration by an empirical correction factor 2.8, specific to this campaign (described in more detail by Gren et al., 2021). The EC was assessed by sampling of undiluted exhaust on quartz filters (Pallflex Tissuequartz, 47 mm), thermal-optically measured according to EUSAAR_2 protocol (Cavalli et al., 2010). The particle effective density was inferred from the APM data according to the method applied by Rissler et al. (2013) and Korhonen et al. (2020).

For organic quantification of non-refractory aerosols, we used unit mass resolution AMS data from "tungsten only vaporization mode", i.e., the soot module in the SP-AMS was not engaged, with the default relative ionization efficiency of 1.4. The ratio between organic aerosol and equivalent black carbon (OA/eBC) was calculated from AMS (tungsten vaporizer only) and aethalometer data. The SP-AMS data in laser on mode using dual vaporizers provided additional information on the chemical composition of refractory soot particles and qualitative markers on the relative abundance of refractory oxygen (for instance, on particle surface) and presence of partially matured (disordered) carbon nanostructures in the soot (Malmborg et al., 2019). Data analysis was performed using PIKA 1.22A and SQUIRREL 1.62A. The instrument was calibrated with nebulized and DMA selected 300 nm mobility equivalent diameter ammonium nitrate particles, by comparing ionization rates with particle concentrations measured with a CPC.

## 3 Results

### 3.1 Particle size distributions

Figure 2 and Table 1 show that the diesel engine emissions were heavily dominated by ultrafine (< 100 nm) particles by number in all experiments, regardless of fuel, aging or aftertreatment used. The size distributions were commonly bimodal. The geometric mean diameter of nucleation and soot modes were typically around 15 nm and 50 nm in dry electrical mobility, respectively, for fresh (unaged) emissions. Aging with the oxidation flow reactor commonly grew the nucleation mode particles to larger sizes, while use of the diesel oxidation catalyst (DOC) in the exhaust aftertreatment system and use of the thermal denuder in the sampling system reduced the concentration of nucleation mode particles. Gren et al. (2021) present a more comprehensive description of the particle size distributions. Use of the renewable fuels (HVO, RME) led to a reduction of soot mode number emissions of 20% for HVO and 56% for RME, as well as an eBC reduction of approximately 50% was observed for both fuels compared to fossil diesel per MJ (Gren et al., 2021). In this study, the soot mode geometric mean diameters (GMDs) were slightly smaller for HVO (GMD = 40 nm, with geometric standard deviation GSD = 2.0) compared to fossil diesel (GMD = 49 nm, GSD = 2.1) and RME (GMD = 49 nm, GSD = 1.9), which means that the soot reduction of HVO originated from both a reduced soot particle size as well as a reduced number concentration, while the soot reduction of RME were dominated by the reduced number concentration. Thermodenuder treatment (250°C) removed a significant fraction of the OA emissions for both fresh and aged aerosol, as indicated by the differences between the respective particle number size distributions presented in Fig. 2, and further discussed in relation to the presentation of CCN results below.

Particles larger than 100 nm accounted typically for less than 10% of the whole number concentration (see Table 1). Regarding the high sample concentrations introduced to the SPIN, the particles larger than 100 nm represented number concentrations above 50 cm$^{-3}$, thus implying a reasonable detection sensitivity upwards from the order of $10^{-5}$. The fraction of particles larger than 250 nm was near negligible, in the order of $10^{-4}$ from the total sample number concentration. Due to this, we discuss the

ice-nucleating abilities of the studied emissions using two different methods: first, we report the results relative to whole
sample population of polydisperse aerosol. This indicates the overall ice-nucleating ability of diesel emission particles. Second, the INAS density method (Appendix A) aims to provide normalized values that are comparable to ones reported in the literature.

## 3.2 Ice nucleation

We present the results from ascending T-scans on the SPIN and compare them to homogeneous freezing test on AS droplets.
In addition, we present the results of INAS density analysis for sampling temperatures when the AF signal was above the detection limit of the SPIN. Ice-activated fraction curves for fossil diesel emissions are presented in Figs. 3 and 4, the former showing results without photochemical aging and latter with the PAM and PAM+TD. We found that the fresh, unaged emissions from fossil diesel were mostly inactive INPs in both engine-out and DOC experiments: the IN activity appears insignificantly higher than for the homogeneous freezing reference in cases with engine-out and one DOC experiment (red
triangles and blue squares in Fig. 3a, respectively). The repetition of the DOC experiment without sample treatment produced particles that facilitated the ice onset at -38.0 ºC, only 0.8 ºC above the homogeneous freezing reference. Photochemical aging was found to have little effect on the observed ice-activated fractions: even if the cases with photochemical aging equivalent of 9.9 days (blue squares and magenta stars in Fig. 4a) appear to increase the IN activity in comparison to cases with shorter equivalent aging (4.1 days, red and green triangles in Fig. 4a) and the ones without aging, the difference is still within 1.0 ºC
from the homogeneous freezing reference. It may be non-ideal to compare the ice-activated fractions directly between the PAM-processed samples versus the fresh emissions: the particle number size distributions may differ significantly between the samples as illustrated in Fig. 2, so the ice-activated fractions have been normalized to particle number concentrations representing different average particle sizes – and different hygroscopic properties. Nevertheless, we observe no indications of heterogeneous freezing more than 1.0 ºC above the homogeneous freezing in any experiment. Considering the potential
errors and biases, we find it unlikely that the type of photochemical aging simulated in these experiments significantly improve the ice-nucleating ability of the engine emissions studied. However, that does not rule out that other types of atmospheric processing can be of importance for the ice-nucleating ability of diesel engine emissions. The INAS density analysis (Figs. 3b and 4b) shows a steep decrease of approximately three orders of magnitude within 2 ºC above the homogeneous freezing reference temperature, ending up at $10^3$-$10^4$ cm$^{-2}$ even when in close vicinity of temperatures where homogeneous freezing
begins to dominate. This furtherly strengthens the interpretation that fossil diesel emission particles are inefficient INPs in immersion mode freezing at temperatures relevant to MPC conditions.

The HVO produced emission particles that all were ice-active within approximately 0.7 °C above the homogeneous freezing reference and thus, there is no indication of significant heterogeneous ice nucleation outside of the instrument uncertainty
range as Fig. 5a shows. It is also obvious that photochemical aging had no significant effect on the ice activity. When compared to all cases on fossil diesel fuel, the INAS densities were at the same order of magnitude within 2 ºC from the homogeneous

freezing reference, Fig. 5b shows. This suggests that the particulate emissions from the HVO are no more active INPs than the ones from fossil diesel, in both unaged and photooxidated cases.

Finally, we present the observations from the RME emissions in Fig. 6. The most ice-active cases, namely the ones with DOC+TD and PAM+DOC, facilitated ice onset at -38.0 ºC which is 0.8 ºC higher than the homogeneous freezing reference while the untreated emission produced no active INPs. Within 2 ºC above the homogeneous freezing reference, the INAS densities (Fig. 6b) from the RME appeared to be approximately one order of magnitude higher than the ones from fossil diesel and HVO cases, yet it is noteworthy that the steep decrease (three orders of magnitude) within 2 ºC above the homogeneous

freezing reference suggests that also the RME emissions studied are inefficient INPs in the MPC-relevant temperature range in immersion freezing mode.

Figure 7 shows the data from Figs. 3b-6b compared to four parameterizations of soot IN activity in immersion freezing mode. The plotted cases are separated to fresh and an aged ones for fossil diesel fuel due to a larger total number of these cases, but

not for HVO and RME because the photochemical aging was found to have no detectable effect on the IN activity in those cases. It is noteworthy that the available parameterizations span about five orders of magnitude. We find a comparison to the parameterization reported by Schill et al. (2016) most relevant, since they also studied soot particles emitted from a diesel engine with a similar experimental approach. Schill et al. (2016) carried out measurements at -30°C with a CFDC, and they detected no significant INP concentrations above their detection limit. Thus, their parameterization is based on the highest

potential non-observable ice-activity in their experiment representing the detection limit with extrapolations to a wider temperature range. Our inferred ice-activities compare reasonably well with the parameterization reported by Schill et al. (2016) around a temperature of about -35°C. Generally, the INP concentrations were below the detection limit for temperatures >-36°C in our experiments, and judging from the slope of our inferred INAS data, the ice-activity of our studied particles were highly likely well below the Schill-parameterization for temperatures well above -35°C. Hence, it is highly likely that a fraction

of atmospheric diesel soot particles may show ice nucleating abilities significantly below the available parameterizations for temperatures around and higher than -30°C.

**3.3 Emission particle properties**

In addition to ice nucleation experiments, we investigated several physical and chemical properties of the sample particles using the supportive instrumentation, and Table 2 shows the studied properties of the particles in this study. In the engine-out

experiments, very strong SOA formation was obtained upon aging in the PAM. The OA enhancement (processed OA / fresh OA) was of a factor of ~20 for diesel and ~7 for RME. After the DOC the OA enhancement was substantially lower (~2 for both HVO and fossil diesel). The exhaust from fossil diesel combustion after simulated atmospheric aging and without emission aftertreatment was dominated by secondary organic mass (OA/eBC =3.4). All experiments except aged fossil diesel

and aged HVO at engine-out conditions provided eBC-dominated emissions. Substantial SOA production that led to increased

OA/eBC ratios also occurred for RME at engine out and to a lesser extent HVO after the DOC. As shown by Gren et al. (2021), a majority of the formed SOA condensed on the nucleation mode particles and only a minor fraction ended up in the soot mode. This can be related to the engine type and operating conditions that led to relatively low soot emissions, typical for modern engines. The measurements of particle mass of mobility classified particles with the DMA-APM demonstrated a small but significant increase in effective density of soot particles at both 200 nm and 300 nm upon aging (sizes where the soot mode

was completely dominating the size distribution without contribution from the nucleation mode). This is consistent with minor SOA uptake on the aged soot particles. However, it cannot be concluded if the SOA was evenly distributed as a thin coating on the primary particles in the soot aggregates or if it was unevenly distributed for example in the pores between the primary particles.

Detection of significant concentrations of CCN for any of the bypass or PAM+TD experiments, even for the highest supersaturation approaching ~2.4%, was impossible. We conclude that the associated particles were hydrophobic. For the PAM experiments, we observed droplet formation, and it was possible to infer the CCN activity. For the PAM experiments related to different fuels, the 58 nm particles had kappa values of 0.08-0.10, the 107 nm particles had kappa values of 0.04-0.08, while they were about 0.01 for the 196 nm particles. The number concentrations for the 296 nm particles were too low

for the CCN measurements. An increase in the $\kappa_\alpha$ when the aerosol is exposed to photochemical ageing can be due to (1) formation of SOA, (2) oxidation of the soot surface, and (3) collapse of fractal soot aggregates (Tritscher et al., 2011). Gren et al. (2021) reported the effective density to increase significantly for the very same particles after sampling through PAM, which is indicative of significant SOA coatings which appeared to be more pronounced for the 58 and the 107 nm particles. In combination with the dramatic increase in $\kappa_\alpha$ from undetectably low to levels of up to 0.1, formation of SOA is likely to be the

dominant process. The kappa for SOA formed from diesel engine exhaust has been reported in the range 0.09 to 0.14 (Tritscher et al., 2011). Following the approach described by Wittbom et al., (2014) about the $\kappa_\alpha$ of aged diesel soot particles, our observations indicate that the aged 58 nm particles were likely to be dominated by SOA by volume, while the 107 nm particles were likely to be soot particles with a significant SOA coating. The more modest kappa values for the 196 nm are indicative of a relatively lower ratios between the SOA and the primary soot particle volumes.


The abovementioned suggests that the SOA coating has little or no effect on ice-activity on the studied diesel emission particles, as we observed between the PAM experiments on the three studied fuels. The AAEs and presented ion ratios ($C_{11}/C_3$) indicate mature soot particles typical for diesel engine emissions. Notably the ratio $C_3O_2/C_3$ measured in the laser-on mode with the SP-AMS was higher here for each sample compared with those found for solid biomass fuel combustion in Korhonen

et al. (2020), which indicates the diesel soot was more oxidized in this study. This is expected as modern diesel engines achieve low BC emissions by highly efficient late cycle soot removal by oxidation primarily from OH radicals (Malmborg et al. 2017). However, the true surface of the mixed particles relevant for INPs may also be affected by the non-refractory organic aerosol.

The mass spectra of OA in fresh emissions showed similarities with past experiments where it has been linked to lube oil (Worton et al. 2014). The O:C ratio and H:C ratios of OA measured in the conventional laser off AMS mode was X and Y for typical cases of fresh and aged aerosol respectively.

## 4 Summary and conclusions

We measured the ice nucleating abilities of particulate emissions from three different types of diesel fuel using different emission aftertreatment methods and found that fresh emissions were inefficient INPs in immersion and condensation ice nucleation modes at temperatures relevant to MPC conditions ($RH_{water}$ = 110%, -32 °C > $T$ > -43 °C). All fuels were tested without blending, which enabled investigation on fossil and biodiesel fuels separately. The ice nucleation results agree with previous studies on diesel emissions [Chou et al., 2013; Schill et al., 2016], and we conclude that the diesel emission particles studied using our setup were inefficient INPs in the MPC-relevant temperature range. Our results, in conjunction with a lot of previous studies, indicate that a wide range of black carbon and soot particle types are inefficient as immersion freezing INPs. We found that photochemical aging had minor if any effects on the ice-activated particle fractions. The observed differences from unaged emissions, however, were almost negligible and therefore, we conclude that photochemical aging had little or no effect on the IN activity of studied emissions. With these observations altogether, it can be implied that the emission particles from the diesel fuels studied produced INPs active at temperatures only slightly above (<2 °C) that of homogeneous freezing. However, it cannot be excluded that the studied particles may act as INPs at colder temperatures in the cirrus regime. Moreover, all emissions were by number dominated by ultrafine (<100 nm in electrical mobility) particles which clearly had no contribution to ice nucleation. The alternative method, i.e., the INAS densities inferred from the measurements, strengthens our conclusion that the diesel combustion emissions studied play no or little role in ice crystal formation within MPCs. Regarding that all single-component fuels of this study produced results much alike to each other, it can be expected that fuel blends including similar components also produce no or few active INPs active at temperatures significantly above that of homogeneous freezing.

It is worth mentioning that all experiments in this study were conducted under well-controlled laboratory conditions and using the test engine on its optimal operation settings in all experiments. Contrary to that, the emission particle properties may differ in real-life use of diesel engines, depending on factors such as engine model and age, technical condition of the engine, fuel quality and varying load during the operation. Other aging effects, in addition to studied photooxidation, may affect the IN abilities of emission particles. Old diesel vehicles and fuels with e.g. much higher sulfur content, which all are still commonly used in the developing parts of the world may produce different types of PM emissions; this link should be explored before evaluating the total global climate effect.

**Data availability.** Data sets are available from the authors upon request.

**Author contribution.** TBK planned the ice nucleation experiment methodology on the SPIN instrument, which was operated by KK during the campaign. AL, AV, JF, KK, MK and TBK participated in data analysis of ice nucleation experiments and/or interpretation of results. AE, JF, JP, LM, LG, PK, TBK and VBM participated in collection of supportive data and/or interpretation of results. JP, MT, PK, SS and VMB designed the experiment matrix for the engine operating conditions, the aftertreatment systems and the fuels used in this study. MN and SS operated the diesel engine during the campaign and performed its maintenance, with assistance by LM and PK. BS, JP and MT participated as the organizers and supervisors of the experiment campaign. All authors participated in scientific discussions on this study and reviewed/edited the manuscript during its preparation process. KK prepared the manuscript with contributions from all co-authors.

**Competing interests.** The authors declare that they have no conflict of interest.

**Acknowledgements.** Thomas Bjerring Kristensen gratefully acknowledges funding from the Swedish Research Council VR (grant 2017-05016). Joakim Pagels, Louise Gren, Vilhelm B. Malmborg and Axel Eriksson acknowledge funding from the Swedish Research Council FORMAS (grant 2016-00697) and the Swedish Research Council VR (grant 2018-04200). PK acknowledges the Academy of Finland project "EFFi" grant No. 322120. This research has received funding from the Academy of Finland Flagship Programme "ACCC" (Grant numbers 337550 and 337551). We thank Vikram Singh for technical assistance in diesel engine operation.

## Appendix A: The INAS density formulation

Our approach for INAS density formulation includes assumption of spherical particles, and thus the surface area of single particle is defined as:

$$S_p = 4\pi \left(\frac{d_p}{2}\right)^2, \tag{A1}$$

where $d_p$ is the electrical mobility diameter of the particle. Next, the ice-activated fraction α is defined as:

$$\alpha = \frac{N_{ice}}{N_{CPC}}, \tag{A2}$$

where $N_{ice}$ is the background-corrected number concentration of ice crystals observed with the SPIN-OPC, and $N_{CPC}$ the particle number concentration from the parallel CPC. With these definitions, formula for the INAS density follows the one used by Schill et al. (2015, 2016), and Genareau et al. (2018).:

$$n_S = -\left[\frac{\ln(1-\alpha)}{S_a}\right], \tag{A3}$$


where $S_a$ is the average surface area per particle (total surface area divided by total number of particles), inferred from the SMPS scans during the T-scan on the SPIN.

**Appendix B: CCNC calibration procedure**


The calibration of the supersaturation (SS) of the cloud condensation nuclei counter (CCNC) was carried out with ammonium sulfate particles. Ammonium sulfate (Sigma-Aldrich, purity >99.999%,) was dissolved in ultrapure water (resistivity 18.2 MΩcm, total organic carbon <0.04 ppb) and the solution was nebulized. The produced aerosol was dried with diffusion dryers to a relative humidity below 10%. Quasi-monodisperse particle populations were selected with a differential mobility analyzer
(DMA 3082, TSI) with a 1:10 ratio between the sample and the sheath flows. A condensation particle counter (CPC 3772, TSI) and the CCNC were operated in parallel after the DMA. The temperature increase (ΔT) along the CCNC activation column was kept at three different constant levels of 4 K, 10 K and 18 K, respectively, to cover a wide range of supersaturations. The CCNC was for the calibration operated in flow scan mode in the same manner as done for the measurements on engine emissions. The CCNC total flow rate ($Q_t$) increased linearly from 0.20 to 1.0 lpm over the course of 120 s followed by a
constant $Q_t$ =1.0 lpm for 20 s before a decrease in $Q_t$ to 0.20 lpm. The $Q_t$ was maintained at a constant level of 0.20 lpm for 20 s before the cycle was repeated. The CCNC was operated in parallel with a bypass mass flow controller, which ensured that the total flow rate through those two units added up to 1.0 lpm at any given time during operation. Only data obtained during the upscan in $Q_t$ were used for the analysis.

CCN spectra with the droplet activated particle fraction versus $Q_t$ were produced. The contribution of multiply-charged particles to the CCN was identified from the CCN plateau present for $Q_t$ in the range somewhat below when the dominant activation of singly-charged particles were observed in the spectra. Relaxed step functions were fitted to the CCN spectra associated with the singly-charged particles. The critical flow rate ($Q_{50}$) for which 50% of the singly-charged particles activated were inferred from the relaxed step functions. The theoretical supersaturation associated with a given dry ammonium sulfate
particle diameter and a given aerosol temperature was determined from Köhler theory based on input from the online interface of the aerosol inorganic model (AIM) (Wexler and Clegg, 2002). The ammonium sulfate particles were assumed spherical and the bulk density of ammonium sulfate (1.77 gcm$^{-3}$) was applied in the calculations. The activation temperature ($T_{act}$) was assumed to be $T_{act}=(T_1+T_2)/2$, where $T_1$ and $T_2$ were the temperatures at the top and the center of the CCNC activation column, respectively.


The obtained supersaturation calibration curves are presented in Fig. B1 with the associated 95% confidence intervals. The presented calibration curves were obtained immediately prior to the measurements on engine emissions. The obtained calibration curves for $\Delta T$ equal to 4 K and 10 K were identical to similarly obtained calibration curves over the previous 1.5 years for the same CCNC, while the minor differences observed for the $\Delta T$=18 K calibration over the previous 1.5 years were within the random errors. Hence, the CCNC performance under these operation conditions appeared highly reproducible and robust over longer periods of time. In the current study, the main focus was related to whether the particles had CCN activities within the detection range of the CCNC, or whether the particles were too hydrophobic for detection with the CCNC. Hence, we put no effort into random error propagation analysis in that context.

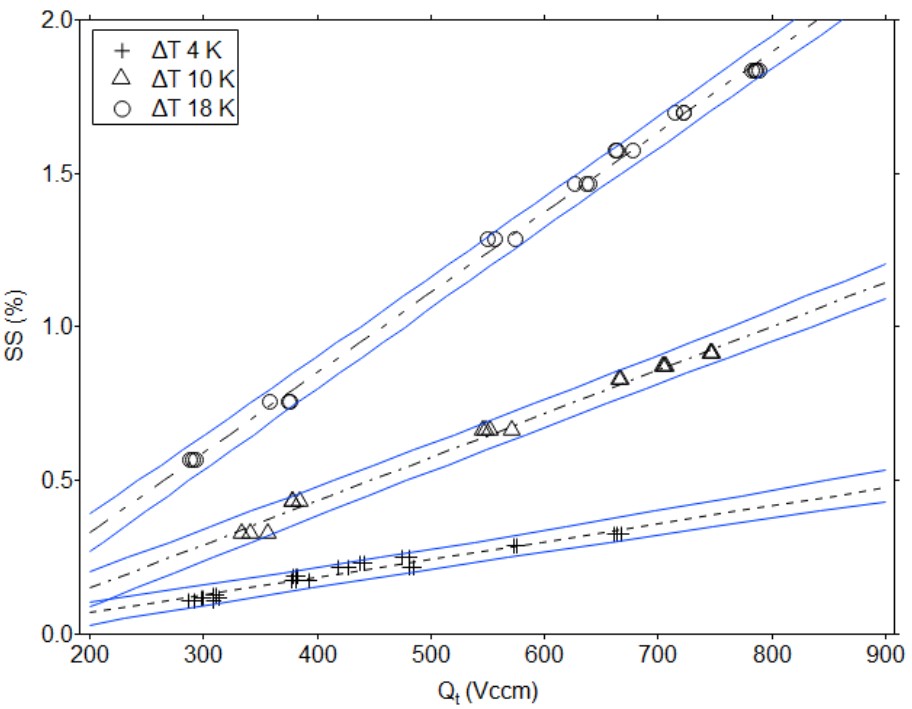

**Figure B1: Calibration curves for the cloud condensation nuclei counter (CCNC) operated in flow scan mode with the supersaturation (SS) versus the flow rate ($Q_t$). Three constant temperature differences ($\Delta T$) were applied along the CCNC column of 4 K, 10 K and 18 K, respectively. The dashed lines represent least squares linear regressions for the three respective $\Delta T$s. The full blue lines indicate the 95% percent confidence interval inferred according to Miller and Miller (2010).**

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

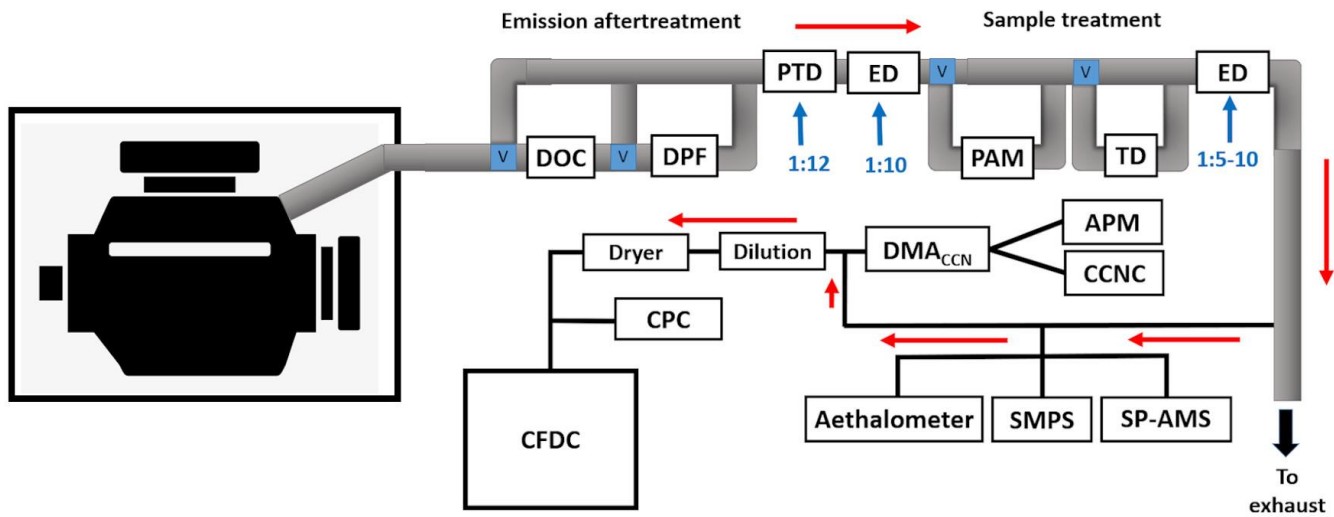


**Figure 1: Schematic of the sampling strategy relevant for the IN experiments. Blue squares with letter "V" indicate valves to change between different sampling modes. The red arrows indicate direction of sample flow. The acronyms are defined as follows: APM = aerosol particle mass analyzer, CCNC = cloud condensation nuclei counter, CFDC = continuous-flow diffusion chamber, CPC = condensation particle counter, DOC = diesel oxidation catalyst, DMA = differential mobility analyzer, DPF = diesel particulate filter,**

**ED = ejector dilution, PAM = potential aerosol mass oxidation flow reactor, PTD = porous tube diluter, SMPS = scanning mobility particle sizer, SP-AMS = soot particle aerosol mass spectrometer and TD = thermodenuder. The dimensions and sample line lengths are not in same dimensions to one another.**

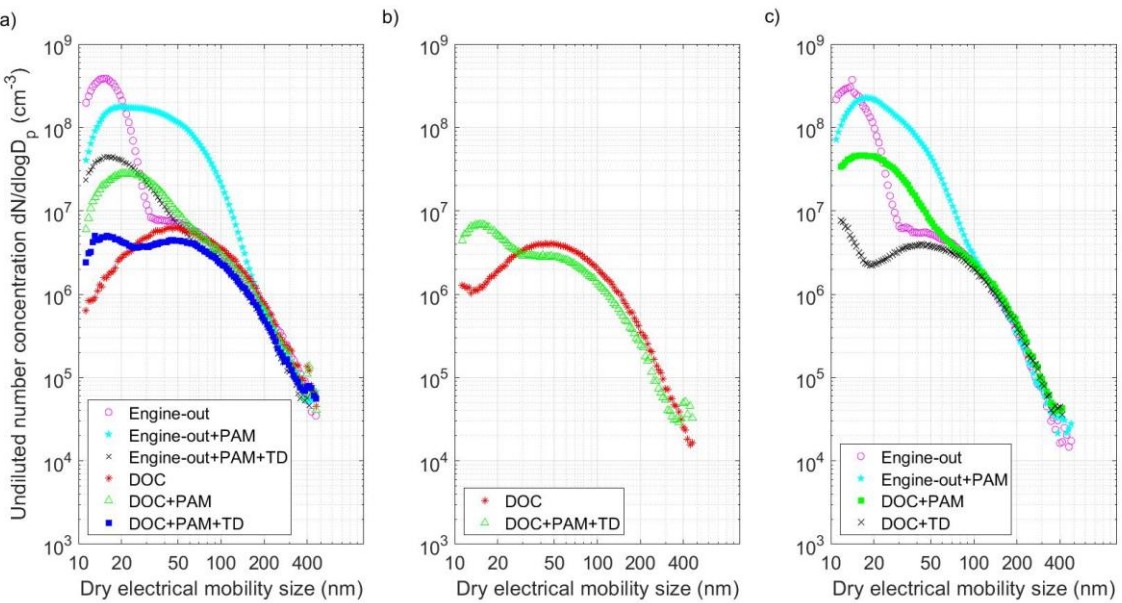

**Figure 2: Number size distributions of sampled emissions, converted to represent an undiluted situation: panels a), b) and c) present the results for fossil diesel, HVO and RME, respectively. The number concentrations represent the average that was measured during the engine operation period in each experiment. "Engine-out" refers to a case without emission aftertreatment. See Gren et al. (2021) for a more detailed analysis on emission particle properties of this study.**

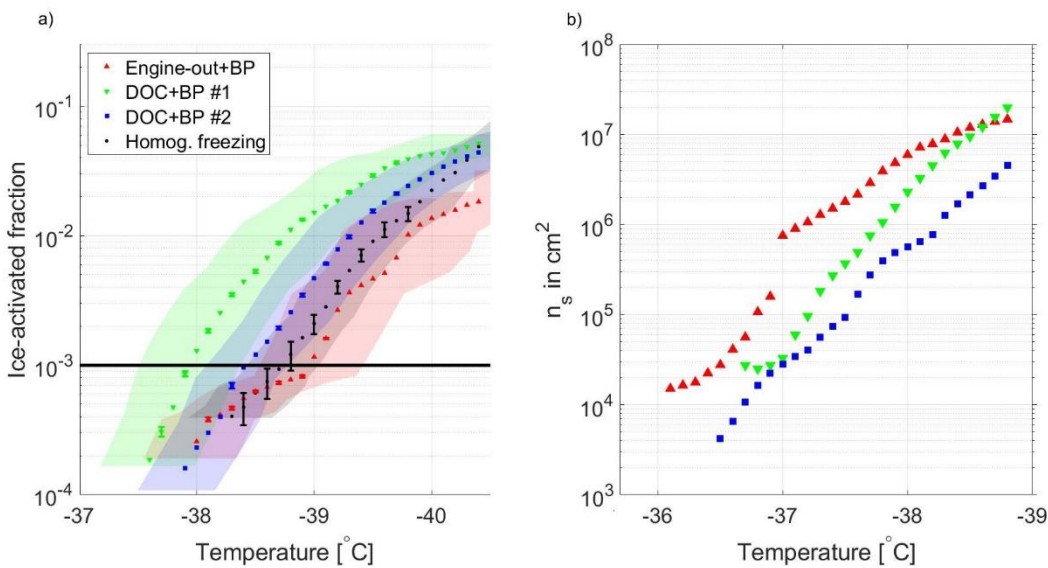

Figure 3: Ice-activation spectra of fossil diesel emissions without photochemical aging at $RH_{water}$ = 110%, compared to homogeneous freezing test of droplets (Panel a), and the corresponding INAS densities (Panel b). In panel a), the error bars represent the random error from detectors (1σ), and the shaded areas of respective colors represent the temperature deviation (1σ) on the lamina during the experiments. The solid black line indicates 0.1% activated fraction. "Engine-out" refers to a case without emission aftertreatment, and BP = bypass (sample treatment method).


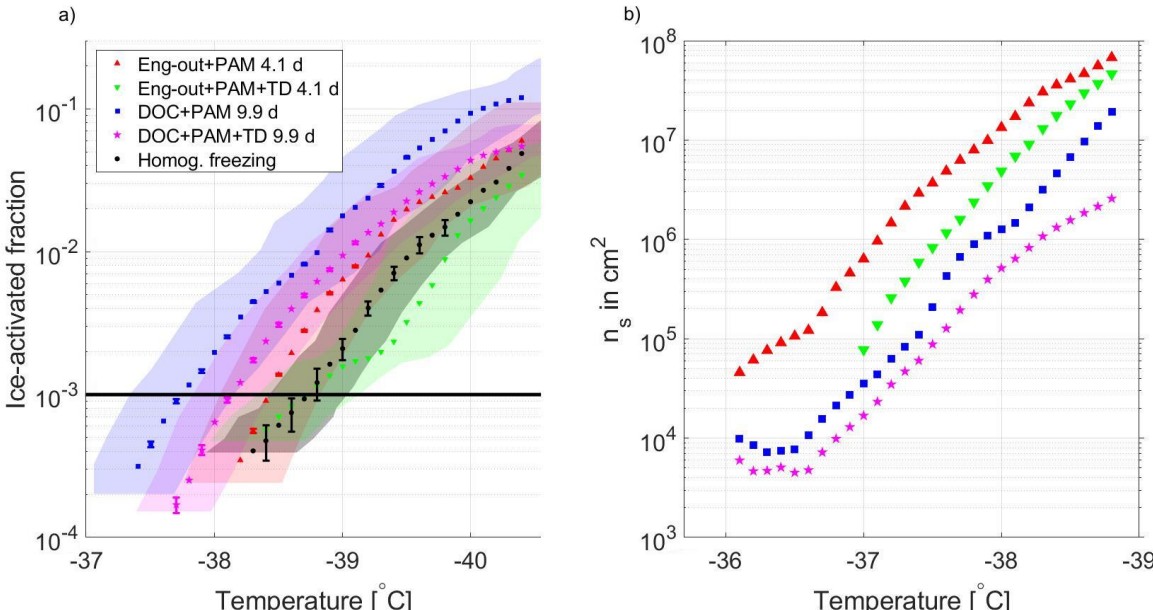

**Figure 4: Ice-activation spectra of fossil diesel emissions with two different photochemical aging levels at $RH_{water}$ = 110%, compared to homogeneous freezing test of droplets (panel a), and the corresponding INAS densities (panel b). In panel a), the error bars represent the random error from detectors (1σ), and the shaded areas of respective colors represent the temperature deviation (1σ) on the lamina during the experiments. The solid black line indicates 0.1% activated fraction. "Eng-out" refers to a case without emission aftertreatment.**


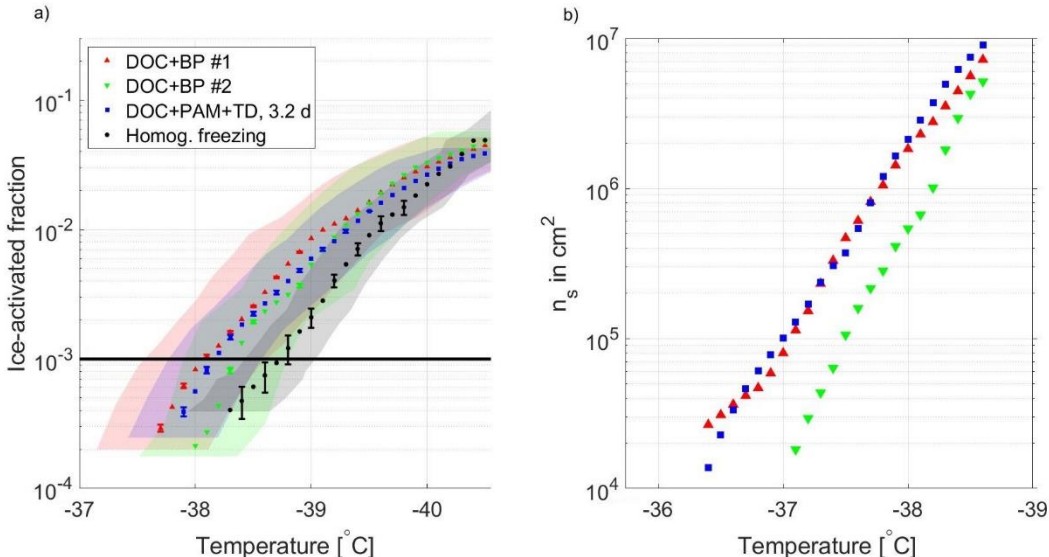

**Figure 5: Ice-activation spectra of HVO emissions at $RH_{water}$ = 110%, compared to homogeneous freezing test of droplets (panel a), and the corresponding INAS densities (panel b). In panel a), the error bars represent the random error from detectors ($1\sigma$), and the shaded areas of respective colors represent the temperature deviation ($1\sigma$) on the lamina during the experiments.**

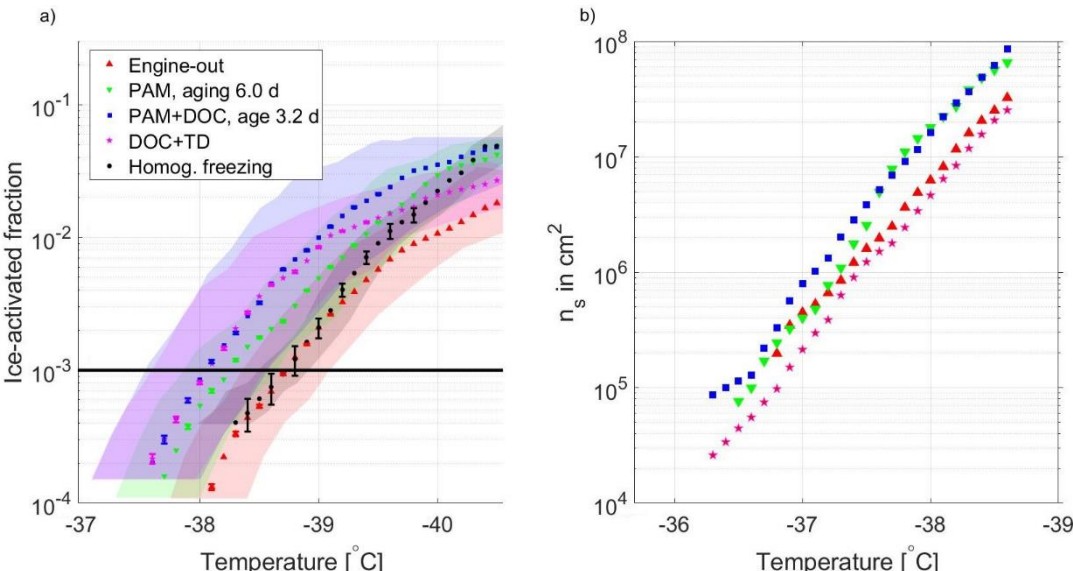

**Figure 6: Ice-activation spectra of RME emissions at $RH_{water}$ = 110%, compared to homogeneous freezing test of droplets (panel a), and the corresponding INAS densities (panel b). In panel a), the error bars represent the random error from detectors (1σ), and the shaded areas of respective colors represent the temperature deviation (1σ) on the lamina during the experiments. The solid black line indicates 0.1% activated fraction. "Engine-out" refers to a case without emission aftertreatment, and BP = bypass (sample treatment method).**


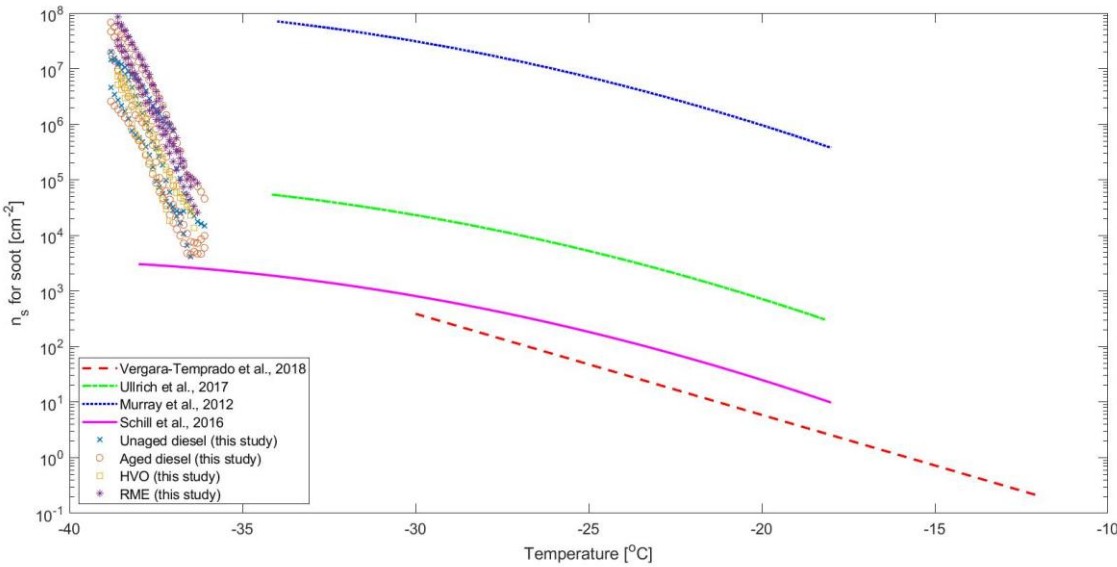

**Figure 7: The INAS densities inferred from the experiments compared to parameterizations of soot IN activities in MPC-relevant temperature range in immersion mode freezing.**

**Table 1: Summary of experiments on different aftertreatment and sample treatment combinations and photochemical aging levels. The PAM aging refers to equivalent atmospheric aging due to photooxidation in days, when average atmospheric OH concentration of $1.5\times10^6$ molecules cm-$^3$ is assumed following Mao et al. (2009).**

| Fuel | Emission aftertreatment | Sample treatment | PAM aging [days] | Percentage of 11-100 nm particles [%] |
|---|---|---|---|---|
| Fossil diesel | engine-out | bypass | - | 98.4 |
| Fossil diesel | engine-out | PAM | 4.1 | 97.8 |
| Fossil diesel | engine-out | PAM+TD | 4.1 | 97.8 |
| Fossil diesel | DOC | bypass | - | 84.4 |
| Fossil diesel | DOC | bypass | - | 84.4 |
| Fossil diesel | DOC | PAM | 9.9 | 96.3 |
| Fossil diesel | DOC | PAM+TD | 9.9 | 89.2 |
| HVO | DOC | bypass | - | 87.7 |
| HVO | DOC | PAM | - | 98.0 |
| HVO | DOC | PAM+TD | 3.2 | 93.5 |
| RME | engine-out | bypass | - | 99.4 |
| RME | engine-out | PAM | 6.0 | 99.6 |
| RME | DOC | PAM | 3.2 | 98.1 |
| RME | DOC | TD | - | 89.8 |

**Table 2: Physical and chemical properties of studied particles: hygroscopicity parameter κα for scanned sample particle sizes, absorption Angstrom exponent (AAE), a marker of surface oxides ($C_3O_2/C_3$), a marker of soot maturity ($C_{11}/C_3$), abundance of organic aerosol (OA: adjusted for dilution by 1:600-900) and OA-to-BC ratio. Empty entries indicate unavailable/unreliable data.**

| Fuel | Emission aftertreatment | Sample treatment | $\kappa_\alpha$ (58nm/107nm/196nm) | AAE | $C_3O_2 / C_3$ | $C_{11} / C_3$ | OA [$\mu g\ m^{-3}$] | OA/eBC |
|---|---|---|---|---|---|---|---|---|
| Fossil diesel | engine-out | bypass | N/A | 1.23 | 0.016 | 0.024 | 118 | 0.15 |
| Fossil diesel | engine-out | PAM | 0.09/0.08/0.01 | 1.24 | - | - | 2339 | 3.4 |
| Fossil diesel | engine-out | PAM+TD | N/A | 1.18 | - | 0.015 | 90 | 0.17 |
| Fossil diesel | DOC | bypass | N/A | 1.17 | 0.018 | 0.023 | 84 | 0.1 |
| Fossil diesel | DOC | bypass | N/A | 1.17 | 0.018 | 0.023 | - | - |
| Fossil diesel | DOC | PAM | 0.08/0.04/0.007 | 1.19 | 0.020 | 0.019 | 153 | 0.24 |
| Fossil diesel | DOC | PAM+TD | N/A | 1.19 | 0.013 | - | 87 | 0.16 |
| HVO | DOC | bypass | N/A | 1.20 | - | - | 46 | 0.12 |
| HVO | DOC | PAM | 0.10/0.04/- | 1.22 | - | 0.014 | 104 | 0.31 |
| HVO | DOC | PAM+TD | N/A | 1.20 | 0.017 | 0.008 | 21 | 0.08 |
| RME | engine-out | bypass | N/A | 1.26 | - | - | 62 | 0.12 |
| RME | engine-out | PAM | 0.09/0.04/0.01 | - | - | 0.032 | 409 | 0.85 |
| RME | DOC | PAM | N/A | - | - | - | 73 | 0.17 |
| RME | DOC | TD | N/A | - | - | - | 22 | 0.06 |
