# Peer review of "Particle emissions from a modern heavy-duty diesel engine as ice nuclei in immersion freezing mode: a laboratory study on fossil and renewable fuels"

_Atmospheric Chemistry and Physics, 2021_

## Referee Comment (RC1)

**Review to "Particle emissions from a modern heavy-duty diesel engine as ice-nuclei in immersion freezing mode: an experimental study on fossil and renewable fuels" by Korhonen et al. ACPD, 2021**

Korhonen et al. present laboratory experiments of the ice nucleation ability of soot particles. Combustion particles are generated in a controlled laboratory setup using a diesel engine, operated with three different fuel types. The ice nucleation ability is tested on with a commercial continuous flow diffusion chamber (SPIN), operated at a fixed relative humidity (*RH*) of 110% and performing *T*-scans over a temperature range between -32 °C to -43 °C. These conditions are relevant for ice formation in tropospheric mixed-phase clouds. The ice nucleation activity is tested for fresh exhaust particles and compared to the ice nucleation activity of exhaust particles that underwent different types of exhaust aftertreatments. Also included are ice nucleation experiments where exhaust particles were first (photochemically) aged in an oxidation flow reactor (PAM chamber) prior to testing the ice nucleation in SPIN. All ice nucleation experiments are performed on polydisperse aerosol populations, with most particles having diameters well below 100 nm. The ice nucleation experiments are supported by a suite of auxiliary measurements to characterize the chemical and physical properties of the exhaust particles. Overall, the authors find the tested soot particles to be poor ice nucleation particles (INPs) in the immersion freezing mode. Photochemical aging in the PAM chamber slightly increased the ice nucleation activity for exhaust particles when burning fossil diesel, but no enhancement was found when burning HVO or RME fuel, although a direct comparison between these measurements is somewhat hampered by using different aging times (and/or combinations of aftertreatments) when operating the engine with these different fuel types.

Overall, I find these results very interesting and within the scope of Atmospheric Chemistry and Physics (ACP). Certainly, the conclusions of this study are largely in-line with previous work and further help to establish the notion that soot particles are inefficient INPs in immersion freezing mode and more generally at temperatures above -38 °C, i.e. above the homogeneous freezing temperature of water. The manuscript is clearly written in most parts (some structural improvements are suggested below), and conclusions drawn are mostly supported by the data shown in the figures. Nonetheless, some issues need further clarifications. Below I list my comments and suggestions that should be addressed upon revising the manuscript. My main concern is related to the "alternative method" that is presented and used to calculate the ice active fraction. In my eyes this point warrants major changes and additional explanations before this paper can be accepted for ACP. In addition, the analysis of measurement uncertainty reported for the AF curves warrants clarification.

**Major comments:**
The authors analyze their ice nucleation measurements in two different ways: In a first ("classical") approach the ice nucleation of the soot particles is presented in terms of the activated fraction (AF), given by the ratio of the ice counts detected by the optical particle counter of SPIN to the particle counts detected by a condensation particle counter operated in parallel to SPIN (see their Eq. 1). This way of data analysis corresponds to the "default" analysis of CFDC data in the ice nucleation community. In an "alternative method", each AF curve is normalized to the maximum ice-active fraction of an AF curve, as the authors note on L217-225. The goal of this alternative method is to estimate the immersion freezing ability of (the largest) particles in the polydisperse aerosol population, which acted as CCN inside SPIN (see L278-281). While the classical approach determines the immersion freezing ability of the entire polydisperse particle population, the alternative approach can be interpreted as a scaled/normalized AF resulting mainly from the larger particles.

For each fuel type tested, the authors report and compare the ice nucleation activity using the classical and the alternative approach (see Figs. 3-6). My interpretation is that this alternative approach, which overall shows a slightly enhanced ice formation signal compared to the homogeneous freezing (ammonium sulfate) reference, aims at teasing out the ice signal resulting from the larger aerosol particles. However, this alternative approach left me somewhat puzzled and the authors will need to significantly revise the text, in order to clarify the added value/benefit of interpreting their ice nucleation data using this alternative approach. Upon revision of the text, I suggest to also combine the paragraphs L215-225 and L273-282, which contain similar/identical information, into one paragraph that should be located within Sect. 2.2.

Related to this issue, questions that should be addressed include:
- The homogeneous reference curve was obtained by testing the ice nucleation activity 350 nm monodisperse ammonium sulfate particles (L212), whereas the combustion particles

are polydisperse aerosols with diameters mainly below 100 nm (Fig. 1). How do the authors justify using their alternative method to compare ice nucleation from such aerosol populations that significantly differ in size? Why not using e.g. the frequently applied ice nucleation active surface site density (INAS)?

- E.g. Fig. 3b: Why does the red line not go all the way up to unity? See also your statement on L223-226. Why is there no uncertainty for this red curve? In Fig. 3a, no data points are depicted for any of the AF curves at temperatures above -38 °C. However, for AF curves depicted in Fig. 3a calculated with the alternative method, data points show up at T > -38 °C. Is this an artefact resulting from extremely low AF (in Fig 3a, presumable below the detection limit of SPIN), showing up in Fig. 3b? Similar comments apply to Figs. 4-6 and to your statement on e.g. L315-317.
- Figs. 3-6: For consistency the y-axis labels should be "α" and "Normalized α" for panels a and b, respectively.

**Minor, specific and technical comments:**
L27: change to "the energy budget of the"
L30 add Kreidenweis et al. (2018)
L32: add „within them via immersion freezing (Murray et al., 2012)."
L33: replace reference by Korolev et al. (2017)
L33: "Furthermore…", this sentence seems a bit unconnected to the topic of immersion freezing and MPC, consider rephrasing.
L36: add Lohmann et al. (2016)
L39: „ice nucleation", here and elsewhere, e.g. L45.
L39: add: "Particles that are ice-active at temperatures above…"
L41: change to: "fuels can act as..:"
L42-44: Should also include Ikhenazene et al. (2020), Thomson et al. (2018)
L45: Remove reference to Hoose and Höhler (2012) and instead add: Schill et al. (2016, 2020), Vergara-Temprado et al. (2018), Adams et al. (2020)
L45: "In most of those…" I suggest to rephrase this sentence as there are a handful of ice nucleation papers on combustion aerosol, which carefully determine the properties of the particles (e.g. Mahrt et al., 2018; Nichman et al., 2019; Zhang et al., 2020).
L50 : Delete reference to Mahrt et al. (2018) and instead consider adding: Mahrt et al. (2020)
L59: Add space after "m$^{-2}$"
L61-8: These paragraphs discuss engine types and aftertreatment, whereas the paragraphs before and after this focus on ice nucleation on combustion aerosol. I suggest restructuring this part upon revision and keep the section relevant to ice nucleation together to improve readability. For instance, the references on L86 should be included/moved to the discussion of ice nucleation on soot (L36-60). As another suggestion, much of the description of the different fuel types (L76-85) could be taken out from the introduction and moved to a subsection of Sect. 2.1.
L68: Please add a brief description here (or where appropriate) what the diesel oxidation catalyst (DOC) does to the exhaust particles in your setup.
L77: consider adding Bove et al. (2019)
L93: What do you mean by "was added in the aging phase"? Were the particles coated with SOA?
L97: What do you mean by "where aging played an important role"? Was the ice nucleation activity enhanced or decreased?
L107: Change to "ability down to temperatures where homogeneous freezing starts to dominate."
L119: Please specify the type of aging in the PAM chamber. On L339 you write that "SOA formation on diesel emission took place in the PAM chamber", but I could not find any detailed information on what this means. Did you coat your exhaust particles? You might want to also refer to the work of Zhang et al. (2020) and relate your results to those presented in this study.
L122 and L126: Please comment on how the dilution steps in your set-up affect the gas-particle partitioning of semi-volatile material associated with the engine exhaust in the manuscript.
L130: Change to: "with sample and sheath flows of 0.3 L min$^{-1}$ and 3 L min$^{-1}$, respectively" (or give as ratio without units).
L138: What physical particle properties are derived from the AMS data?
L157: change to: "at a flow"
L161: change to: "as the aerosol-lamina"
L163: "whose temperature corresponded to the average aerosol-lamina temperature"
L170: change to: "allowed lower detection limit…"
L171: "sample". Do you mean aerosol number concentration here?
L174: change to: "done using T-scans at constant..."

L181: Please quantify aerosol residence time in evaporation section.

L183: replace "in the IN chamber" by "CFDC"

L184: The size threshold to discriminate ice crystals and droplets depends on the RH and T within the CFDC, i.e. the growth conditions of the hydrometeors. Is the indicated threshold valid for all the experimental conditions within your paper? How does this threshold compare to theoretical hydrometeor sizes assuming pure condensational/diffusional growth? See e.g. Rogers and Yau (1989).

L190: Delete "the IN chamber of the"

L192: Change to: "counts measured during… from the OPC signal. The background values between…"

L194-200: This part of the description of the uncertainties and error analysis should be expanded and written more clearly (e.g. by using equations), see also my main comment above. E.g. you might want to at least briefly comment how the "statistical error" (L198) compares to the other uncertainties associated with CFDC measurements.

L203-213: The discussion of the particle number concentrations used within SPIN should be expanded. Please be more quantitative when discussing the results of Levin et al. (2016) and how these values compare to the number concentrations used in your experiments. For instance, what does "high sample particle number concentration" (L207) mean? Considering the upper limit of your combustion aerosol number concentrations (20000 cm$^{-3}$; L205) and those used for ammonium sulfate (150 cm$^{-3}$; L213) and applying a dilution factor of ~10 due to the aerosol-to-sheath flow ratio within SPIN, one still obtains very high number concentrations of around 2000 cm$^{-3}$ for the combustion aerosols and low concentrations for the homogeneous freezing tests. Can such high number concentrations be reliably detected in SPIN? Can you still ensure that there is one INP per ice crystal at these high concentrations? What would happen if you were to use number concentrations of 2000 cm$^{-3}$ for your homogeneous freezing tests?

L230: Please indicate approximate number concentrations for these size-selected CCN measurements. Related to the comment above; can competition of water vapor from high aerosol number concentrations lead to a weak CCN signal? How were multiple-charged particles handled?

L241: Change to: "at seven different wavelengths between 370 nm to 950 nm…"

L243: add space after "<"

L252: do you mean ion number concentrations or ionization rates?

L260: Please specify the type of aging in the OFR. See also my comment above.

L265: Please define "GMD" on L259 and also give GMD for fossil diesel exhaust. In addition, please specify the standard deviations associated with each of the GMD listed.

L268: add space after "250 °C"

L269: How do you know that it is SOA? Combustion exhaust is often also associated with large fractions of hydrocarbon-like organic aerosol (HOA), which can be volatilized. For instance, HOA in engine exhaust is often associated with lubricating oil particles (Canagaratna et al., 2004; Worton et al., 2014); and you note on L149 that your engines was lubricated. Do you have AMS measurement to support your statement?

L272: change to : "SPIN, particles larger than 100 nm represented…"

L272: How does this number relate to the number concentrations listed on L205? See my comment above. I would have expected 10% of 2000 cm$^{-3}$, so the number you state here seems high.

L273: change to: "from the total sample number concentration."

L287: Here and elsewhere (e.g. L291), be consistent on referring to your figures, e.g. use "Fig. 3a".

L289: add space "-41 °C"

L295: "it can be expected that particles with little surface area have passed through the SPIN without any detectable effect." I interpret your statement that you assume the surface area of the exhaust particles to correlate to their ice nucleation activity. Would it then not be more meaningful to use INAS densities, i.e. normalize to surface area instead of maximum AF? Please also see my main comment above. At the same time, I would like to point out that the recent studies by Nichman et al. (2019) and Mahrt et al. (2018) have identified the ice nucleation mechanism on soot particles as pore condensation and freezing (PCF). More recently the studies by Jantsch and Koop (2021) and Marcolli et al. (2020) have developed detailed frameworks how ice nucleation in complex pores of combustion particles can be modelled/predicted. Can such frameworks also be applied to your particles?

L296: Please see my comment to L195: The description of the error analysis should be expanded.

L301: I suggest to state this a bit more careful here. Homogeneous freezing of what? E.g. water freezes homogeneously below -38 °C, where you start seeing a signal. Could this be an indication that you are actually freezing pore water on your exhaust particles homogeneously, i.e. that ice nucleation takes place via PCF, even though homogeneous freezing rates at these temperatures are low? (see my comment to L295). Maybe a better formulation would be: "higher temperatures compared to the ice nucleation curve of ammonium sulfate."

L300-305: Where to the authors expect aging mechanisms for combustion exhaust particles as sampled here and how to these aging times compare to typical tropospheric lifetimes of particles emitted by diesel engines?

L309: change to: "within 0.5 °C of the homogeneous freezing reference"

L339: What SOA formation took place in the PAM? Please see my previous comments.

L343: Write as "$C_3O_2/C_3$"

L339-346: For the discussion of the impact of aging and SOA coating on the ice nucleation activity of combustion aerosol, the authors might want to relate their results to previous work, e.g. Zhang et al. (2020).

L344: To support your statement on the oxidation state here and also your statement on the "extreme hydrophobicity" of the exhaust particles (L337), would it be possible to provide e.g. elemental oxygen-to-carbon and hydrogen-to-carbon ratios of these particles in addition to the fraction of surface oxides listed in Table 2?

L344-346: Is this statement based on the OA/eBC ratios listed in Table 2? If I understand your L253 correctly, the OA values in Table 2 were determine from the AMS measurements. These mass loadings seem extremely high, e.g. 2339 μg m$^{-3}$ for fossil diesel engine-out +PAM. Please comment on the atmospheric relevance of such high mass loadings.

L346-247: The "HVO + engine out" has an OA/eBC ratio of 0.189 according to your Table 2. Do you mean the "RME + engine-out + PAM" instead?

L351: I suggest to explicitly repeat your T and RH conditions here again.

L353: "and we conclude…" this statement should be phrased more carefully and constrained to the experimental conditions studied here, as other studies have demonstrated that combustion particles can form ice at e.g. cirrus temperatures.

L354: delete "complete"

L356: should be phrased more carefully, e.g. "increase the IN activity of particles emitted from fossil diesel combustion…"

L359-360: "With these…"; see my comment to L353.

L361: I suggest to quantify "ultrafine" here in the conclusions again, i.e. give the values in parenthesis.

L362: What do you mean with "to an extent"? Please be more specific in your conclusion section.

L363: replace "production" by "signal"

L365: "slight potential as active INPs"; I suggest to tune this statement down. In the end your observed heterogeneous ice nucleation activity is extremely weak and in the atmosphere such combustion particles will not be able to compete with more efficient INPs such as e.g. mineral dust.

Fig. 2:
> Panel 2a:
> - Comparing the "engine-out + PAM" with the "engine-out + PAM + TD" it appears that your exhaust particles are associated with a large fraction of semi-volatile material. How do you ensure that this material is not list in the additional dilution stage upstream of SPIN (see your Fig. 1)?
> - Comparing the "engine-out" and the "DOC" curves, does the difference in the signal mean that all particles smaller than approximately 40 nm are not soot but volatile material?
>
> Panel 2b:
> - Please add "engine-out curve"
>
> Panel 2c:
> - Why is there an increase at d < 20 nm for the black line, as the red curves in panels a and b suggest that DOC removes most of the particles in this size range?

Fig. 4:
> - Error bars for red and blue curves in panel b are missing; please add.

Fig. 5:
> - Why are only error bars for the green curve shown in panel b?

Table 1
> - Change "Fraction" to "Percentage".

Adams, M. P., Tarn, M. D., Sanchez-Marroquin, A., Porter, G. C. E., O'Sullivan, D., Harrison, A. D., Cui, Z., Vergara-Temprado, J., Carotenuto, F., Holden, M. A., Daily, M. I., Whale, T. F., Sikora, S. N. F., Burke, I. T., Shim, J.-U., McQuaid, J. B., and Murray, B. J.: A Major Combustion Aerosol Event Had a Negligible Impact on the Atmospheric Ice-Nucleating Particle Population, 125, e2020JD032938, https://doi.org/10.1029/2020JD032938, 2020.

Bové, H., Bongaerts, E., Slenders, E., Bijnens, E. M., Saenen, N. D., Gyselaers, W., Eyken, P. V., Plusquin, M., Roeffaers, M. B. J., Ameloot, M., and Nawrot, T. S.: Ambient black carbon particles reach the fetal side of human placenta, Nat Commun, 10, 1–7, https://doi.org/10.1038/s41467-019-11654-3, 2019.

Canagaratna, M. R., Jayne, J. T., Ghertner, D. A., Herndon, S., Shi, Q., Jimenez, J. L., Silva, P. J., Williams, P., Lanni, T., Drewnick, F., Demerjian, K. L., Kolb, C. E., and Worsnop, D. R.: Chase Studies of Particulate Emissions from in-use New York City Vehicles, 38, 555–573, https://doi.org/10.1080/02786820490465504, 2004.

Ikhenazene, R., Pirim, C., Noble, J. A., Irimiea, C., Carpentier, Y., Ortega, I. K., Ouf, F.-X., Focsa, C., and Chazallon, B.: Ice Nucleation Activities of Carbon-Bearing Materials in Deposition Mode: From Graphite to Airplane Soot Surrogates, J. Phys. Chem. C, 124, 489–503, https://doi.org/10.1021/acs.jpcc.9b08715, 2020.

Jantsch, E. and Koop, T.: Cloud Activation via Formation of Water and Ice on Various Types of Porous Aerosol Particles, ACS Earth Space Chem., https://doi.org/10.1021/acsearthspacechem.0c00330, 2021.

Korolev, A., McFarquhar, G., Field, P. R., Franklin, C., Lawson, P., Wang, Z., Williams, E., Abel, S. J., Axisa, D., Borrmann, S., Crosier, J., Fugal, J., Krämer, M., Lohmann, U., Schlenczek, O., Schnaiter, M., and Wendisch, M.: Mixed-Phase Clouds: Progress and Challenges, 58, 5.1-5.50, https://doi.org/10.1175/amsmonographs-d-17-0001.1, 2017.

Kreidenweis, S. M., Petters, M., and Lohmann, U.: 100 Years of Progress in Cloud Physics, Aerosols, and Aerosol Chemistry Research, 59, 11.1-11.72, https://doi.org/10.1175/amsmonographs-d-18-0024.1, 2018.

Lohmann, U., Lüönd, F., and Mahrt, F.: An Introduction to Clouds: From the Microscale to Climate, 1st edition., Cambridge University Press, Cambridge, https://doi.org/10.1017/CBO9781139087513, 2016.

Mahrt, F., Marcolli, C., David, R. O., Grönquist, P., Barthazy Meier, E. J., Lohmann, U., and Kanji, Z. A.: Ice nucleation abilities of soot particles determined with the Horizontal Ice Nucleation Chamber, Atmos. Chem. Phys., 18, 13363–13392, https://doi.org/10.5194/acp-18-13363-2018, 2018.

Mahrt, F., Alpert, P. A., Dou, J., Grönquist, P., Arroyo, P. C., Ammann, M., Lohmann, U., and Kanji, Z. A.: Aging induced changes in ice nucleation activity of combustion aerosol as determined by near edge X-ray absorption fine structure (NEXAFS) spectroscopy, Environ. Sci.: Processes Impacts, https://doi.org/10.1039/C9EM00525K, 2020.

Marcolli, C., Mahrt, F., and Kärcher, B.: Soot-PCF: Pore condensation and freezing framework for soot aggregates, 1–68, https://doi.org/10.5194/acp-2020-1134, 2020.

Murray, B. J., O'Sullivan, D., Atkinson, J. D., and Webb, M. E.: Ice nucleation by particles immersed in supercooled cloud droplets, 41, 6519–6554, https://doi.org/10.1039/C2CS35200A, 2012.

Nichman, L., Wolf, M., Davidovits, P., Onasch, T. B., Zhang, Y., Worsnop, D. R., Bhandari, J., Mazzoleni, C., and Cziczo, D. J.: Laboratory study of the heterogeneous ice nucleation on black-carbon-containing aerosol, 19, 12175–12194, https://doi.org/10.5194/acp-19-12175-2019, 2019.

Rogers, R. R. and Yau, M. K.: A short course in cloud physics, Pergamon Press, Oxford; New York, 1989.

Schill, G. P., Jathar, S. H., Kodros, J. K., Levin, E. J. T., Galang, A. M., Friedman, B., Link, M. F., Farmer, D. K., Pierce, J. R., Kreidenweis, S. M., and DeMott, P. J.: Ice-nucleating particle emissions from photochemically aged diesel and biodiesel exhaust, Geophys Res Lett, 43, 5524–5531, https://doi.org/10.1002/2016gl069529, 2016.

Schill, G. P., DeMott, P. J., Emerson, E. W., Rauker, A. M. C., Kodros, J. K., Suski, K. J., Hill, T. C. J., Levin, E. J. T., Pierce, J. R., Farmer, D. K., and Kreidenweis, S. M.: The contribution of black carbon to global ice nucleating particle concentrations relevant to mixed-phase clouds, PNAS, https://doi.org/10.1073/pnas.2001674117, 2020.

Thomson, E. S., Weber, D., Bingemer, H. G., Tuomi, J., Ebert, M., and Pettersson, J. B. C.: Intensification of ice nucleation observed in ocean ship emissions, 8, 1111, https://doi.org/10.1038/s41598-018-19297-y, 2018.

Vergara-Temprado, J., Holden, M. A., Orton, T. R., O'Sullivan, D., Umo, N. S., Browse, J., Reddington, C., Baeza-Romero, M. T., Jones, J. M., Lea-Langton, A., Williams, A., Carslaw, K. S., and Murray, B. J.: Is Black Carbon an Unimportant Ice-Nucleating Particle in Mixed-Phase Clouds?, 123, 4273–4283, https://doi.org/10.1002/2017JD027831, 2018.

Worton, D. R., Isaacman, G., Gentner, D. R., Dallmann, T. R., Chan, A. W. H., Ruehl, C., Kirchstetter, T. W., Wilson, K. R., Harley, R. A., and Goldstein, A. H.: Lubricating Oil Dominates Primary Organic Aerosol Emissions from Motor Vehicles, Environ. Sci. Technol., 48, 3698–3706, https://doi.org/10.1021/es405375j, 2014.

Zhang, C., Zhang, Y., Wolf, M. J., Nichman, L., Shen, C., Onasch, T. B., Chen, L., and Cziczo, D. J.: The effects of morphology, mobility size, and secondary organic aerosol (SOA) material coating on the ice nucleation activity of black carbon in the cirrus regime, 20, 13957–13984, https://doi.org/10.5194/acp-20-13957-2020, 2020.

---

## Author Comment (AC1)

We thank Referee #2 for his or her comments and constructive suggestions on how to improve the contents of the manuscript. The comments as posted are listed below on green font, our responses to them on red font and the specified modifications to the text on blue font.

Review to "Particle emissions from a modern heavy-duty diesel engine as ice-nuclei in immersion freezing mode: an experimental study on fossil and renewable fuels" by Korhonen et al. ACPD, 2021

Korhonen et al. present laboratory experiments of the ice nucleation ability of soot particles. Combustion particles are generated in a controlled laboratory setup using a diesel engine, operated with three different fuel types. The ice nucleation ability is tested on with a commercial continuous flow diffusion chamber (SPIN), operated at a fixed relative humidity (RH) of 110% and performing T-scans over a temperature range between -32 °C to -43 °C. These conditions are relevant for ice formation in tropospheric mixed-phase clouds. The ice nucleation activity is tested for fresh exhaust particles and compared to the ice nucleation activity of exhaust particles that underwent different types of exhaust aftertreatments. Also included are ice nucleation experiments where exhaust particles were first (photochemically) aged in an oxidation flow reactor (PAM chamber) prior to testing the ice nucleation in SPIN. All ice nucleation experiments are performed on polydisperse aerosol populations, with most particles having diameters well below 100 nm. The ice nucleation experiments are supported by a suite of auxiliary measurements to characterize the chemical and physical properties of the exhaust particles.

Overall, the authors find the tested soot particles to be poor ice nucleation particles (INPs) in the immersion freezing mode. Photochemical aging in the PAM chamber slightly increased the ice nucleation activity for exhaust particles when burning fossil diesel, but no enhancement was found when burning HVO or RME fuel, although a direct comparison between these measurements is somewhat hampered by using different aging times (and/or combinations of aftertreatments) when operating the engine with these different fuel types.

Overall, I find these results very interesting and within the scope of Atmospheric Chemistry and Physics (ACP). Certainly, the conclusions of this study are largely in-line with previous work and further help to establish the notion that soot particles are inefficient INPs in immersion freezing mode and more generally at temperatures above -38 °C, i.e. above the homogeneous freezing temperature of water. The manuscript is clearly written in most parts (some structural improvements are suggested below), and conclusions drawn are mostly supported by the data shown in the figures. Nonetheless, some issues need further clarifications. Below I list my comments and suggestions that should be addressed upon revising the manuscript. My main concern is related to the "alternative method" that is presented and used to calculate the ice active fraction. In my eyes this point warrants major changes and additional explanations before this paper can be accepted for ACP. In addition, the analysis of measurement uncertainty reported for the AF curves warrants clarification.

Major comments:

The authors analyze their ice nucleation measurements in two different ways: In a first ("classical") approach the ice nucleation of the soot particles is presented in terms of the activated fraction (AF), given by the ratio of the ice counts detected by the optical particle counter of SPIN to the particle counts detected by a condensation particle counter operated in parallel to SPIN (see their Eq. 1).

This way of data analysis corresponds to the "default" analysis of CFDC data in the ice nucleation community. In an "alternative method", each AF curve is normalized to the maximum ice-active fraction of an AF curve, as the authors note on L217-225. The goal of this alternative method is to estimate the immersion freezing ability of (the largest) particles in the polydisperse aerosol population, which acted as CCN inside SPIN (see L278-281). While the classical approach determines the immersion freezing ability of the entire polydisperse particle population, the alternative approach can be interpreted as a scaled/normalized AF resulting mainly from the larger particles.

For each fuel type tested, the authors report and compare the ice nucleation activity using the classical and the alternative approach (see Figs. 3-6). My interpretation is that this alternative approach, which overall shows a slightly enhanced ice formation signal compared to the homogeneous freezing (ammonium sulfate) reference, aims at teasing out the ice signal resulting from the larger aerosol particles. However, this alternative approach left me somewhat puzzled and the authors will need to significantly revise the text, in order to clarify the added value/benefit of interpreting their ice nucleation data using this alternative approach. Upon revision of the text, I suggest to also combine the paragraphs L215-225 and L273-282, which contain similar/identical information, into one paragraph that should be located within Sect. 2.2.

Related to this issue, questions that should be addressed include:

- The homogeneous reference curve was obtained by testing the ice nucleation activity 350 nm monodisperse ammonium sulfate particles (L212), whereas the combustion particles are polydisperse aerosols with diameters mainly below 100 nm (Fig. 1). How do the authors justify using their alternative method to compare ice nucleation from such aerosol populations that significantly differ in size? Why not using e.g. the frequently applied ice nucleation active surface site density (INAS)? - E.g. Fig. 3b: Why does the red line not go all the way up to unity? See also your statement on L223-226. Why is there no uncertainty for this red curve? In Fig. 3a, no data points are depicted for any of the AF curves at temperatures above -38 °C. However, for AF curves depicted in Fig. 3a calculated with the alternative method, data points show up at T > -38 °C. Is this an artefact resulting from extremely low AF (in Fig 3a, presumable below the detection limit of SPIN), showing up in Fig. 3b? Similar comments apply to Figs. 4-6 and to your statement on e.g. L315-317.

We have re-analyzed all SPIN data, and we have identified an 'artifact' in the data processing. Whenever the SPIN OPC counts per second exceed a certain threshold, then the SPIN data is stored in a slightly different manner, which is very simple to correct for. That correction was unfortunately not carried out for the ice-activated fractions (AFs) presented in figures 3 to 6 in the previous version of the manuscript. The correct AFs may be up to about 1 order of magnitude higher than what was previously presented for the very lowest temperatures around -41°C, while the differences in the AFs were relatively smaller or insignificant for temperatures above -39°C. Generally, these corrections do not affect the main findings of little to no indications of heterogeneous immersion freezing. Previously, we interpreted the pronounced low AF levels for several diesel samples near T=-41°C relative to the homogeneous freezing reference to be due to small and/or very hydrophobic diesel particles not activating into cloud droplets with the potential for subsequent immersion freezing inside SPIN. For that reason, we found that there was a risk of misinterpreting low AFs as a result of low immersion freezing ice-nucleating ability – rather than an effect of very low CCN activity. That was the motivation for the AF scaling carried out in the previous version. With the

correctly analysed AFs, there are still indications of low CCN activity influencing the AFs but to a significantly less pronounced level. We still think that it is of relevance to consider how very high concentrations of nucleation mode particles and/or very hydrophobic seed particles may bias the immersion freezing AF low by contributing to the total particle concentration while not acting as CCN/INPs. However, we no longer consider it a risk that the 'raw' AFs can be misinterpreted in this context, so we have decided not to carry out the AF scaling and the figures 3.b to 6.b do not appear in the revised version of the manuscript. As a consequence of the revisions described above, the paragraphs in L215-225 and L273-282 have been deleted, and the merging of these two paragraphs is no longer of relevance.

For the comment concerning usage of INAS density normalization: we agree on this and have added the suggested calculation method (formulation and results) to Supplement S1. Moreover, the figures mentioned in the comments have been revised.

- Figs. 3-6: For consistency the y-axis labels should be "α" and "Normalized α" for panels a and b, respectively.

We have omitted the "Normalized α" from the revised manuscript, but we have kept 'Ice-activated fraction' as y-axis labels in the revised figures. Figures should appear as readable as possible, so in this case we do not see a need to introduce an abbreviation in labels.

**Minor, specific and technical comments:**

L27: change to "the energy budget of the"

Following the recommendations by all referees, the text has been changed from:

"Atmospheric aerosols affect the radiative forcing budget of the Earth and thus climate in multiple different ways, directly through absorption/scattering of radiation and indirectly through impacts on cloud properties."

to:

"Atmospheric aerosols affect the energy budget of the Earth and thus climate in different ways: directly through absorption and scattering of heat and light, respectively, and indirectly via affecting cloud formation and lifetime."

L30 add Kreidenweis et al. (2018)

Done.

L32: add „within them via immersion freezing (Murray et al., 2012)."

Done.

L33: replace reference by Korolev et al. (2017)

Done.

L33: "Furthermore…", this sentence seems a bit unconnected to the topic of immersion freezing and MPC, consider rephrasing.

We have deleted the statement and the reference from the revised text.

L36: add Lohmann et al. (2016)

Done.

L39: „ice nucleation", here and elsewhere, e.g. L45.

Corrected throughout the text.

L39: add: "Particles that are ice-active at temperatures above…"

Done.

L41: change to: "fuels can act as..:"

Done.

L42-44: Should also include Ikhenazene et al. (2020), Thomson et al. (2018)

References added.

L45: Remove reference to Hoose and Höhler (2012) and instead add: Schill et al. (2016, 2020), Vergara-Temprado et al. (2018), Adams et al. (2020)

All the suggested studies were on deposition freezing. Similar studies associated with immersion freezing are scarce.

The statement has been changed from:

"In most of those past ice-nucleation studies of soot particles, little if any information about the aerosol particle properties is provided."

to:

"In most of those past immersion freezing ice nucleation studies of soot particles dating back to pre-2010s, little if any information about the aerosol particle properties is provided."

L45: "In most of those…" I suggest to rephrase this sentence as there are a handful of ice nucleation papers on combustion aerosol, which carefully determine the properties of the particles (e.g. Mahrt et al., 2018; Nichman et al., 2019; Zhang et al., 2020).

The statement has been changed from:

"In most of those past ice-nucleation studies of soot particles, little if any information about the aerosol particle properties is provided."

to:

"In most of those past immersion freezing ice nucleation studies of soot particles dating back to pre-2010s, little if any information about the aerosol particle properties is provided."

L50: Delete reference to Mahrt et al. (2018) and instead consider adding: Mahrt et al. (2020)

Done and reference added. Following the suggestion by Referee #2, also added another study by Mahrt et al. (2020).

L59: Add space after "m-2"

Done.

L61-8: These paragraphs discuss engine types and aftertreatment, whereas the paragraphs before and after this focus on ice nucleation on combustion aerosol. I suggest restructuring this part upon revision and keep the section relevant to ice nucleation together to improve readability. For instance, the references on L86 should be included/moved to the discussion of ice nucleation on soot (L36-60). As another suggestion, much of the description of the different fuel types (L76-85) could be taken out from the introduction and moved to a subsection of Sect. 2.1.

We agree with the suggestion and have moved the information about fuels to Sect 2.1. In addition to that, we have made the paragraph more concise and put the focus on 1) the atmospheric relevance of diesel emissions and 2) the motivation for ice nucleation studies, such as this one.

L68: Please add a brief description here (or where appropriate) what the diesel oxidation catalyst (DOC) does to the exhaust particles in your setup.

We agree about this and have added the following statement in Sect. 2.1 where the aftertreatment system combinations are described:

"Using the DOC reduces engine emissions via completing the oxidation of particulate matter (PM), hydrocarbon (HC), and carbon monoxide (CO) components from exhaust gas (Russell and Epling, 2011)."

L77: consider adding Bove et al. (2019)

Reference added.

L93: What do you mean by "was added in the aging phase"? Were the particles coated with SOA?

Yes. Please see our response to your comment on L269 where we discuss the OA formation due to photochemical aging of the particles.

L97: What do you mean by "where aging played an important role"? Was the ice nucleation activity enhanced or decreased?

The text has been changed from:

"Kulkarni et al. (2016) studied the deposition freezing ice nucleating potential at temperatures from -50 to -40 $^{\circ}$C of fresh and aged diesel engine emissions and they reported heterogeneous freezing in many cases, where particle aging played an important role."

to:

"Kulkarni et al. (2016) studied the deposition freezing ice nucleating potential at temperatures from -50 to -40 $^{\circ}$C of fresh and aged diesel engine emissions and they reported heterogeneous freezing in many cases, and found that humidification of organic-coated (aged) soot particles increased their ice nucleating abilities."

L107: Change to "ability down to temperatures where homogeneous freezing starts to dominate."

Done.

L119: Please specify the type of aging in the PAM chamber. On L339 you write that "SOA formation on diesel emission took place in the PAM chamber", but I could not find any detailed information on what this means. Did you coat your exhaust particles? You might want to also refer to the work of Zhang et al. (2020) and relate your results to those presented in this study.

Please see our response to the comment below.

L122 and L126: Please comment on how the dilution steps in your set-up affect the gas-particle partitioning of semi-volatile material associated with the engine exhaust in the manuscript.

The following details about the aging and simulated atmospheric dilution phase in this study have been added to the text:

"The aerosol was diluted ~100 times upstream the PAM and the TD. It should be pointed out that just as upon atmospheric dilution after the tailpipe, a fraction of the primary organic aerosol will evaporate as the aerosol is progressively diluted towards atmospherically relevant concentrations. Furthermore, the OA formed will partition depending on OA ladings: high concentration favors particle phase partitioning. The OA concentration in the PAM chamber ranged from 0.2-25 ug/m3, atmospherically relevant concentrations. The PAM reactor used in this study consisted of a 13 L steel chamber containing two Hg lamps with peak intensities at 185 and 254 nm. The UV light generates ozone and hydroxyl radicals (OH) that oxidize the aerosol as it moves through the chamber. The flow rate through the PAM was controlled to 5–7 L min-1. The same UV light intensity was used in all experiments. During the experiments involving the PAM, the OH concentrations varied between $3.2 \times 10^{11} – 1.3 \times 10^{12}$ molecules cm-3 s, resulting in corresponding atmospheric aging between 2.5-9.9 days, when an average OH concentration of $1.5 \times 10^{6}$ molecules cm-3 (Mao et al., 2009) was assumed. Extensive SOA formation occurred in the PAM in all engine-out experiments, while less SOA was formed in measurements after the DOC. The thermodenuder (Aerodyne Inc.,) was held at 250 °C in all experiments where it was used."

A discussion about the coating of particles has been added in the manuscript. Please see our response to comment on L339 below.

The following text has been added:

"It should be pointed out that just as upon atmospheric dilution after the tailpipe, a fraction of the primary organic aerosol will evaporate as the aerosol is progressively diluted towards atmospherically relevant concentrations. Furthermore, the OA formed will partition depending on OA loadings: high concentration favors particle phase partitioning. The OA concentration in the PAM chamber ranged from 0.2-25 µg m-3, which are atmospherically relevant mass concentrations"

L130: Change to: "with sample and sheath flows of 0.3 L min-1 and 3 L min-1, respectively" (or give as ratio without units).

Done.

L138: What physical particle properties are derived from the AMS data?

The text has been changed from:

"A soot particle aerosol mass spectrometer (SP-AMS, Aerodyne inc.) was used for physicochemical analysis of the exhaust particles."

to:

"A soot particle aerosol mass spectrometer (SP-AMS, Aerodyne Inc.) was used for chemical analysis of the exhaust particles."

L157: change to: "at a flow"

Done.

L161: change to: "as the aerosol-lamina"

Done.

L163: "whose temperature corresponded to the average aerosol-lamina temperature"

Done.

L170: change to: "allowed lower detection limit…"

Done.

L171: "sample". Do you mean aerosol number concentration here?

Yes, we do. We have changed the term to 'sample aerosol number concentrations' in the revised text.

L174: change to: "done using T-scans at constant..."

Done.

L181: Please quantify aerosol residence time in evaporation section.

Done. The text has been changed from:

"The co-existence of droplets and ice is due to the evaporation section of the SPIN, which is less efficient than for most other CFDCs (Garimella et al., 2017)."

to:

"The co-existence of droplets and ice is due to the evaporation section of the SPIN, which is less efficient than in most other CFDCs due to insufficient residence time of approximately 2 seconds (Garimella et al., 2017)."

L183: replace "in the IN chamber" by "CFDC"

Done.

L184: The size threshold to discriminate ice crystals and droplets depends on the RH and T within the CFDC, i.e. the growth conditions of the hydrometeors. Is the indicated threshold valid for all the

experimental conditions within your paper? How does this threshold compare to theoretical hydrometeor sizes assuming pure condensational/diffusional growth? See e.g. Rogers and Yau (1989).

The previously applied ice size threshold of 6 μm would ensure that no droplets would contribute to the ice counts for the temperature range up to -32°C. The size of the droplet mode after the evaporation section decreases with decreasing sample temperature (Korhonen et al., 2020) in part because the evaporation section turns more efficient at lower temperatures. Since, we do not observe any indications of ice formation for temperatures above -36°C, it allows us to apply a lower ice size threshold of 4 μm without liquid droplets contributing to detection of ice crystals. This correction has been performed in the re-analysis of all ice activated fractions presented in the figures 3-6. For the homogeneous freezing reference, there is only a modest difference between applying a threshold of 4 μm versus 6 μm (a factor between 1.3 and 1.7). Differences in the ice-activated fractions from applying an ice size threshold of 4 μm versus 6 μm would typically correspond to factors of about 1.5 to about 5 in the inferred ice-activated fractions for the diesel exhaust samples. Hence, for some of the studied samples, a significant fraction of the ice crystals may appear in the size range from 4 μm to 6 μm. We cannot rule out that ice crystals potentially can be present at sizes below 4 μm. Unfortunately, we do not have a simple way to quantify that whenever ice crystal counts above the ice size threshold are low as typically observed for average lamina temperatures above -38.5°C to -38.0°C. For lower temperatures when ice crystal size modes can be identified in the OPC data, they are in many cases centered around sizes well above 4 μm. However, in some cases, significant and varying fractions of the ice crystal mode appears to be present at sizes below 4 μm overlapping with the size range where liquid droplets may be present. Hence, the maxima in ice-activated fractions observed for temperatures around -41°C may to some extent be biased low due to (i) ice crystals being present at sizes below 4 μm, and (ii) due to only a fraction of the sample being focused in the lamina. It is unclear to which extent those two effects may be coupled – in the sense that the very smallest ice crystals potentially may be associated with particles present outside the lamina. Further detailed studies are needed for such assessments.

L190: Delete "the IN chamber of the"

Done.

L192: Change to: "counts measured during… from the OPC signal. The background values between…"

Done.

L194-200: This part of the description of the uncertainties and error analysis should be expanded and written more clearly (e.g. by using equations), see also my main comment above. E.g. you might want to at least briefly comment how the "statistical error" (L198) compares to the other uncertainties associated with CFDC measurements.

The calculations of random errors follow standard approaches described in basic textbooks. Hence, we do not see a need to document all such very basic and simple steps in a scientific publication. We have included a reference to the textbook: 'Taylor: An introduction to Error analysis' in the lines 196 and 197. As we state in this context in L. 197-199, it is important to keep in mind that the random

error only relates to counting statistics and not any potential biases (systematic errors). The biases in CFDC measurements are known to be pronounced though not well-constrained (Garimella et al., 2017).

L203-213: The discussion of the particle number concentrations used within SPIN should be expanded. Please be more quantitative when discussing the results of Levin et al. (2016) and how these values compare to the number concentrations used in your experiments. For instance, what does "high sample particle number concentration" (L207) mean? Considering the upper limit of your combustion aerosol number concentrations (20000 cm-3; L205) and those used for ammonium sulfate (150 cm-3; L213) and applying a dilution factor of ~10 due to the aerosol-to-sheath flow ratio within SPIN, one still obtains very high number concentrations of around 2000 cm-3 for the combustion aerosols and low concentrations for the homogeneous freezing tests. Can such high number concentrations be reliably detected in SPIN? Can you still ensure that there is one INP per ice crystal at these high concentrations? What would happen if you were to use number concentrations of 2000 cm-3 for your homogeneous freezing tests?

The study by Levin et al. (2016) is, to our knowledge, the only one where the potential effect of water vapor depletion inside a CFDC is studied in detail. Their empirical results are not comparable to our experiments, and their modeling is carried out for other operation conditions (a lamina temperature of -30°C and a nominal supersaturation of 5% with respect to liquid water). Their modeling results indicate that minor water vapor depletion may occur when particle number concentrations in the lamina reach levels somewhere between $10^3$ and $10^4$ cm$^{-3}$, or potentially for higher concentrations depending on the accommodation coefficient used in their model. Hence, it is not straightforward to report anything quantitatively on the potential effect of water vapor depletion in our study, based on the results reported by Levin et al. (2016). This is further complicated by the following fact: It is obvious that potential water vapor depletion inside a CFDC will not depend on the sample number concentration alone, it will depend on the fraction of the particle number concentration focused in the lamina, which serves as cloud condensation nuclei and/or INPs. In other words, very hydrophobic and/or particles too small to serve as neither CCN nor INPs will not play a role in this context. For clarification we have modified L 204-209 from:

"The sampled particle number concentration was diluted to 5000 to 20000 cm$^{-3}$ before introduction to the SPIN (marked 'dilution' in Fig. 1), depending on sample treatment; higher number concentrations were used in experiments involving the PAM with extensive formation of new relatively small secondary aerosol particles. It is known that too high sample particle number concentrations may lead to depletion of water vapor inside a CFDC (Levin et al., 2016), while too low number concentrations reduce the measurement sensitivity."

to:

"The sampled particle number concentration was diluted to 2000-20000 cm$^{-3}$ before introduction to the SPIN (marked 'dilution' in Fig. 1), depending on sample treatment; higher number concentrations were used in experiments involving the PAM with extensive formation of new relatively small secondary aerosol particles. It is known that too high sample particle number concentrations may lead to depletion of water vapor inside a CFDC (Levin et al., 2016), while too low number concentrations reduce the measurement sensitivity. We cannot rule out that the

effective water saturation inside the lamina could had been biased slightly low when the highest particle number concentrations were studied (Levin et al., 2016). However, only a fraction of the studied particles was focused within the lamina, and only a fraction of the particles focused in the lamina was likely to activate into cloud droplets due to small sizes and/or hydrophobicity. Hence, it is unlikely that cloud droplet concentrations inside SPIN exceeded 10000 cm$^{-3}$, for which minor water vapor depletion may occur for slightly different CFDC operation conditions (Levin et al., 2016)."

The number concentrations detected with the SPIN OPC are substantially lower than the sample concentrations detected with the CPC. We have no reason to expect that the SPIN OPC got 'saturated' to a level where the results were significantly biased, at least not for average lamina temperatures above -38°C, where concentrations of super-micron particles were limited. Also, for lower temperatures, we do not observe substantial differences in the ice-activated fractions between samples with concentrations at the order of 200 cm$^{-3}$ over 2000 cm$^{-3}$, to 20 000 cm$^{-3}$.

Considering the very low number concentrations of INPs active at temperatures well above -38°C, we consider it unlikely that more than one INP was present inside ice crystals formed at those temperatures. Ice forming at lower temperatures may be due to homogeneous freezing.

We did not test varying particle number concentrations for the homogeneous freezing experiments. A larger sample number concentration should improve the signal to noise ratio allowing for reliable detection of smaller ice-activated fractions, ranging about an order of magnitude lower in the suggested case. Apart from that, we have no reason to expect any other substantial effects from applying a sample number concentration at the order of 2000 cm$^{-3}$ instead of 150 cm$^{-3}$.

L230: Please indicate approximate number concentrations for these size-selected CCN measurements. Related to the comment above; can competition of water vapor from high aerosol number concentrations lead to a weak CCN signal? How were multiple-charged particles handled?

The CCN concentrations varied substantially between experiments and selected mobility diameter. When CCN concentrations inside the CCN counter approach 5000 cm$^{-3}$, then a minor bias (~10%) is likely to occur due to depletion of water vapour (Lathem and Nenes, 2011). The CCN number concentrations never exceeded 1000 cm$^{-3}$ in our experiments, so there is no reason to assume that depletion of water vapor influenced our CCN results.

The following statement has been added to L230:

"The CCN number concentrations never exceeded 1000 cm$^{-3}$ in these experiments."

The influence from the multiply charged CCN population was identified from the level of the lower plateau in the CCN spectra (Rose et al., 2008). We applied a standard procedure.

L241: Change to: "at seven different wavelengths between 370 nm to 950 nm…"

Done.

L243: add space after "<"

Done.

L252: do you mean ion number concentrations or ionization rates?

We mean ionization rates, expression corrected.

L260: Please specify the type of aging in the OFR. See also my comment above.

Please see our response to your comment on L269 below.

L265: Please define "GMD" on L259 and also give GMD for fossil diesel exhaust. In addition, please specify the standard deviations associated with each of the GMD listed.

We have added definitions for both GMD and GSD, and added the GMD and GSD for all fuels.

L268: add space after "250 ℃"

Done.

L269: How do you know that it is SOA? Combustion exhaust is often also associated with large fractions of hydrocarbon-like organic aerosol (HOA), which can be volatilized. For instance, HOA in engine exhaust is often associated with lubricating oil particles (Canagaratna et al., 2004; Worton et al., 2014); and you note on L149 that your engines was lubricated. Do you have AMS measurement to support your statement?

We know that it is secondary PM as it forms upon processing of exhaust in the PAM reactor. That is to say, engaging the reactor increases the mass concentration several-fold (referred to as "OA enhancement" in the revised manuscript (please see lines 355-370 in the revised version), consistent which oxidation of gas phase precursors that form lower vapor pressure products which condense. The mass spectra show this material is organic, hence we conclude that it is SOA. Indeed the mass spectra of the unprocessed exhaust particles do resemble "hydrocarbon-like organic aerosol" (HOA) while the processed OA resembles "Oxidized Organic Aerosol" (OOA) as observed in numerous field and laboratory AMS experiments. Please see figure 1 below.

[Figure]

**Figure 1**: The effect of PAM treatment on exhaust organic aerosol mass spectra. N.B. the change in y-axis (OA is more abundant after PAM) and the transition from hydrocarbon to oxygen containing peaks.

The text on L273 has been changed from:

"Thermodenuder treatment (250°C) removed a significant fraction of the SOA emissions."

to:

"Thermodenuder treatment (250°C) removed a significant fraction of the OA emissions for both fresh and aged aerosol."

Lube oil and HOA are discussed below on the comment at L344.

L272: change to: "SPIN, particles larger than 100 nm represented…"

Done.

L272: How does this number relate to the number concentrations listed on L205? See my comment above. I would have expected 10% of 2000 cm-3, so the number you state here seems high.

We agree that the highest estimate (in one case) is not the most representative indicator of the number concentrations of large particles. The text has been changed from:

"Regarding the high sample concentrations introduced to the SPIN, the particles larger than 100 nm represented number concentrations up to approximately 1200 cm$^{-3}$, implying a reasonable detection sensitivity."

to:

"Regarding the high sample concentrations introduced to the SPIN, the particles larger than 100 nm represented number concentrations above 50 cm$^{-3}$, thus implying a reasonable detection sensitivity."

L273: change to: "from the total sample number concentration."

Done.

L287: Here and elsewhere (e.g. L291), be consistent on referring to your figures, e.g. use "Fig. 3a".

Done. The references to all figures with two or more panels are now of form e.g. "Fig. 3a".

L289: add space "-41 °C"

Done.

L295: "it can be expected that particles with little surface area have passed through the SPIN without any detectable effect." I interpret your statement that you assume the surface area of the exhaust particles to correlate to their ice nucleation activity. Would it then not be more meaningful to use INAS densities, i.e. normalize to surface area instead of maximum AF? Please also see my main comment above. At the same time, I would like to point out that the recent studies by Nichman et al. (2019) and Mahrt et al. (2018) have identified the ice nucleation mechanism on soot particles as pore condensation and freezing (PCF). More recently the studies by Jantsch and Koop (2021) and Marcolli et al. (2020) have developed detailed frameworks how ice nucleation in complex pores of combustion particles can be modelled/predicted. Can such frameworks also be applied to your particles?

Mahrt et al. (2018) and Nichman et al. (2019) studied deposition freezing. In our study, we operate the CFDC with conditions of relevance to condensation or immersion freezing. Hence, do not expect the reported mechanisms to be of direct relevance to our study. That said, it would of course be possible to investigate the studied soot particles with respect to properties potentially of relevance to deposition freezing.

When it comes to reporting our ice nucleation results, we agree with the reviewer that normalization to particle surface area would be the ideal approach. However, there are a few challenges associated with such an approach. Firstly, we did not measure the particle surface area, and secondly, several not well-constrained systematic errors will have to be corrected for in the calculation. Both issues add to rather substantial errors associated with the INAS density. However, we provide the best possible INAS density results in the supplementary based on the assumption of spherical particles.

L296: Please see my comment to L195: The description of the error analysis should be expanded.

Please see our response above.

L301: I suggest to state this a bit more careful here. Homogeneous freezing of what? E.g. water freezes homogeneously below -38 °C, where you start seeing a signal. Could this be an indication that you are actually freezing pore water on your exhaust particles homogeneously, i.e. that ice nucleation takes place via PCF, even though homogeneous freezing rates at these temperatures are low? (see my comment to L295). Maybe a better formulation would be: "higher temperatures compared to the ice nucleation curve of ammonium sulfate."

As described in detail before, the reference based on ammonium sulfate seeds represents the homogeneous freezing of super-micron sized dilute aqueous droplets, for which the freezing temperature may be biased slightly low by less than 0.1 K relative to that for pure water droplets. The homogenous freezing temperature of aqueous solution droplets increases with increasing droplet size, so we find no reason to expect that significantly smaller volumes of liquid water associated with the combustion particles could freeze homogeneously at temperatures higher than what we observe for our homogeneous freezing reference. Hence, if pores in the soot particles somehow facilitate freezing at temperatures well above the homogeneous freezing reference, then we would consider that heterogeneous freezing.

L300-305: Where to the authors expect aging mechanisms for combustion exhaust particles as sampled here and how to these aging times compare to typical tropospheric lifetimes of particles emitted by diesel engines?

Aging processes are likely to influence soot particle properties in most environments at most times. The respective impacts of various aging processes are likely to depend on parameters such as the actinic flux, available oxidants, condensable vapors, and the relative humidity. Those parameters are to some extent co-varying, but they vary substantially spatially and temporally worldwide. It is outside the scope of this study to go into detail with such complex atmospheric variability. However, it is evident that significant atmospheric aging of soot particles occurs over time scales of a couple to a few hours under various atmospheric conditions (e.g. Riemer et al., 2004; Moffet and Prather, 2009; Eriksson et al., 2017). Hence, it is highly likely that atmospheric aging would influence properties of most soot particles emitted near ground level before they potentially would be transported to mixed-phase cloud levels. Hence, it is clearly of relevance to study whether aging processes influence the ice nucleating ability of soot particles. We studied the effect of simulated intense photochemical aging in a diesel exhaust 'atmosphere', and we only observed modest changes in the ice nucleating ability. However, that does not rule out that other atmospheric aging processes could be of importance in this context.

In the experiments, we applied relatively high levels of photochemical exposure to age the particles in order to observe potential effects on the ice nucleating ability. In other words, we intended to study how significantly different particle properties potentially could be of relevance to ice nucleation properties. Directly linking the applied particle processing to atmospheric processing is not something we can justify doing. Nevertheless, since atmospheric aging of soot particles potentially occurs within a couple of hours, it is most likely so, that if diesel particles emitted near the ground level act as INPs in the atmosphere – then they have in many cases likely underwent atmospheric aging before reaching the cloud base.

L309: change to: "within 0.5 °C of the homogeneous freezing reference"

Done.

L339: What SOA formation took place in the PAM? Please see my previous comments.

Please see our response to your comment on L269.

L343: Write as "C3O2/C3"

Done.

L339-346: For the discussion of the impact of aging and SOA coating on the ice nucleation activity of combustion aerosol, the authors might want to relate their results to previous work, e.g. Zhang et al. (2020).

We acknowledge that this was too briefly described. The following text has been added:

"In the engine-out experiments, very strong SOA formation was obtained upon aging in the PAM. The OA enhancement (Processed OA/Fresh OA) was of a factor of ~20 for diesel and ~7 for RME. After the DOC the OA enhancement was substantially lower (~2 for both HVO and fossil diesel). The exhaust from fossil diesel combustion after simulated atmospheric aging and without emission aftertreatment was dominated by secondary organic mass (OA/eBC =3.4). All experiments except aged fossil diesel and aged HVO at engine-out conditions provided eBC-dominated emissions. Substantial SOA production that led to increased OA/eBC ratios also occurred for RME at engine out and to a lesser extent HVO after the DOC. As shown by Gren et al. (2021), a majority of the formed SOA condensed on the nucleation mode particles and only a minor fraction ended up in the soot mode. This can be related to the engine type and operating conditions that led to relatively low soot emissions, typical for modern engines. The measurements of particle mass of mobility classified particles with the DMA-APM demonstrated a small but significant increase in effective density of soot particles at both 200 nm and 300 nm upon aging (sizes where the soot mode was completely dominating the size distribution without contribution from the nucleation mode). This is consistent with minor SOA uptake on the aged soot particles. However, it cannot be concluded if the SOA was evenly distributed as a thin coating on the primary particles in the soot aggregates or if it us unevenly distributed for example in the pores between the primary particles."

L344: To support your statement on the oxidation state here and also your statement on the "extreme hydrophobicity" of the exhaust particles (L337), would it be possible to provide e.g. elemental oxygen-to-carbon and hydrogen-to-carbon ratios of these particles in addition to the fraction of surface oxides listed in Table 2?

We acknowledge that the statement on the oxidation state was too brief. We have clarified that this refers to refractory surface oxides in the soot cores measured in the SP-AMS (laser on) mode. At present these measurements are not quantitative We add a sentence and a reference discussing how these surface oxides may form during combustion. We also suspect that the reviewer refers to O:C and H:C ratios of the OA (coating) that can readily be quantified in the regular AMS mode. We have added example values for these and discuss the contribution OA may give to the surface properties of soot. We consider that adding the elemental oxygen-to-carbon and hydrogen-to-carbon ratios to Table 2 would include large uncertainties in many of the cases, due to the low mass loadings sampled (and short time).

The following text has been added:

"This is expected as modern diesel engines achieve low BC emissions by highly efficient late cycle soot removal by oxidation by OH radicals (Malmborg et al. 2017). However, the true surface of the mixed particles may also be affected by the organic aerosol. The mass spectra of OA in fresh emissions showed similarities with past experiments where it has been linked to lube oil (Worton et al. 2014). The O:C ratio and H:C ratios of OA measured in the conventional laser off AMS mode was X and Y for typical cases of fresh and aged aerosol respectively."

L344-346: Is this statement based on the OA/eBC ratios listed in Table 2? If I understand your L253 correctly, the OA values in Table 2 were determine from the AMS measurements. These mass loadings seem extremely high, e.g. 2339 µg m-3 for fossil diesel engine-out +PAM. Please comment on the atmospheric relevance of such high mass loadings.

The reported concentrations are dilution adjusted (500-900 times dilution) to allow direct comparison between experiments as dilution ratios varied slightly between each experiment. A note about this has been added in the table heading. The concentrations measured with the AMS were thus low and corresponding to typical atmospheric conditions.

L346-247: The "HVO + engine out" has an OA/eBC ratio of 0.189 according to your Table 2. Do you mean the "RME + engine-out + PAM" instead?

We have revised the discussion on emission particle properties, and this statement has been removed from the revised text. A more detailed analysis on the results is provided instead.

L351: I suggest to explicitly repeat your T and RH conditions here again.

We agree and have added the scanned T-RH range to the statement. The text has been changed from:

"at temperatures relevant to MPC conditions"

to:

"at temperatures relevant to MPC conditions ($RH_{water}$ = 110%, -32 °C > $T$ > -43 °C)."

L353: "and we conclude…" this statement should be phrased more carefully and constrained to the experimental conditions studied here, as other studies have demonstrated that combustion particles can form ice at e.g. cirrus temperatures.

We agree that the formulation may mislead the reader and have corrected it as follows. The text has been changed from:

"and we conclude that the studied diesel emission particles are poor INPs in general."

to:

"and we conclude that the diesel emission particles studied using our setup were poor INPs in MPC-relevant temperature range."

L354: delete "complete"

Done.

L356: should be phrased more carefully, e.g. "increase the IN activity of particles emitted from fossil diesel combustion…"

Corrected as suggested.

L359-360: "With these…"; see my comment to L353.

The statement has been changed from:

"With these observations altogether, it can be implied that the emissions from the studied diesel fuels produce no significant number concentrations of INPs to the atmosphere."

to:

"With these observations altogether, it can be implied that the emission particles from the studied diesel fuels produce no significant number concentrations of active INPs at temperatures above homogeneous freezing. However, the particles can still act as IN in cirrus regime."

L361: I suggest to quantify "ultrafine" here in the conclusions again, i.e. give the values in parenthesis.

Done. Size range (<100 nm in electrical mobility) is now given in parentheses.

L362: What do you mean with "to an extent"? Please be more specific in your conclusion section.

We agree that the statement was ambiguous and therefore, we have omitted it from the revised text.

L363: replace "production" by "signal"

Done.

L365: "slight potential as active INPs"; I suggest to tune this statement down. In the end your observed heterogeneous ice nucleation activity is extremely weak and in the atmosphere such combustion particles will not be able to compete with more efficient INPs such as e.g. mineral dust.

Statement tuned down as suggested. The text has been changed from:

"The only exception to this was for the HVO fuel whose emissions may have a slight potential as active INPs."

to:

"The only exception to this was for the HVO fuel whose particle emissions showed slightly elevated, yet still weak INP activity."

Fig. 2:

Panel 2a:

- Comparing the "engine-out + PAM" with the "engine-out + PAM + TD" it appears that your exhaust particles are associated with a large fraction of semi-volatile material. How do you ensure that this material is not list in the additional dilution stage upstream of SPIN (see your Fig. 1)?

We assume that the referee means lost, instead of "list" as written in the comment. The additional dilution stage, just upstream of the SPIN, was a simple dilutor loop where a part of the sample flow was diverted through a HEPA filter to reduce the sample number concentration. In other words, no external air supply was used in this stage, and no pressure changes took place. This type of dilution should not affect the size distribution.

- Comparing the "engine-out" and the "DOC" curves, does the difference in the signal mean that all particles smaller than approximately 40 nm are not soot but volatile material?

The function of the DOC is to enhance the oxidation of unburnt fuel, hydrocarbons, and carbon monoxide (please see our response to comment on L68). The two latter components are gaseous after exiting the engine, so the particles mentioned in the comment must consist of unburnt fuel. In that sense yes, they are volatile material.

Panel 2b:

- Please add "engine-out curve"

Done.

Panel 2c:

- Why is there an increase at d < 20 nm for the black line, as the red curves in panels a and b suggest that DOC removes most of the particles in this size range?

Regarding that the black curve appears to be unimodal in a similar way than the red curve in panel b, this suggests that there is also a nucleation mode, but it is beyond the size range of the SMPS system. In most other experiments, the nucleation mode occurred at 15 nm which is also close to the low end of this size range.

Fig. 4:

- Error bars for red and blue curves in panel b are missing; please add.

Upon revision of the calculation method, the panel b of Figs. 3-6 has been omitted from the revised manuscript. Therefore, this comment is no longer of relevance.

Fig. 5:

- Why are only error bars for the green curve shown in panel b?

Upon revision of the calculation method, the panel b of Figs. 3-6 has been omitted from the revised manuscript. Therefore, this comment is no longer of relevance.

Table 1

- Change "Fraction" to "Percentage".

Done.

**Added References:**

Eriksson, A.C., Wittbom, C., Roldin, P., Sporre, M., Öström, E., Nilsson, P., Martinsson, J., Rissler, E., Nordin, E.Z., Svenningsson, B. and Swietlicki, E.: Diesel soot aging in urban plumes within hours under cold dark and humid conditions. *Sci.Rep.*, 7, 12364 doi:10.1038/s41598-017-12433-0, 2017.

Mahrt, F., Kilchhofer, K., Marcolli, C., Grönquist, P., David, R. O., Rösch, M., Lohmann, U., and Kanji, Z. A.: The Impact of Cloud Processing on the Ice Nucleation Abilities of Soot Particles at Cirrus Temperatures, *J. Geophys. Res. Atmos.*, 125, 1-23, https://10.1029/2019jd030922, 2020.

Rose, D., Gunthe, S. S., Mikhailov, E., Frank, G. P., Dusek, U., Andreae, M. O., and Pöschl, U.: Calibration and measurement uncertainties of a continuous-flow cloud condensation nuclei counter (DMT-CCNC): CCN activation of ammonium sulfate and sodium chloride aerosol particles in theory and experiment, *Atmos. Chem. Phys.* 8, 1153–1179, 2008.

Lathem, T. L., and Nenes, A.: Water Vapor Depletion in the DMT Continuous-Flow CCN Chamber: Effects on Supersaturation and Droplet Growth, Aerosol Science and Technology, 45, 604–615, 2011.

Moffet, R. C. and Prather, K. A.: In-situ measurements of the mixing state and optical properties of soot with implications for radiative forcing estimates, *P. Natl. Acad. Sci. USA*, 2062, 11872-11877, doi:10.1073/Pnas.0900040106, 2009

Riemer, N., Vogel, H., and Vogel, B.: Soot aging time scales in polluted regions during day and night, *Atmos. Chem. Phys.*, 4, 1885–1893, https://doi.org/10.5194/acp-4-1885-2004, 2004.

**Suggested References:**

Adams, M. P., Tarn, M. D., Sanchez-Marroquin, A., Porter, G. C. E., O'Sullivan, D., Harrison, A. D., Cui, Z., Vergara-Temprado, J., Carotenuto, F., Holden, M. A., Daily, M. I., Whale, T. F., Sikora, S. N. F., Burke, I. T., Shim, J.-U., McQuaid, J. B., and Murray, B. J.: A Major Combustion Aerosol Event Had a Negligible Impact on the Atmospheric Ice-Nucleating Particle Population, 125, e2020JD032938, https://doi.org/10.1029/2020JD032938, 2020.

Bové, H., Bongaerts, E., Slenders, E., Bijnens, E. M., Saenen, N. D., Gyselaers, W., Eyken, P. V., Plusquin, M., Roeffaers, M. B. J., Ameloot, M., and Nawrot, T. S.: Ambient black carbon particles reach the fetal side of human placenta, Nat Commun, 10, 1–7, https://doi.org/10.1038/s41467-019-11654-3, 2019.

Canagaratna, M. R., Jayne, J. T., Ghertner, D. A., Herndon, S., Shi, Q., Jimenez, J. L., Silva, P. J., Williams, P., Lanni, T., Drewnick, F., Demerjian, K. L., Kolb, C. E., and Worsnop, D. R.: Chase Studies of Particulate Emissions from in-use New York City Vehicles, 38, 555–573, https://doi.org/10.1080/02786820490465504, 2004.

Ikhenazene, R., Pirim, C., Noble, J. A., Irimiea, C., Carpentier, Y., Ortega, I. K., Ouf, F.-X., Focsa, C., and Chazallon, B.: Ice Nucleation Activities of Carbon-Bearing Materials in Deposition Mode: From Graphite to Airplane Soot Surrogates, J. Phys. Chem. C, 124, 489–503, https://doi.org/10.1021/acs.jpcc.9b08715, 2020.

Jantsch, E. and Koop, T.: Cloud Activation via Formation of Water and Ice on Various Types of Porous Aerosol Particles, ACS Earth Space Chem., https://doi.org/10.1021/acsearthspacechem.0c00330, 2021.

Korolev, A., McFarquhar, G., Field, P. R., Franklin, C., Lawson, P., Wang, Z., Williams, E., Abel, S. J., Axisa, D., Borrmann, S., Crosier, J., Fugal, J., Krämer, M., Lohmann, U., Schlenczek, O., Schnaiter, M., and Wendisch, M.: Mixed-Phase Clouds: Progress and Challenges, 58, 5.1-5.50, https://doi.org/10.1175/amsmonographs-d-17-0001.1, 2017.

Kreidenweis, S. M., Petters, M., and Lohmann, U.: 100 Years of Progress in Cloud Physics, Aerosols, and Aerosol Chemistry Research, 59, 11.1-11.72, https://doi.org/10.1175/amsmonographs-d-18-0024.1, 2018.

Lohmann, U., Lüönd, F., and Mahrt, F.: An Introduction to Clouds: From the Microscale to Climate, 1st edition., Cambridge University Press, Cambridge, https://doi.org/10.1017/CBO9781139087513, 2016.

Mahrt, F., Marcolli, C., David, R. O., Grönquist, P., Barthazy Meier, E. J., Lohmann, U., and Kanji, Z. A.: Ice nucleation abilities of soot particles determined with the Horizontal Ice Nucleation Chamber, Atmos. Chem. Phys., 18, 13363–13392, https://doi.org/10.5194/acp-18-13363-2018, 2018.

Mahrt, F., Alpert, P. A., Dou, J., Grönquist, P., Arroyo, P. C., Ammann, M., Lohmann, U., and Kanji, Z. A.: Aging induced changes in ice nucleation activity of combustion aerosol as determined by near edge X-ray absorption fine structure (NEXAFS) spectroscopy, Environ. Sci.: Processes Impacts, https://doi.org/10.1039/C9EM00525K, 2020.

Marcolli, C., Mahrt, F., and Kärcher, B.: Soot-PCF: Pore condensation and freezing framework for soot aggregates, 1–68, https://doi.org/10.5194/acp-2020-1134, 2020.

Murray, B. J., O'Sullivan, D., Atkinson, J. D., and Webb, M. E.: Ice nucleation by particles immersed in supercooled cloud droplets, 41, 6519–6554, https://doi.org/10.1039/C2CS35200A, 2012.

Nichman, L., Wolf, M., Davidovits, P., Onasch, T. B., Zhang, Y., Worsnop, D. R., Bhandari, J., Mazzoleni, C., and Cziczo, D. J.: Laboratory study of the heterogeneous ice nucleation on black-carbon-containing aerosol, 19, 12175–12194, https://doi.org/10.5194/acp-19-12175-2019, 2019.

Rogers, R. R. and Yau, M. K.: A short course in cloud physics, Pergamon Press, Oxford; New York, 1989.

Schill, G. P., Jathar, S. H., Kodros, J. K., Levin, E. J. T., Galang, A. M., Friedman, B., Link, M. F., Farmer, D. K., Pierce, J. R., Kreidenweis, S. M., and DeMott, P. J.: Ice-nucleating particle emissions from photochemically aged diesel and biodiesel exhaust, *Geophys Res Lett*, 43, 5524–5531, https://doi.org/10.1002/2016gl069529, 2016.

Schill, G. P., DeMott, P. J., Emerson, E. W., Rauker, A. M. C., Kodros, J. K., Suski, K. J., Hill, T. C. J., Levin, E. J. T., Pierce, J. R., Farmer, D. K., and Kreidenweis, S. M.: The contribution of black carbon to global ice nucleating particle concentrations relevant to mixed-phase clouds, *PNAS*, https://doi.org/10.1073/pnas.2001674117, 2020.

Thomson, E. S., Weber, D., Bingemer, H. G., Tuomi, J., Ebert, M., and Pettersson, J. B. C.: Intensification of ice nucleation observed in ocean ship emissions, 8, 1111, https://doi.org/10.1038/s41598-018-19297-y, 2018.

Vergara-Temprado, J., Holden, M. A., Orton, T. R., O'Sullivan, D., Umo, N. S., Browse, J., Reddington, C., Baeza-Romero, M. T., Jones, J. M., Lea-Langton, A., Williams, A., Carslaw, K. S., and Murray, B. J.: Is Black Carbon an Unimportant Ice-Nucleating Particle in Mixed-Phase Clouds?, 123, 4273–4283, https://doi.org/10.1002/2017JD027831, 2018.

Worton, D. R., Isaacman, G., Gentner, D. R., Dallmann, T. R., Chan, A. W. H., Ruehl, C., Kirchstetter, T. W., Wilson, K. R., Harley, R. A., and Goldstein, A. H.: Lubricating Oil Dominates Primary Organic Aerosol Emissions from Motor Vehicles, Environ. Sci. Technol., 48, 3698–3706, https://doi.org/10.1021/es405375j, 2014.

Zhang, C., Zhang, Y., Wolf, M. J., Nichman, L., Shen, C., Onasch, T. B., Chen, L., and Cziczo, D. J.: The effects of morphology, mobility size, and secondary organic aerosol (SOA) material coating on the ice nucleation activity of black carbon in the cirrus regime, 20, 13957–13984,

https://doi.org/10.5194/acp-20-13957-2020, 2020.

---

## Author Comment (AC2)

We thank Referee #3 for his or her comments and constructive suggestions on how to improve the contents of the manuscript. The comments as posted are listed below on green font, our responses to them on red font and the specified modifications to the text on blue font.

Review of Manuscript acp-2021-111:

Particle emissions from a modern heavy-duty diesel engine as ice-nuclei in immersion freezing mode: an experimental study on fossil and renewable fuels, by Korhonen et al.

**General comments:**

Korhonen et al. perform systematic experiments about the ice nucleation ability of diesel engine particulate emissions at high relative humidity and mixed phase cloud relevant conditions, with auxiliary measurements about particle size distribution, hygroscopicity and chemical composition. The ice nucleation activity data and conclusion presented, demonstrating the poor ice formation ability of diesel engine particles and limited effects from photochemical ageing processes on the particle ice nucleation activity, are of interests to the ice nucleation community. However, some part of discussion and data interpretation are not thoroughly or comprehensively presented. I would like to suggest further revisions before recommending acceptance.

The ice nucleation experiments performed cover a variety of diesel engine emissions generated by three kinds of fuels. Different engine exhaust treatment techniques are used to mimic diesel engine particulate emission atmospheric relevant ageing processes. The research interests are of significance of the atmospheric ice nucleating particles and thus climate. An interesting ice nucleation story is clear. Nevertheless, some improvements need to be made, regarding to data interpretation and results discussion. In general, I have five major comments on this manuscript.

The authors should clarify their samples and research focus clearly and construct an unambiguous approach about the research story, illustrating the relations among all the measurements. For instance, the manuscript title and abstract tell me the particulate emission from a diesel engine will be the object for this experimental study, but the authors directly introduce soot particles in the introduction and the following parts. Note, the particulate emissions from diesel engines are not only comprised of soot particles. Differentiating the concept of soot particles from diesel engine particulate emissions is necessary. The authors also need to explain why soot particles emitted by land transportation diesel engines are atmospherically relevant.

We have added the following paragraph to the introduction, and consider that it clarifies the atmospheric relevance of the studied particles:

"Particulate emissions from diesel engines can form a notable fraction of total aerosol burden in urban areas, especially in regions where diesel-powered vehicles outnumber ones using different types of fuel (DeWitt et al., 2014). For instance, such regions can comprise arterial roads near seaports, transport hubs, or other facilities whose function relies on heavy-duty transport. In addition to impairment of air quality near the surface level, convective draft mixes the aerosol within the atmospheric boundary layer and can lift the particles to altitudes up to several kilometers, depending on weather conditions. Studies focusing on urban impacts on climate and weather show evidence of precipitation anomalies downwind from large cities (Han et al., 2012; Han et al., 2013; Zhong et al., 2015). Hence, understanding the role of urban aerosol particles in cloud processes is essential for further understanding of the origins of the abovementioned anomalies in precipitation due to urban aerosol particles. However, the number of previous studies focusing on the ice nucleating potential of fossil and renewable diesel fuels (Schill et al., 2016; Chou et al., 2013) is limited, to our knowledge."

Second, I highly recommend to harmonise the ice nucleation terminology through the manuscript, see Vali et al. (2015) as a reference. The aim is to make dissemination more uniform and consistent within the community.

We agree on the importance of uniform terminology and explain why we refer to immersion freezing in the methodology section of the revised manuscript. Also reference to Vali et al. (2015) added. For instance, the following statement has been added to Sect. 2.2:

"The high relative humidity on water enabled investigation of the IN activity in immersion freezing mode as it is defined by Vali et al. (2015) because liquid formation on the sample particles is expected before possible ice formation takes place; immersion and condensation freezing modes are indistinguishable in the SPIN."

Third, the ice nucleation pathway, immersion mode freezing, is not well introduced. Only referring to a reference (Korhonen et al. 2020) without a brief introduction about how this can be achieved is not enough, in my opinion. Both the concept and the approach to achieve it should be explained even as a summary and then referring to the previous literature can be of more clarity and convincingness.

We acknowledge that the initial formulation required reading Korhonen et al. (2020) and have revised Sect. 2.2 in a way that it explicitly explains the behavior of droplets and ice crystals inside the SPIN, when the settings used in this study are applied. Besides, the reasoning for using the 4  $\mu$ m size threshold in size-separation is now explained. The reasoning for the selection of the size threshold has been added to the text:

"Therefore, we used basic size-separation for ice crystals and droplets and consider particles larger than 4  $\mu$ m ice crystals. Given the high relative humidity, 110% over water and 150-163% over ice (depending on lamina temperature during the T-scan), the formed ice crystals grow fast inside the SPIN to optical sizes much higher than the 4  $\mu$ m threshold used, typically to a range between 7-15  $\mu$ m."

Forth, a new approach to present normalized ice activation fraction results should be explained more clearly in Sect. 2 and 3. Also, it should be applied carefully. Note that, the particle samples are polydisperse with different particle size distributions (see Fig. 2) and the proportion of large particles (e.g. > 100 nm  $\sim$  10 %) is still comparable or even higher than the highest ice activation fraction measured. For instance, the highest ice activation fraction for unaged RME fuel engine particles is less than 0.35 % even at the lowest temperature addressed, as shown in Fig. 6. To figure out the contribution of large particles (e.g. 100, 200 or 300 nm) to the ice activation fraction, ice nucleation experiments for size selected (e.g. 100, 200 or 300 nm) large particles are expected to performed. In addition, the data processing or the calculation method needs to be formulated and then the specific equation can help the understanding.

We have re-analyzed all SPIN data, and we have identified an artifact in the data processing. Whenever the SPIN OPC counts per second exceeds a certain threshold, then the SPIN data are stored in a slightly different manner, which is very simple to correct for. That correction was unfortunately not carried out for the ice-activated fractions (AFs) presented in figures 3 to 6 in the previous version of the manuscript. The correct AFs may be up to about 1 order of magnitude higher than what was previously presented for the very lowest temperatures around -41°C, while the differences in the AFs were relatively smaller or insignificant for temperatures above -39°C. Generally, these corrections do not affect the main findings of little to no indications of heterogeneous immersion freezing. Previously, we interpreted the pronounced low AF levels for several diesel samples near T=-41°C relative to the homogeneous freezing reference to be due to small and/or very hydrophobic diesel particles not activating into cloud droplets with the potential for subsequent immersion freezing inside SPIN. For that reason, we found that there was a risk of misinterpreting low AFs as a result of low immersion freezing ice nucleating ability – rather than an effect of very low CCN activity. That was the motivation for the AF scaling carried out. With the correctly analyzed AFs, there are still indications of low CCN activity influencing the AFs but to a significantly less pronounced level. We still think that it is of relevance to consider how high concentrations of nucleation mode particles and/or very hydrophobic seed particles may bias the immersion freezing AF low by contributing to the total particle concentration while not acting as CCN/INPs. However, we no longer consider it a risk that the 'raw' AFs can be misinterpreted in this context, so we have decided not to carry out the AF scaling and the figures 3.b to 6.b do not appear in the revised version of the manuscript. Consequently, the paragraphs in L215-225 and L273-282 have been deleted.

The main foci of the study were to (1) directly link the ice-nucleating ability of the studied particles to their physico-chemical properties, and (2) to investigate to which extent simulated photochemical aging potentially would influence the ice-nucleating ability of the particles. Hence, there were a number of constraints on the particle number concentration and particle mass that could be studied. It would typically not be possible to obtain reasonable number concentrations of 200 or 300 nm mono-disperse particles for the SPIN measurements. It is not uncommon to study poly-disperse particle populations with a CFDC as it is generally done in ambient measurements and a number of laboratory studies (e.g. Petters et al., 2009; Levin et al., 2016). Furthermore, the fact that we do not observe any indications of heterogeneous freezing for temperatures well above the homogeneous freezing reference, indicates a low ice-nucleating ability for any particle size of the studied population. Thus, with the experimental constraints and the scientific aims, it was not meaningful to carry out size-selected CFDC measurements, and it is highly questionable whether such activities would add any substantial additional and essential information regarding the low ice-nucleating ability of the studied particles.

Finally and most importantly, the results discussion is performed not thoroughly and reasonably. For example, comparing the ice nucleation results of polydisperse aerosol dominated by fine particles (< 100 nm) with the homogeneous freezing of larger (350 nm) ammonium sulfate (AS) particles is inappropriate. Instead of 350 nm AS particles, the comparison with the homogeneous freezing results of small (< = 100 or 150 nm) AS particles would be of more relevance to the results of diesel soot particles which has a small size distribution. Especially, the diesel engine particles already exhibit a homogeneous freezing depression event at temperatures lower than the homogeneous freezing temperature at such a high relative humidity ( $RH_w = 110 \%$ ). In addition, the

ice nucleation data is not well linked to the auxiliary measurement results. Similar findings in the literature are also helpful to support the conclusion (see detailed comments in next part).

It is not clear why the reviewer finds that it would make more sense to study e.g. 100 nm particles rather than 350 nm ammonium sulfate particles to obtain a homogeneous freezing reference, since no scientific arguments are provided supporting that point of view. In this context, it is relevant to point out that the critical supersaturation needed to activate a soot particle with a mobility diameter of 100 nm is much higher than the corresponding value for a 100 nm ammonium sulfate particle. So, we do not expect that to be the reason, and we cannot think of another reason to justify the argument.

The aim of the homogeneous freezing reference is simply to detect when dilute aqueous cloud droplets freeze. The chosen ammonium sulfate particle size should ensure that (1) as large a fraction as possible of the particles activate into droplets inside SPIN, and (2) the dissolved ammonium sulfate seeds in the cloud droplets do not depress the freezing point substantially.

We speculate, that for ammonium sulfate particles located in the vicinity of the lamina closest to the cold wall would contribute to droplet formation and freezing, with that contribution potentially increasing with the increasing particle seed size (associated with lower critical supersaturation). The freezing point depression is about 0.03 K for 4  $\mu$ m droplets and 350 nm ammonium sulfate seeds following Ignatius et al. (2016). Hence, we do not expect the solute effect to substantially lower the freezing point, while a more than reasonable fraction of the particles is likely to contribute to the formation of dilute droplets. Hence, we consider 350 nm particles a reasonable choice in this context. We could of course obtain very similar spectra for e.g. 100 nm seed particles, but the comparability to measurements carried out years ago is questionable due to potential minor differences in the flow control and aspects similar to that.

**Specific comments:**

Line 17: change 'continuous-flow diffusion chamber' to the same as it is in Line 87

Done.

Line 19: change to '-43 and -32°C'. The same for Line 43, 92, 96, 175, 234 and 268. Please check through the mathematical notation and make it satisfy ACP terminology.

Done.

Line 23: change 'present' to 'presented

**The sentence has been rephrased from:**

"In addition to ice-nucleation experiments, we used supportive instrumentation to characterize the emission particles and present six different physical and chemical properties of them.

to:

"In addition to ice-nucleation experiments, we used supportive instrumentation to characterize the emitted particles for their physicochemical properties and presented six of them.

Line 24: make 'different emission after-treatment systems' specified

The text has been changed from:

"We found that the studied emissions were poor ice-nucleators and substitution of fossil diesel with renewable fuels, using different emission after-treatment systems and photochemical aging of total exhaust had only little effect on their ice-nucleating abilities."

to:

"We found that the studied emissions contained no significant concentrations of ice nucleating particles likely to be of atmospheric relevance. The substitution of fossil diesel with renewable fuels, using different emission after-treatment systems such as a diesel oxidation catalyst, and photochemical aging of total exhaust had only little effect on their ice-nucleating abilities."

Line 27: change to 'the radiative forcing of the Earth and thus climate in different ways'

**Following the recommendations by all referees, the text has been changed from:**

"Atmospheric aerosols affect the radiative forcing budget of the Earth and thus climate in multiple different ways, directly through absorption/scattering of radiation and indirectly through impacts on cloud properties."

to:

"Atmospheric aerosols affect the energy budget of the Earth and thus climate in different ways: directly through absorption and scattering of heat and light, respectively, and indirectly via affecting cloud formation and lifetime."

Line 39: change to 'homogeneous ice nucleation'

Done.

Line 41: change to 'ice nucleating particles' and specify its abbreviation 'INPs'

According to suggestion by Referee #4, we have rephrased the sentence from:

"but there is evidence that combustion emissions from different hydrocarbon fuels can have potential as active ice-nucleating particles in temperatures higher than that."

to:

"but there is evidence that combustion emissions from hydrocarbon fuels have the potential to nucleate ice at higher temperatures."

In the revised text, we introduce the abbreviation INP later on L44.

Line 39 to 41: Please provide evidence or reference to show combustion emissions are relevant to the lower troposphere ice nucleation activities.

We assume that Referee #1 has also pointed out the same aspect and have added Ikhenazene et al. (2020) and Thomson et al. (2018).

Line 44 and 45: If you write that soot particles are not active INPs, the relative humidity and temperature condition also need to be reported.

We have added mentioning that the reported soot particles were not active INPs at MPC-relevant temperature and humidity conditions.

Line 50: the reference 'Mahrt et al. 2018' should be irrelevant to the atmospheric aging processes for INPs but Mahrt et al. (2020a) and (2020b) can be references.

Removed Mahrt et al. (2018), suggested references added to replace it.

Line 58 and 59: change to 'the climate forcing due to anthropogenic soot particles immersion freezing'

Done.

Line 60: change to 'ice nucleation abilities'

Done.

Line 72 to 74: The environmental pollution caused by diesel engine without DPF or DOC technique is not relevant to this research topic.

Following the suggestion by Referee #1, we have revised the entire paragraph to be more concise and more focusing on 1) the atmospheric relevance of diesel emissions and 2) the motivation for ice nucleation studies, such as this one. Please check the revised chapter where this statement is corrected.

Line 129 to 131: was the Aerosol Instrument Manager (AIM) software used to log the SMPS data? If so, the SMPS scan size upper limit should be much larger than 500 nm with such a high sheath to aerosol sample flow ratio (10:1) and a 180 s scanning time. And if the size scan did not cover the whole range of the aerosol particle size distribution, the multiple charge correction is not finished and then the results are biased by the uncomplete correction calculation.

Yes, we used the AIM software. The parameters describing the SMPS settings in the original manuscript were unfortunately wrong, we deeply apologize for this. The nominal flow rates used in the majority of the campaign was 5 lpm sheath and 1 lpm aerosol flow rate. The size range was 11-500 nm as stated previously. The number fraction of particles above 500 nm in the soot mode was according to the lognormal fits less than 0.5%. This means there were only very small uncertainties from the multiple charge correction. There is some minor noise in the channels 450-500 nm for some of the experiments as seen in figure 2. These are not from multiple charge correction. The signal is extremely low at these channels it is unclear where this noise comes from.

Line 130 to 142: Better to introduce the measurements work flow following the sample flow sequence depicted in Fig. 1.

We have revised the chapter to depict the work flow better.

Line 179 to 181: The authors need to make a more conceivable and clear statement for distinguishing water droplets from ice crystals. I understand that the basic idea is to let the OPC running in different size channels and then to differentiate the particle phase according to their survival abilities through the evaporation section, i.e. water droplet can be evaporated because of the relative humidity condition. The statement about CCN ability and immersion mode freezing make readers confused. In addition, referring to the study performed by Korhonen et al. (2020) as an example does not make sense for me. This is because the samples are different between the current study (i.e. diesel engine particulate emissions) and the previous study (i.e. particulate emissions from solid-biomass-fired cookstoves). The OPC channel size used to discriminate water droplets from ice crystals should be stated from the current study.

The reference to Korhonen et al. (2020) relates to the experimental approach and associated advantages and limitations. The approach can be applied in general for any type of particle studied, so we disagree with the reviewer comment in this context, and it would be bad practice not to let readers know where the relevant information about the instrument operation was introduced.

**The statement:**

"The main motivation for this operation procedure is 1) to ensure that a significant number of particles with low CCN activity form droplets inside SPIN, and 2) that immersion freezing over a wider temperature range can be investigated."

**has been modified to:**

"In the present study, the main motivation for this operation procedure was 1) to ensure that particles with low CCN activity form droplets inside SPIN due to a relatively high supersaturation, and 2) that immersion freezing over a wider temperature range can be investigated within a single run."

Using size to discriminate between ice and non-ice is fundamental in CFDC studies, and widely applied. The SPIN OPC is advanced and not limited to discrete channels. The ice size threshold has been changed from 6 to 4  $\mu$ m, which was clearly stated in the previous as well as the revised manuscript.

Line 190: change to 'exiting the IN chamber' or 'exiting the SPIN'. Or, the authors can decide to use 'SPIN' or 'the SPIN' through the whole manuscript.

Done. We have decided to use form 'the SPIN' throughout the text.

Line 215 to 225: In this paragraph is not well organised. In my point of view, the authors may need to explain how the freezing of a particle immersed in a water droplet could happen when the temperature decreases lower than the homogeneous freezing temperature (HNT), to illustrate the results presentation. A suggestion could be that sample particles might be activated as cloud

droplets at RHw = 110 % for temperature conditions higher than the HNT, thus makes it possible to investigate the particles immersion mode freezing ability at T < HNT in the flowing temperature scan because a droplet would freeze homogeneously when T is lower than HNT. Here again, the ice crystal formation of droplet activated particles at T < HNT should be homogeneous freezing. If the authors claim this is immersion freezing, evidence of this should be presented. But if the freezing occurs at RH conditions above homogeneous RH condition at the same T, then it is unclear how the authors can conclude immersion freezing to be the relevant mechanism.

It seems as if the reviewer is of the impression that there is a single homogeneous freezing temperature (HFT) determining whether a droplet will freeze or not. We do not share that point of view, and in the following we will describe why the observed ice spectra are very much in line with what can be expected from homogeneous freezing.

Homogenous freezing is generally expected to be explained by classical nucleation theory (Ickes et al., 2015). Hence, the freezing probability of a droplet is a function of temperature, time, and droplet size. In addition, solutes may depress the freezing point as described further above.

Inside SPIN, the aerosol sample located in the lamina will be exposed to a range of temperatures, supersaturations and different residence times in the lamina due to the non-uniform flow profile.

It is not straightforward, which droplet size and corresponding residence time of such a droplet in the growth part of the SPIN chamber that should be applied in the calculations. Ignatius et al. (2016) estimated that inside the SPIN, a frozen fraction of 10% could be expected pure water droplets with a temperature between -38.2 and -37.6°C. In the figures 3-6, the average lamina temperature is shown on the ordinate axis. Hence, roughly 50% of the aerosol population can be expected to be exposed to either a lower or alternatively a higher temperature than the average lamina temperature depending on the respective particle trajectories inside SPIN. That, in conjunction with a higher freezing probability for lower droplet temperatures explain why the ice-activated fraction increases with decreasing average lamina temperatures in the temperature range from -41 to -38°C. So qualitatively, the ice spectra are in line with what can be expected for homogenous freezing.

When it comes to the quantitative aspects of the ice-activated fraction, then the AFs are lower than what we could expect from the theoretical considerations by Ignatius et al. (2016). As discussed in the present manuscript, we would expect the AFs to be biased significantly low due to only a fraction of the sample particles being focused in the lamina (Garimella et al., 2017). In addition, a few other effects may bias the ice crystal counts low, such as (i) the potential presence of ice crystals of sizes below 4  $\mu$ m, and (ii) losses of ice crystals between the evaporation section and the OPC detection volume. Finally, the 'theoretical' calculations related to (i) homogeneous freezing conditions, and (ii) the lamina conditions inside SPIN are associated with some uncertainty. Hence, overall, we find a reasonable agreement between observations and what can be expected from 'theory' considering relevant errors and biases. Since most potential biases are likely to be comparable between samples studied with identical SPIN operation conditions, we consider the homogeneous freezing reference a reasonable reference for comparison within the present study.

Line 223 to 225: A clear definition for the normalization of the ice activation fraction curves for each sample should be made. A formulation for this approach or an example may help.

We have omitted this normalization method from the revised manuscript and present only the 'classical' method for calculating the AF in the text. Hence, the lines mentioned in the comment have been deleted.

However, we have added the INAS density normalization and present its formulation and results in supplementary information.

Line 232 and 233: change 'L/min' to 'L min-1'. Please check the unit through the manuscript.

Corrected to form "L min-1" throughout the revised text.

Line 235: The CCNC calibration curves should be provided in the following section or in an Appendix part.

We have included a figure with the CCNC calibration curves in the supplementary information. We also provide the necessary information regarding the figure.

Line 251: The calibration results should be provided in the following section or in an Appendix part.

We have included a figure with the CCNC calibration curves in the supplementary.

Line 265: What is the 'GMDs'?

The GMD means geometric mean diameter of the particles, we have defined the acronym in the revised text.

Line 285: change to 'Ice activation fraction curves for fossil diesel emissions are presented in Figs. 3 and 4'

Done.

Line 287: change to 'Fig. 3a'. And the similar suggestion to that of Line 298, 310, 314, 324, 326 and 327. Please check the abbreviation for 'Figure' through the manuscript.

Done.

Line 291: change to 'Fig. 3b'; Specify which two samples

Done.

Line 294: Here, what is the size range for the so-called ultrafine particles? It should be 100 nm if the number 90 % refers to the size distribution results mentioned in Sect. 3.1. Please make the ultrafine particle with a quantitative value for clear discussion.

Quantification added.

Line 300 to 307: The discussion in this paragraph can be better. First, the ice formation enhancement by lowering the ice onset temperature values should be clearly connected to the sample to make it

easier for readability. Second, some evidence form auxiliary measurements should be provided to interpret the results. Also, relevant studies in the literature can be referred to for comparison, e.g. Zhang et al. (2020) also investigated the photochemical aging effects on soot particles ice nucleation activities at T < HNT.

The focus of that section is on the potential of heterogeneous immersion freezing. Hence, Zhang et al. (2020) is not a relevant study to consider in this context, since they studied deposition freezing.

The discussion concerning the ice-nucleation results from photochemically treated particles has been changed from:

"Photochemical aging was found to have little effect on ice-activity, as Fig. 4 shows: for the whole sample population, the most ice-active case produced ice onset at approximately 0.5 °C higher temperature than homogeneous freezing. The normalization test revealed that the highest activity was only slightly higher than in unaged emissions (1.2 °C vs. 0.6 °C, relative to the homogeneous freezing reference), which indicates that the simulated photochemical aging through OH and O3 exposure has some, but only little effect on IN activity of fossil diesel emissions. The aged samples showed similar behavior in ice-activity relative to the unaged ones, with maximum activated fraction reaching up to 1.7% from whole sample population, regardless of the equivalent atmospheric aging time. Generally, the ice-activated fractions never reached above 1.7% in any experiment on polydisperse sample emissions from fossil diesel."

to:

"Photochemical aging was found to have little effect on the observed ice-activated fractions. It may be non-ideal to compare the ice-activated fractions directly between the PAM-processed samples versus the fresh emissions. The particle number size distributions may differ significantly between the samples as illustrated in Fig. 2, so the ice-activated fractions have been normalized to particle number concentrations representing different average particle sizes – and different hygroscopic properties. Nevertheless, we observe no indications of heterogeneous freezing more than about 1.0 °C above the homogeneous freezing in any experiment. Considering the potential errors and biases, we find it unlikely that the type of photochemical aging simulated in these experiments are likely to significantly improve the ice-nucleating ability of the engine emissions studied. However, that does not rule out that other types of atmospheric processing can be of importance for the ice-nucleating ability of diesel engine emissions."

Line 313: change to 'the lowest temperature'

Done.

Line 316 and 317: Arguing that the -36.1 °C is outside of the instrument uncertainty should refer to the homogeneous freezing temperate detection ability of SPIN.

We agree, reference to that added.

Line 327 and 328: Why use the size distribution results about fossil fuel in Fig. 2a ('left-hand panel of Fig. 2' in text) to interpret the ice activation results of RME emissions?

We apologize the wrong reference to the figure, corrected to "Fig. 2c".

Line 335 to 347: The discussion in this paragraph is too general and not specific enough. For example, the auxiliary measurement results for each sample should be connected to the sample directly, instead of making a general statement or a conclusion (e.g. Line 336 to 338 about CCNC results) for the overall study. The statements also should be clearly related to the quantitative values obtained from the supportive measurements. In addition, explanation or definition about each measurement result, e.g. OA,  $C_{11}/C_3$ , should be made in the main text. Necessary references in the literature also need to be referred.

We acknowledge this shortcoming and have added depth to the discussion. For instance, the analysis regarding the CCNC has been expanded. The statement:

"The CCNC experiments showed no CCN activity in any experiment without the PAM, which is a clear indicator of extreme hydrophobicity of the studied particles that was also observed in the SPIN as weak droplet formation despite high supersaturation RHwater =110 % during the T-scans. The PAM experiments where SOA formation on diesel emission particles took place increased the  $\kappa_{\alpha}$  values to distinguishable levels, yet this increased hygroscopicity had no observed effect on ice-activity."

**Has been replaced with:**

"Detection of significant concentrations of CCN for any of the bypass or PAM-TD experiments, even for the highest supersaturation approaching ~2.4%, was impossible. We conclude that the associated particles were hydrophobic. For the PAM experiments, we observed droplet formation, and it was possible to infer the CCN activity. For the PAM experiments related to different fuels, the 58 nm particles had kappa values of 0.08-0.10, the 107 nm particles had kappa values of 0.04-0.08, while they were about 0.01 for the 196 nm particles. The number concentrations for the 296 nm particles were too low for the CCN measurements. An increase in the κα when the aerosol is exposed to photochemical ageing can be due to (i) formation of SOA, (ii) oxidation of the soot surface, and (iii) collapse of fractal soot aggregates (Tritscher et al., 2011). Gren et al. (2021) reported the effective density to increase significantly for the very same particles after sampling through PAM, which is indicative of significant SOA coatings which appeared to be relatively more pronounced for the 58 and the 107 nm particles. In combination with the dramatic increase in  $\kappa\alpha$  from undetectably low to levels of up to 0.1, formation of SOA is likely to be the dominant process. The kappa for SOA formed from diesel engine exhaust has been reported in the range 0.09 to 0.14 (Tritscher et al., 2011). Following the approach described by Wittbom et al., (2014) about the  $\kappa\alpha$  of aged diesel soot particles, our observations indicate that the aged 58 nm particles were likely to be dominated by SOA by volume, while the 107 nm particles were likely to be soot particles with a significant SOA coating. The more modest kappa values for the 196 nm are indicative of a relatively lower ratios between the SOA and the primary soot particle volumes."

We would expect the increase in  $\kappa_{\alpha}$  and associated increase in particle volume due to the PAM processing to increase the fraction of particles activating into cloud droplets inside SPIN. Hence, that would increase the number of particles potentially contributing to immersion freezing. On the other hand, the PAM processing clearly leads to pronounced new particle formation with modes centered near ~20 nm (Fig. 2). It is questionable to which extent such particles (i) act as CCN inside SPIN, and (ii) subsequently may play a role as INPs in immersion freezing mode. Hence, the ice-activated fractions observed with SPIN may get biased high for the PAM-processed aerosol due to a larger

fraction of soot particles being immersed in cloud droplets, or the activated fraction may get biased low due to the pronounced increases in nucleation mode particles, which are unlikely to contribute significantly to immersion freezing as studied with SPIN. These observations illustrate that it is complicated to directly compare the AF between these samples with substantially different  $\kappa_{\alpha}$  and particle number size distributions.

Line 348: change to 'Summary and conclusion'. Because the discussion is largely presented in previous sections and this part is more about conclusions.

**Done.**

Line 349 to 368: I disagree with the logicality in this part. On the one hand, the authors conclude that small diesel engine particles have no contribution to ice nucleation activities (Line 361). On the other hand, they are comparing the ice nucleation ability of the particles produced by different fuels. The authors need to firstly demonstrate ice nucleation activity via the immersion mode really occurs then they can make statement about the efficiency of the soot particles as potential ice nucleating particles (INPs). The reference about the study presented by Kanji et al. (2020) in Line 354 is inappropriate, which states that their findings are in complete agreement with Kanji et al. (2020).

It is not possible to discern between condensation freezing and immersion freezing with typical CFDC measurements. However, as described above, the homogeneous freezing reference agrees well with what could be expected from homogeneous immersion freezing considering potential errors and biases. In addition, we have replaced the statement including the reference to Kanji et al. (2020) from:

"Besides, our results are in complete agreement with Kanji et al. (2020) who studied whether hydrocarbon soot from propane combustion and different commercially available black carbon particles can induce immersion freezing and found all of them inefficient ice-nucleators."

**to:**

"Our results, in conjunction with a lot of previous studies indicate that a wide range of black carbon and soot particle types are inefficient as immersion freezing INPs"

**Figures and Tables:**

Figure 1: I cannot find where the 'FPA-fast particle analyser' is in the figure. It is not mentioned in the main text, either.

The referee is correct that no FPA data was used in our analysis. We have removed the mentioning from the caption.

Figures 2: The size distribution measurement for 'engine-out + BP' sample presented in Fig. 5 is missed in Fig. 2b and should be provided. And there is no SPIN experiment corresponding to the sample 'DOC + PAM' in Fig. 2b. In addition, it would be helpful if the figure grids are on to guide reader's eyes.

Upon revision of the calculation method, the panel b of Figs. 3-6 has been omitted from the revised manuscript. Therefore, this comment is no longer of relevance.

Figure 3: Is the ice activation curve for "Engine-out + BP' sample normalized by the sample approach as those of other samples? The highest ice activation fraction should be the unity. It looks in corrected.

Upon revision of the calculation method, the panel b of Figs. 3-6 has been omitted from the revised manuscript. Therefore, this comment is no longer of relevance.

**Added References:**

Garimella, S., Rothenberg, D. A., Wolf, M. J., David, R. O., Kanji, Z. A., Wang, C., Rösch, M., and Cziczo, D. J.: Uncertainty in counting ice nucleating particles with continuous flow diffusion chambers, Atmos. Chem. Phys., 17, 10855–10864, https://doi.org/10.5194/acp-17-10855-2017, 2017.

Gren, L., Malmborg, V.B., Falk, J., Markkula, L., Novakovic, M., Shamun, S., Eriksson, A., Kristensen, T.B., Svenningsson, B., Tunér, M., Karjalainen, P. and Pagels, J., Effects of renewable fuel and exhaust aftertreatment on primary and secondary emissions from a modern heavy-duty diesel engine, J. Aerosol Sci., 156, 105781, 2021.

Ickes, L., Welti, A., Hoose, C., and Lohmann, U.: Classical nucleation theory of homogeneous freezing of water: thermodynamic and kinetic parameters, Phys. Chem. Chem. Phys., 17, 5514–5537, 2015.

Ignatius et al.: Heterogeneous ice nucleation of viscous secondary organic aerosol produced from ozonolysis of a-pinene, Atmos. Chem. Phys., 16, 6495–6509, https://doi.org/10.5194/acp-16-6495-2016, 2016.

Ikhenazene, R., Pirim, C., Noble, J. A., Irimiea, C., Carpentier, Y., Ortega, I. K., Ouf, F.-X., Focsa, C., and Chazallon, B.: Ice Nucleation Activities of Carbon-Bearing Materials in Deposition Mode: From Graphite to Airplane Soot Surrogates, J. Phys. Chem. C, 124, 489–503, https://doi.org/10.1021/acs.jpcc.9b08715, 2020.

Levin, E. J. T., McMeeking G.R., DeMott, P.J., McCluskey C.S., Carrico C.M., Nakao S., Jayrathne T., Stone E.A., Stockwell C.E., Yokelson R.J. and Kreidenweis S.M., Ice-nucleating particle emissions from biomass combustion and the potential importance of soot aerosol, J. Geophys. Res. Atmos., 121, 5888–5903, doi:10.1002/2016JD024879, 2016.

Thomson, E. S., Weber, D., Bingemer, H. G., Tuomi, J., Ebert, M., and Pettersson, J. B. C.: Intensification of ice nucleation observed in ocean ship emissions, 8, 1111, https://doi.org/10.1038/s41598-018-19297-y, 2018.

Tritscher, T., Jurányi, Z., Martin, M., Chirico, R., Gysel, M., Heringa, M. F., DeCarlo, P. F., Sierau, B., Prévôt, A. S. H., Weingartner, E., and Baltensperger, U.: Changes of hygroscopicity and morphology during ageing of diesel soot, Environ. Res. Lett. 6, 034026, 2011.

Wittbom, C., Eriksson, A. C., Rissler, J., Carlsson, J. E., Roldin, P., Nordin, E. Z., Nilsson, P. T., Swietlicki, E., Pagels, J. H., and Svenningsson, B.: Cloud droplet activity changes of soot aerosol upon smog

chamber ageing, Atmos. Chem. Phys., 14, 9831–9854, https://doi.org/10.5194/acp-14-9831-2014, 2014.

**Suggested References:**

Kanji, Z. A., Welti, A., Corbin, J. C., and Mensah, A. A.: Black Carbon Particles Do Not Matter for Immersion Mode Ice Nucleation, Geophys Res. Lett., 47, https://10.1029/2019gl086764, 2020.

Korhonen, K., Kristensen, T. B., Falk, J., Lindgren, R., Andersen, C., Carvalho, R. L., Malmborg, V., Eriksson, A., Boman, C., Pagels, J., Svenningsson, B., Komppula, M., Lehtinen, K. E. J., and Virtanen, A.: Ice-nucleating ability of particulate emissions from solid-biomass-fired cookstoves: an experimental study, Atmos. Chem. Phys., 20, 4951-4968, http://10.5194/acp-20-4951-2020, 2020.

Mahrt, F., Alpert, P. A., Dou, J., Gronquist, P., Arroyo, P. C., Ammann, M., Lohmann, U., and Kanji, Z. A.: Aging induced changes in ice nucleation activity of combustion aerosol as determined by near edge X-ray absorption fine structure (NEXAFS) spectroscopy, Environ. Sci.: Processes Impacts, https://10.1039/c9em00525k, 2020b.

Mahrt, F., Kilchhofer, K., Marcolli, C., Grönquist, P., David, R. O., Rösch, M., Lohmann, U., and Kanji, Z. A.: The Impact of Cloud Processing on the Ice Nucleation Abilities of Soot Particles at Cirrus Temperatures, J. Geophys. Res. Atmos., 125, 1-23, https://10.1029/2019jd030922, 2020a.

Vali, G., DeMott, P. J., Möhler, O., and Whale, T. F.: Technical Note: A proposal for ice nucleation terminology, Atmos. Chem. Phys., 15, 10263-10270, https://10.5194/acp-15-10263-2015, 2015.

Zhang, C., Zhang, Y., Wolf, M. J., Nichman, L., Shen, C., Onasch, T. B., Chen, L., and Cziczo, D. J.: The effects of morphology, mobility size, and secondary organic aerosol (SOA) material coating on the ice nucleation activity of black carbon in the cirrus regime, Atmos. Chem. Phys., 20, 13957-13984, https://10.5194/acp-20-13957-2020, 2020.

Citation: https://doi.org/10.5194/acp-2021-111-RC2

---

## Author Comment (AC3)

We thank Referee #4 for his or her comments and constructive suggestions on how to improve the contents of the manuscript. The comments as posted are listed below on green font, our responses to them on red font and the specified modifications to the text on blue font.

The authors have submitted an article titled: Particle emissions from a modern heavy-duty diesel engine as ice-nuclei in immersion freezing mode: an experimental study on fossil and renewable fuels. The article describes an ice nucleation study with combustion exhaust aerosol ice nuclei. The authors used 3 types of fuel and a diesel engine for aerosol generation. The polydisperse aerosol was introduced to a continuous-flow diffusion chamber at a constant RH while ramping the temperature between -43 â•°C and -32 â•°C. Reference aerosol and 2 processing steps of the engine exhaust after-treatment system were intercompared. In addition, the impacts of different atmospheric processing and ageing steps on ice nucleation activity were evaluated. A range of complementary measurements were taken to characterize the aerosol and explain the IN observations. The authors report overall a poor ice-nucleation performance for all the different aerosol types, after-treatment systems, and photochemical aging with minor differences between the 15 experiments. Overall, the article fits within the scope of the journal and has the potential to have environmental relevance. The authors used state of the art instrumentation and decent experimental planning however, there are few points that would need to be addressed before this manuscript is published.

**General comments**

The connection between the selected diesel engine and the atmosphere is not established, not in the introduction, nor in the discussions of the results. The relevance of the specific diesel engine and these fuel types globally is further diminished in the paper. This undermines the justification of the selected MPC conditions for the experiments and the global relevance of this study. Consequently causing vagueness in the conclusions.

We have added the following paragraph to the introduction, and consider that it clarifies the atmospheric relevance of the studied particles:

"Particulate emissions from diesel engines can form a notable fraction of total aerosol burden in urban areas, especially in regions where diesel-powered vehicles outnumber ones using different types of fuel (DeWitt et al., 2014). For instance, such regions can comprise arterial roads near seaports, transport hubs, or other facilities whose function relies on heavy-duty transport. Studies focusing on urban impacts on climate and weather show evidence of precipitation anomalies downwind from large cities (Han et al., 2012; Han et al., 2013; Zhong et al., 2015). Hence, understanding the role of urban aerosol particles in cloud processes is essential for further understanding of the origins of the abovementioned anomalies in precipitation due to urban aerosol particles. However, the number of previous studies focusing on the ice nucleating potential of fossil and renewable diesel fuels (Schill et al., 2016; Chou et al., 2013) is limited, to our knowledge. Understanding the potential changes in atmospheric INP budget due to altering anthropogenic emissions, such as ones reported by Klimont et al. (2017) in this context, is essential for a further understanding of processes that may affect the climate change."

The manuscript lacking substantial information, some of which is split between several other publications e.g. setup details, characterization of the aerosol, size distributions, laminar flow uncertainty etc. (more details in minor comments).

We acknowledge this structural shortcoming of the initial submission and have largely revised the text. Please see our responses to the specific comments.

The selected threshold of 6 micron for ice classification and calculation of activated fraction needs further clarification. This selection and the new approach for calculations of activated fraction are not sufficiently discussed (see minor comments).

The following revisions to address this shortcoming are done:

1) As we investigated the data again and found that the diesel emission particles exhibit no ice activity at sample temperatures greater than -35 $^{o}$C even with lower threshold sizes for separating the ice crystals from droplets. After this discovery, we have revised re-analyzed the ice-nucleation data.
2) We have revised Sect 2.2 (methodology: ice nucleation experiments) to explicitly explain the reasoning behind the choice of 4 μm size threshold I the revised text: please see our responses to the minor comments.

To address the new approach as named in the comment, the following procedure has been performed to address this: We have re-analyzed all SPIN data, and we have identified an 'artifact' in data processing. Whenever the SPIN OPC counts per second exceeds a certain threshold, then the SPIN data are stored in a slightly different manner, which is very simple to correct for. That correction was unfortunately not carried out for the ice-activated fractions (AFs) presented in figures 3 to 6 in the previous version of the manuscript. The correct AFs may be up to about 1 order of magnitude higher than what was previously presented for the very lowest temperatures around -41°C, while the differences in the AFs were relatively smaller or insignificant for temperatures above -39°C. Generally, these corrections do not affect the main findings of little to no indications of heterogeneous immersion freezing. Previously, we interpreted the pronounced low AF levels for several diesel samples near T=-41°C relative to the homogeneous freezing reference to be due to small and/or very hydrophobic diesel particles not activating into cloud droplets with the potential for subsequent immersion freezing inside SPIN. For that reason, we found that there was a risk of misinterpreting low AFs as a result of low immersion freezing ice nucleating ability – rather than an effect of very low CCN activity. That was the motivation for the AF scaling carried out. With the correctly analyzed AFs, there are still indications of low CCN activity influencing the AFs but to a significantly less pronounced level. We still think that it is of relevance to consider how high concentrations of nucleation mode particles and/or very hydrophobic seed particles may bias the immersion freezing AF low by contributing to the total particle concentration while not acting as CCN/INPs. However, we no longer consider it a risk that the 'raw' AFs can be misinterpreted in this context, so we have decided not to carry out the AF scaling and the figures 3.b to 6.b do not appear in the revised version of the manuscript. Consequently, the paragraphs in L215-225 and L273-282 have been deleted.

Given the comprehensive list of instruments used, one would expect to see a deeper analysis. For example, shape factor, effective density, composition accompanied with figures or histograms, intercomparing aerosol properties in this study with other studies or IN activity of similar particles

in previous studies that were mentioned here. This additional information would help to expand the discussion on the observations.

The main focus of the current study is the potential contribution of diesel exhaust emissions to immersion freezing. Gren et al. (2021) report a lot of the suggested physico-chemical particle properties of the very same samples. The presentation and discussion of the CCN results have been expanded in the revised manuscript.

Grammar: in particular sentence structure, fluency, and connection between sentences require significant revisions.

We have revised the text following the comments by all referees and consider the contribution to this comment completed that way.

**Specific comments:**

LN2: "an experimental study" I suggest to change to a laboratory study for clarity

Done.

Ln18: "and" change to "while the…"

Done.

LN19: The tested fuels i.e., EN 590 compliant low-sulfur fossil diesel, hydrotreated vegetable oil (HVO), and rapeseed methyl ester (RME), all were used without blending.

Done.

LN23: "….emitted particles for their physicochemical properties".

Done.

LN23: "We found that the studied particles were poor ice nuclei. The substitution…."

Done.

LN27:  First sentence: reword, don't use "/", what are "impacts on cloud properties" - the sentence is not clear

Following the recommendations by all referees, the text has been changed from:

"Atmospheric aerosols affect the radiative forcing budget of the Earth and thus climate in multiple different ways, directly through absorption/scattering of radiation and indirectly through impacts on cloud properties."

to:

"Atmospheric aerosols affect the energy budget of the Earth and thus climate in different ways: directly through absorption and scattering of heat and light, respectively, and indirectly via affecting cloud formation and lifetime."

LN28: change to "Direct effects can be monitored….", also provide some examples of direct effects

Added aerosol optical thickness and light scattering and absorption as examples.

LN29: "instruments" – provide examples and/or references

Done. We have added sun photometer and lidar as examples of instruments which monitor direct aerosol effects. References to instruments also provided.

LN30: "due to complexity of the processes that contribute to their final effects" – sentence not clear, reword.

The text has been changed from:

"due to complexity of the processes that contribute to their final effects"

to:

"due to complexity of the processes within the clouds that contribute to the total effect"

LN33: "Furthermore, most precipitation events…" - I couldnt find support for this definitive statement in the reference you provided.

We have deleted the statement and the reference from the revised text.

LN38: "than those required"

Done.

LN39: "Particles that are active ice nuclei…"

Done.

LN41: "hydrocarbon fuels have the potential to nucleate ice at temperatures higher.."

Done.

LN41: "than that" – specify what "that" refers to

The two comments above: we have rephrased the sentence from:

"but there is evidence that combustion emissions from different hydrocarbon fuels can have potential as active ice-nucleating particles in temperatures higher than that."

to:

"but there is evidence that combustion emissions from hydrocarbon fuels have the potential to nucleate ice at higher temperatures."

LN43: you haven't defined "INP"

Corrected.

LN44: "on the contrary" – on the other hand?

Corrected.

LN50: add Zhang et al., 2020 to the reference list

Done.

LN50: "these studies taken altogether…" – sentence needs rewording e.g. "The studies mentioned above demonstrate the challenge in associating soot INP properties to the ambient soot particle population".

Sentence corrected as suggested.

LN53: abilities change to activity

Done.

LN54: "Multiple studies such as" – you listed only one study

The text has been changed from:

"The inconsistency in reported observations on soot IN activity implies that it is unclear which types of soot are efficient INPs, and it has been addressed by multiple studies such as Bond et al. (2013)."

To:

"The inconsistency in reported observations on soot IN activity implies that it is unclear which types of soot are efficient INPs, and it has been addressed by multiple studies such as Hoose and Möhler (2012), and Bond et al. (2013)."

We consider that Hoose and Möhler (2012) present an adequate review of soot-IN studies and their results, both positive and negative.

LN54: "In addition to that" - this sentence doesn't add to the previous sentence. Wrong conjunction.

Conjunction changed to "besides".

LN55: " (IPCC) have identified gaps in our knowledge of ice nucleation activity of soot in their…"

Corrected.

LN56: This uncertainty? You haven't mentioned uncertainty, specify what uncertainty you mean

Please see our response to the comment on LN57.

LN56: "..uncertainty reflects to available parameterizations estimating the IN ability of the soot, causing them to range several orders of magnitude" – the context of this sentence is not clear, reword.

Please see our response to the comment below.

LN57: "Consequently, it leads" – what "it" refers to? clarify in the text

We have revised the two sentences mentioned. The text has been changed from:

"This uncertainty reflects to available parameterizations estimating the IN ability of the soot, causing them to range several orders of magnitude (Vergara-Temprado et al., 2018). Consequently, it leads to…"

to:

"The wide range in the reported immersion freezing ability of soot particles is reflected by the available parameterisations spanning several orders of magnitude (Vergara-Temprado et al., 2018)." Consequently, the variation leads to…"

LN58: "challenges when the potential…" – rephrase e.g. "high uncertainty in estimation of the radiative forcing via modeling"

Rephrased as suggested.

LN69-71: reword the sentence

The text has been changed from:

"The introduction of the advanced after-treatment systems, such as the DPF and diesel oxidation catalyst (DOC), have led to a decline of 20-65% in black carbon (BC) emissions from heavy-duty road diesel engines in developed markets between 1990-2010 (Klimont et al., 2017). "

to:

"Klimont et al. (2017) reported a decline of 20-65% in black carbon (BC) emissions from heavy-duty road diesel engines in developed markets between 1990-2010, during the period when advanced emission after-treatment systems, such as the DPF and diesel oxidation catalyst (DOC), became mandatory in new diesel-powered road vehicles."

LN72: globally still widely used – reference? Can you provide global estimates that will support the relevance of your study?

Please see our response to the comment on LN74.

LN73: "long time" - please support this qualitative projection with a reference

Please see our response to the comment below.

LN74: "still about to maintain its global popularity for decades" – reference for this forecast?

Following the suggestion by Referee #1, we have revised the entire paragraph to be more concise, and more focusing on 1) the atmospheric relevance of diesel emissions and 2) the motivation for ice nucleation studies, such as this one. Please check the revised chapter where the commented aspects are either removed or corrected.

LN75: I think there is a missing paragraph here that connects between diesel combustion emissions at ground level and how they reach and interact with atmospheric humidity and temperature to form clouds, what's their known fraction at different altitudes etc. In what clouds they are most predominant to support your choice of temperature range. Perhaps connection to airborne measurements of combustion emissions and their ability to nucleate ice at altitude detected in flight e.g. Brown, 2018.

The following paragraph has been added to the paragraph describing the effects of diesel emissions:

"This diesel vehicle exhaust is emitted into the planetary boundary layer (PBL) close to the ground level. From mid to high latitudes, low-level mixed-phase clouds may be present within the PBL (e.g. Gierens et al., 2020) and can potentially get influenced by diesel vehicle emissions. Alternatively, soot particles can be transported to higher altitudes and over long distances (Storelvmo, 2012), so they can potentially influence clouds far from the source region. Soot particles have been identified in ice crystal residues in mixed-phase clouds (Cozic et al., 2008; Ebert et al., 2011), and thus it is of relevance to study their potential to facilitate immersion freezing."

LN76: the sentence in not clear and is it relevant to this study?

Please check the description of the entirely revised paragraph commented in the comment on LN74.

LN78: diesel combustion emissions

Changed.

LN79: "among other factors, remains less studied" – what other factors and why it's less studied?

We have revised the paragraph as a whole (see our response to comment on LN74): in this revised text, the mentioning of other factors has been removed and the following sentences explain why there are fewer studies on immersion-freezing INP potential of diesel emissions.

LN84: "as well as they impact" – reword

We have moved the description of HVO and RME fuels to Sect 2.1, following the recommendations to improve the structure of the manuscript.

LN91: "temperatures of -35 â•°C and -30 â•°C. No immersion freezing…"

Done.

LN93:  remove ice-nucleating potential

Done.

LN104: alternatives in the near future

Corrected.

LN104: "it is likely that heavy-duty diesel engines will be in use further than that" – here you say likely in LN73 you were much more decisive

We have removed the statement on LN73 due to more comprehensive revision of the entire chapter, please see our response to comments regarding lines 72-74.

LN106: "We investigate…" – reword this sentence

The text has been changed from:

"We investigate the immersion freezing ability at temperatures down to where homogenous freezing dominates, to detect potentially low ice-nucleating abilities."

to:

"Our ice nucleation experiments aimed in detection of potentially low ice nucleating abilities in immersion freezing mode, down to temperatures where homogeneous freezing starts to dominate."

LN111: did you monitor the temperature and humidity in the sampling line, if so, where? Please add to figure 1. Did you monitor pressure and airflow in the different sections of your setup? Would the high concentration cause sedimentation and narrowing of inner tubing diameter? Did that affect your measurements during the experiments?

It is very expectable that the sample temperature has been equal to room temperature after PTD and ED dilution by a factor of 1:600-1200 in total, prior to entering any instruments. Moreover, we used ¼" stainless steel tubing in our set-up between the last ED and the SPIN which should further assure that the sample temperature has been equal to room temperature in the SPIN experiments. Given the total sample flow rate of several liters per minute, we expect no gravitational sedimentation for submicron particles inside the tubing.

LN112: "The test engine used was a six-cylinder inline Scania D13 heavy-duty diesel engine" - why this engine was selected? How representative it is of diesel engines in the world? You should provide few more details to establish how this experiment will provide conclusions relevant to real world (outside the laboratory).

The Scania D13 series comprises commercial and industrial diesel engines ranging 270-390 kW in power output, which is a typical size for modern diesel engines used in heavy road vehicles and we consider it a representative option for testing diesel engine emissions.

LN115: "approach" – perhaps setup or configuration?

Changed to "configuration".

LN116: what is one full temperature scan? How many repetitions did you do to test repeatability?

With one full temperature scan we mean scanning over chosen freezing conditions at constant RH on water (in this study $RH_w$ = 110%), over a temperature range that was -43 to -32 $^oC$ in all experiments – this is later quantified on lines 180-181 in the revised text. Terms 'temperature scan' or 'T-scan' are commonly used in the CFDC community for this scan-type experimentation.

LN122: "were cooled down" – did you monitor the temperature? Where? How?

We used PTD+ED cooling with a total dilution ratio of 1:600-1200, using dry compressed air as diluter. Therefore, we consider that such a high dilution ratio ensures a significant cooling of the sample aerosol at any rate.

LN124: PAM – reference for the instrument?

The following details about the PAM reactor used in this study have been added to the text:

"The PAM reactor used in this study consisted of a 13 L steel chamber containing two Hg lamps with peak intensities at 185 and 254 nm. The UV light generates ozone and hydroxyl radicals (OH) that oxidize the aerosol as it moves through the chamber. The flow rate through the PAM was controlled to 5–7 L min-1. The same UV light intensity was used in all experiments. The aging corresponded to 4.8 ± 2.6 days assuming an average OH concentration of 1.5 × 106 molecules cm-3 (Mao et al., 2009). Extensive SOA formation occurred in the PAM for all engine out experiment, while less SOA was formed in measurements after the DOC. The thermodenuder (Aerodyne Inc.,) was held at 250 $^oC$ in all experiments where it was used."

LN127: RH<10 , where and how it was measured?

Given the high dilution ratio of 1:600-1200 with dry (dew point -40 $^oC$) compressed air, it can be expected that the dew point of the sample flow downstream of the last ED has a $RH_{water}$ way below 10%.

LN129: "The latter scanned continuously" - you mentioned only SMPS so not clear what is the "latter".

We have replaced "The latter" with "The SMPS".

LN130: "The size range of SMPS…" - The mobility diameter sampling range of the SMPS was set between 11 and 500 nm with automatic multiple charge correction in the software.

Corrected as suggested.

LN131: CCNc-100 ? one column?

Yes, that was the instrument we used. the following change has been made from:

"(CCNC, Droplet Measurement Technologies)"

to:

 "(CCNC, Droplet Measurement Technologies, CCN-100)"

LN138: remove Moreover

Done.

LN142: RH<5, in line127 you said RH<10, was it measured?

The *RH* value on line 127 means the situation downstream of the second ejector dilution. There was an additional molecular sieve dryer upstream of the CFDC, which dried the sample aerosol further to <5% *RH*. This statement is based on the calibration of 60cm mol sieve dryer (marked as 'dryer' in Fig. 1), according to which a freshly regenerated desiccant material (13x mol sieves) is capable of drying room air ($T$ = 20 °C, RH = 40-50%) down to dew point of -40 °C with air flow rate of 2.05 SLPM (the SPIN 1 + CPC 1.05).

LN148: "A low level of exhaust gas recirculation (EGR) setting was used, 18% oxygen on intake air to the combustion cylinder" - why these settings were used? Are they common settings?

To limit the formation of $NO_x$ emissions, different levels of EGR are applied depending on engine operating conditions. 18% oxygen is representative for the operating conditions used during the experiments.

LN171: remove in the orders of

Done.

LN184: "Thus, we consider observed particles larger than 6 µm ice crystals" - what about ice smaller than 6um? For example, if you have pore filling happening as described. in Mahrt et al. 2018, where they chose 1 µm size threshold to detect ice crystals in their chamber. How would such shift in this threshold affect your results?

Following the comment by Referee #2 about the same issue, we have revised Sect 2.2 in a way that it explicitly explains the behavior of droplets and ice crystals inside the SPIN, when the settings used in this study are applied. Besides, the reasoning for using the 6 µm size threshold in size-separation is now explained.

LN199: "such as particle losses at the laminar flow are discussed in detail by Korhonen et al. (2020)" - are discussed in Korhonen (2020) but how do you address these deviations from laminarity in your

study? or any of the other issues discussed in the conclusions of Garimella et al 2017. If you dont, how much uncertainty it introduces to your results? are they still valid despite those known issues of SPIN?

Korhonen et al. (2020) discusses the operation of the very same SPIN unit in a manner similar to this study (T scans), thus it can be expected that the systematic errors are unchanged between these two sets of ice-nucleation measurements. In the SPIN, the particle losses due to lamina spreading are a well-acknowledged, yet not well-constrained issue as it is concluded in both Korhonen et al. (2020) and Garimella et al. (2017). Therefore, we prefer not to speculate and introduce arbitrary correction factors to the ice-nucleation data of this study. Given the high experiment reproducibility as reported by Korhonen et al. (2020), we consider the results of this study valid.

LN201: 15 experiments – did you do any repetitions?

Yes, we did whenever the pre-planned operation cycle of the engine allowed to repeat the upward T scan without changing any sampling parameters. For instance, the reported fossil diesel + DOC + bypass and HVO + DOC + bypass runs, two of each, are experiment repetitions.

LN202: "SPIN, and polydisperse" – split into 2 sentences

Done.

LN220: switching back and forth between CFDC and SPIN, stick with one naming for the instrument

We have revised the naming as follows: before introduction of the ice nucleation measurements we refer to CFDCs in more general sense, after that the SPIN is always mentioned as the name of the unit.

LN220: "We expect this particle fraction to be dominated by the larger particles, since they are more likely to act as CCN" – what about the doubly charged particles selected with DMA, how they affect this measurement?

The comment only relates to the poly-disperse aerosol samples studied with SPIN. Hence, there was no selection with a DMA prior to those SPIN measurements.

LN223: "The normalization was calculated via averaging the five highest observed ice-activation..." – do you expect to measure only one nucleation mechanism in these temperatures e.g. Marcolli 2014?

Yes. We expect homogeneous freezing of formed droplets at lamina temperatures significantly below homogeneous freezing temperature that was empirically observed to take place at -38.9 $^{\circ}$C in the SPIN.

LN237: "In many cases, the supersaturation did not suffice to activate the studied particles, and results are presented only when a full CCN spectrum could be identified" - do these "many cases" have impact on the setting of 6um threshold for ice in SPIN, which was set mostly because of droplets?

The ice size threshold is determined from the SPIN data analysis and has in the revised manuscript been changed from 6 to 4 μm. We apply the same ice size threshold for the homogeneous freezing reference and the other samples, and the measurements with the CCN counter did not play a role in this context.

LN243: "to elemental carbon (EC) concentrations" – how was EC measured?

The following text has been added:

"The EC was assessed by sampling of undiluted exhaust on quartz filters (Pallflex Tissuequartz, 47 mm), thermal-optically measured according to EUSAAR_2 protocol (Cavalli et al., 2010)."

LN253: "was calculated from SP-AMS and aethalometer data" – what is the propagated uncertainty combining the measurements for calculation of this ratio?

With respect to the accuracy of this ratio, we estimate approximately 50% (95% CI). However, the precision, which is more important for our study as we seek correlations between ice nucleation ability and other physicochemical parameters, is much better. Yet, upon revision, we find that the third digit in the OA/eBC ratio is indeed influenced by noise and hence, we have removed it. It is our assessment that noise did not influence the ordering or these samples and that the lack of correlation with ice nucleation abilities is not due to noise in AE33 and SP-AMS data."

LN262: "Gren et al. (2021) present a more comprehensive description of the particle size distributions" - There are references to at least 3 other papers that contain substantial information about this study and the reviewer/reader needs to read these to understand this paper e.g. Gren et al. 2021, Kristensen et al. 2020 Korhonen et al. 2020. Some of the information specific to this experiment is missing and needs to be added either as main text, as appendix, or supplementary material.

We acknowledge this issue and have decreased the dependency on other papers using the following actions: First, the ice nucleation pathway inside the SPIN is explained in the Sect 2.2 of the revised manuscript, in a way that it is independent from Korhonen et al. (2020). Second, we have added a deeper discussion on the results supporting the ones from ice nucleation, in a way that understanding their relevance to ice-nucleating abilities of the particles studied does not require reading the abovementioned papers.

LN268: "significant fraction" – how would you quantify this statement?

It is not straightforward to quantify that effect, since it is particle size dependent. For clarification, we have extended the description from:

"Thermodenuder treatment (250°C) removed a significant fraction of the SOA emissions."

to:

"Thermodenuder treatment (250°C) removed a significant fraction of the OA emissions for both fresh and aged aerosol, as indicated by the differences between the respective particle number size

distributions presented in Fig. 2, and further discussed in relation to the presentation of CCN results below."

LN272: what is reasonable detection sensitivity?

When the instrument background signal is close to zero, a number concentration of 1200 cm$^{-3}$ would yield a minimum detection sensitivity in the order of AF = 10$^{-5}$. This is mentioned in the revised text.

LN273: "negligible, in the order of 10$^{-4}$ from total sampled concentration" - how this number compares to activated fraction, could all INP be larger than 250nm?

It is very likely that all ice crystal formation was induced by the largest particles in each sample.

LN273: "Due to this" - Here you claim that most particles were small thus this approach for activated fraction is valid. If the IN are all large, would this approach still hold?

Due to the correct re-analysis of the ice-activated fractions, as described in more detail above, our interpretation of the potential effects of small and/or hydrophobic particles has changed. Consequently, L.273-281 have been deleted in the revised manuscript.

LN274: remove "calculating"

Done.

LN284: In addition, we estimate…

Corrected.

LN334: This short section should include deeper analysis, providing more than 1-2 sentences per instrument. For example you describe APM and effective density in the experimental section but I dont see effective density mention in the results, tables, or in the discussions.

The discussion of the CCN results has been extended significantly. Also, the particle effective density is discussed in the revised manuscript.

LN354: "Besides, our results are in complete agreement with Kanji et al. (2020) who studied…" - how propane combustion aerosol and commercial black carbon particles relevant to this study?

We have replaced the statement from:

"Besides, our results are in complete agreement with Kanji et al. (2020) who studied whether hydrocarbon soot from propane combustion and different commercially available black carbon particles can induce immersion freezing and found all of them inefficient ice-nucleators."

to:

"Our results, in conjunction with a lot of previous studies indicate that a wide range of black carbon and soot particle types are inefficient as immersion freezing INPs"

LN357: "was found on the other fuels, HVO and RME" - where is the discussion part of the results? what is the possible explanation for this difference?

We have changed L357 from:

"We also found that photochemical aging increased the IN activity on fossil diesel, but no distinguishable effect was found on the other fuels, HVO and RME."

to:

"We found that photochemical aging had minor if any effects on the ice-activated particle fractions."

LN370-376: "It is worth mentioning that all experiments in this study were conducted under well-controlled laboratory…" – The authors start the article by suggesting there is an importance for such study in MPC conditions. From this paragraph, if I understand correctly, the authors conclude that their results can't contribute much to our understanding of interactions of diesel combustion emissions in MPC and real atmospheric environment?

Diesel engine emissions comprise a significant fraction of anthropogenic particles in many environments. Hence, it is very important and relevant to investigate their potential impact on clouds and climate. However, it is not possible to capture all potential aspects and complex properties of atmospheric diesel engine emissions within a single study. Therefore, this study can be considered an important contribution to the understanding of the potential effects of diesel emissions on mixed-phase clouds. However, further experimental studies are needed before overall conclusions can be made. That is the reasoning behind the formulation.

LN545-547: switch places

The reference to Mülmenstadt et al. (2015) has been deleted during the revision process.

LN595: Fossil diesel has twice more 400nm particles, would you expect it to affect the comparison to HVO and RME?

The particle number size distributions presented in Fig. 2 represent the undiluted conditions, so the absolute concentrations are not of direct relevance to the SPIN measurements. However, it does appear as if the relative number fraction of 400 nm particles was slightly higher for the diesel emissions relative to the other fuels. In the revised manuscript, we provide estimates of the ice nucleation active surface site (INAS) densities in order to account for differences in surface areas between the samples.

LN598: "see Gren et al. (2021) for a more detailed analysis on emission particle properties of this study" - this sentence is not clear to me. Why more detailed analysis of this study is in another paper? is this an accompanying paper (part1)?

This study, as well as Gren et al. (2021), was a part of an extensive measurement campaign that produced several studies with different foci. We explain this in the introduction of the revised manuscript – the following text has been added:

"This study was a part of an extensive measurement campaign where the particulate emissions from a modern commercial heavy-duty diesel engine that was modified for single-cylinder operation were studied with multiple different foci and more information about the characteristics of the studied particles are presented in separate studies. For instance, Gren et al. (2021) studied the properties of particulate and gas-phase emissions of the test fuels, and how different emission after-treatment systems such as DOC and DPF affected them. This study focuses on immersion freezing ice nucleating abilities of the particulate emissions studied in this experiment campaign."

LN651 Table2: For the values presented with accuracy to 3 decimal places, what is the estimated error on these?

We acknowledge that using 3 digits was excessive and present the values with two digits in the revised text. According to Gren et al. (2021), the repeatability of BC emission factors between experiments was within 25%.

**Suggested references:**

- Brown, A.P. "Contrail Flight Data for a Variety of Jet Fuels," AIAA 2018-3188. 2018 Atmospheric and Space Environments Conference. June 2018.
- Garimella, S., Rothenberg, D. A., Wolf, M. J., David, R. O., Kanji, Z. A., Wang, C., Rösch, M., and Cziczo, D. J.: Uncertainty in counting ice nucleating particles with continuous flow diffusion chambers, Atmos. Chem. Phys., 17, 10855–10864, doi:10.5194/acp-17-10855-2017, 2017.
- Mahrt, F., Marcolli, C., David, R. O., Grönquist, P., Barthazy Meier, E. J., Lohmann, U., and Kanji, Z. A.: Ice nucleation abilities of soot particles determined with the Horizontal Ice Nucleation Chamber, Atmos. Chem. Phys., 18, 13363–13392, https://doi.org/10.5194/acp-18-13363-2018, 2018.
- Marcolli, C.: Deposition nucleation viewed as homogeneous or immersion freezing in pores and cavities, Atmos. Chem. Phys., 14, 2071–2104, https://doi.org/10.5194/acp-14-2071-2014, 2014.
- Zhang, C., Zhang, Y., Wolf, M. J., Nichman, L., Shen, C., Onasch, T. B., Chen, L., and Cziczo, D. J.: The effects of morphology, mobility size, and secondary organic aerosol (SOA) material coating on the ice nucleation activity of black carbon in the cirrus regime, Atmos. Chem. Phys., 20, 13957–13984, https://doi.org/10.5194/acp-20-13957-2020, 2020.

**Citation**: https://doi.org/10.5194/acp-2021-111-RC3

---

## Author Response (AR1)

Dear Editor,

we have performed the revisions, and addressed all comments in the three referee reports we received. Below are the most significant revisions:

- we have re-analysed all ice-nucleation data from the SPIN
- more in-depth analyses of supportive observations (CCNC, SP-AMS, etc.), also descriptions of those measurements improved
- ice nucleation active density model added to interpretation of results
- we have revised all figures due to re-analysis of the ice-nucleation data